# Stable water isotopes and accumulation rates in the Union Glacier region, Ellsworth Mountains, West Antarctica over the last 35 years

Kirstin Hoffmann[1,2], Francisco Fernandoy[3], Hanno Meyer[2], Elizabeth R. Thomas[4], Marcelo Aliaga[3], Dieter Tetzner[4,5], Johannes Freitag[6], Thomas Opel[2,7], Jorge Arigony-Neto[8], Christian Florian Göbel[8], Ricardo Jaña[9], Delia Rodríguez Oroz[10], Rebecca Tuckwell[4], Emily Ludlow[4], Joseph R. McConnell[11], Christoph Schneider[1]

[1]Geographisches Institut, Humboldt-Universität zu Berlin, Unter den Linden 6, Berlin, 10099, Germany
[2]Alfred Wegener Institute, Helmholtz Centre for Polar and Marine Research, Research Unit Potsdam, Telegrafenberg A45, Potsdam, 14473, Germany
[3]Facultad de Ingeniería, Universidad Nacional Andrés Bello, Viña del Mar, 2531015, Chile
[4]Ice Dynamics and Paleoclimate, British Antarctic Survey, High Cross, Cambridge, CB3 0ET, United Kingdom
[5]Department of Earth Sciences, University of Cambridge, Downing Street, Cambridge, CB2 3EQ, United Kingdom
[6]Alfred Wegener Institute, Helmholtz Centre for Polar and Marine Research, Am Alten Hafen 26, Bremerhaven, 27568, Germany
[7]Department of Geography, Permafrost Laboratory, University of Sussex, Falmer, Brighton, BN1 9QJ, United Kingdom
[8]Instituto de Oceanografia, Universidade Federal do Rio Grande, Av. Itália, km 8, CEP 96201900, Rio Grande, RS, Brazil
[9]Departamento Científico, Instituto Antártico Chileno, Plaza Muñoz Gamero 1055, Punta Arenas, Chile
[10]Facultad de Ingeniería, Universidad del Desarrollo, Avenida Plaza 680, Santiago, Chile
[11]Division of Hydrologic Sciences, Desert Research Institute, 2215 Raggio Parkway, Reno, NV 89512, USA

*Correspondence to*: Kirstin Hoffmann (Kirstin.Hoffmann@awi.de)

## Abstract

Antarctica is well-known to be highly susceptible to atmospheric and oceanic warming. However, due to the lack of long-term and in-situ meteorological observations, little is known about the magnitude of the warming and the meteorological conditions in the intersection region between the Antarctic Peninsula (AP), the West Antarctic Ice Sheet (WAIS) and the East Antarctic Ice Sheet (EAIS). Here we present new stable water isotope data ($\delta^{18}$O, $\delta$D, d excess) and accumulation rates from firn cores in the Union Glacier (UG) region, located in the Ellsworth Mountains at the northern edge of the WAIS. The firn core stable oxygen isotopes and the d excess exhibit no statistically significant trend for the period 1980-2014 suggesting that regional changes in near-surface air temperature and moisture source variability have been small during the last 35 years. Backward trajectory modelling revealed the Weddell Sea sector, Coats Land and DML to be the main moisture source regions for the study site throughout the year. We found that mean annual $\delta^{18}$O ($\delta$D) values in the UG region are negatively correlated with sea ice concentrations (SIC) in the northern Weddell Sea, but not influenced by large-scale modes of climate variability such as the Southern Annular Mode (SAM) and the El Niño-Southern Oscillation (ENSO). Only mean annual d excess values show a weak positive correlation with the SAM.

On average annual snow accumulation in the UG region amounts to 0.245 m w.eq.a$^{-1}$ in 1980-2014 and has slightly decreased during this period. It is only weakly related to sea ice conditions in the Weddell Sea sector and not correlated with SAM and ENSO.

We conclude that neither the rapid warming nor the large increases in snow accumulation observed on the AP and in West Antarctica during the last decades have extended inland to the Ellsworth Mountains. Hence, the UG region, although located at the northern edge of the WAIS and relatively close to the AP, exhibits rather stable climate characteristics similar to those observed in East Antarctica.

## 1. Introduction

Antarctic temperature change has been a major research focus in the past decades. Despite the scarcity and short duration of the observations, it shows a contrasting regional pattern between the Antarctic Peninsula (AP), the West Antarctic Ice Sheet (WAIS) and the East Antarctic Ice Sheet (EAIS; Stenni et al., 2017).

Both the AP and the WAIS have experienced significant atmospheric and oceanic changes during recent decades. The WAIS is considered as one of the fastest warming regions on Earth based on the analysis of meteorological records (Steig et al., 2009; Bromwich et al., 2013) and ice cores (Steig et al., 2013; Stenni et al., 2017). Bromwich et al. (2013) reported for central West Antarctica an increase in annual air temperature by more than 2°C since the end of the 1950s. The rapid warming in West Antarctica at the end of the last century is anomalous, but seems to be not unprecedented in the past 300 years (Thomas et al., 2013) and even in the past two millennia (Steig et al., 2013), respectively. The significant increase in near-surface air temperatures is accompanied by regionally different trends in accumulation rates. Snow accumulation in coastal regions of the eastern WAIS has experienced a dramatic increase during the 20th century that is unprecedented in the past 300 years (Thomas et al., 2015; Medley and Thomas, 2019). In contrast, statistically significant negative trends have been found across the central and western parts of the WAIS (Burgener et al., 2013; Medley and Thomas, 2019).

For the AP, time series of near-surface air temperature from weather stations (Vaughan et al., 2003; Turner et al., 2005a) as well as from ice-core stable water isotope records (Thomas et al., 2009; Abram et al. 2011; Stenni et al., 2017) provide evidence of a significant warming over the last 100 years reaching more than 3°C since the 1950s. Contemporaneously, precipitation and snow accumulation have significantly increased during the 20th century at a rate that is exceptional in the past 200-300 years (Turner et al., 2005b; Thomas et al., 2008; Thomas et al., 2017; Medley and Thomas, 2019). The rapidity of the 20th century warming of the AP is unusual, but not unprecedented in the context of late-Holocene natural climate variability (i.e. 2000 years BP; Mulvaney et al., 2012). Furthermore, Turner et al. (2016) revealed that air temperatures on the AP have decreased since the late 1990s, contrasting the warming of previous decades.

Contrary to the AP and the WAIS, the EAIS has experienced rather a cooling or constant climate conditions in recent decades (Turner et al., 2005a; Nicolas and Bromwich, 2014; Smith and Polvani, 2017; Goursaud et al., 2017). However, Steig et al. (2009) reported slight, but statistically significant warming in East Antarctica at a rate of + 0.10±0.07°C per decade (1957-2006) similar to the continent-wide trend. A positive and significant near-surface air temperature trend has also been found for Dronning Maud Land (DML) for the last 100 years (Stenni et al., 2017) and in particular for the period 1998-2016 (+1.15±0.71°C per decade; Medley et al., 2018). Snowfall and accumulation rates on the EAIS are usually considered to show no significant changes (Monaghan et al., 2006) or clear overall trends due to the counterbalancing of increases and decreases in different regions (van den Broeke et al., 2006; Schlosser et al., 2014; Altnau et al., 2015; Phillipe et al., 2016). However, more recent studies provide a different picture: based on an ice core from western DML, Medley et al. (2018) derived a significant increase in snowfall since the 1950s, which is unprecedented in the past two millennia. Furthermore, Medley and Thomas (2019) show that snow accumulation over the EAIS has steadily increased during the 20th century, despite a decrease since 1979.

Hence, there is no conclusive evidence about general trends in near-surface air temperature and accumulation rates in Antarctica. In summary, the detection and assessment of trends in climate variables, such as air temperature and precipitation (accumulation), for the three Antarctic regions - AP, WAIS and EAIS - is challenging, due to the shortness of available instrumental records and the incompatibility between climate model simulations and in-situ observations (Jones et al., 2016; Stenni et al., 2017). In addition, determined trends are often regionally and/or seasonally contradicting on interannual to decadal or multiannual timescales, and are at the same order of magnitude as the associated uncertainties.

Factors affecting mechanisms that force the anomalously strong and rapid warming of the AP and the WAIS with the associated precipitation changes as well as the contrasting constant air temperatures on the EAIS have been widely discussed: For the AP

the significant warming and increase in snow accumulation has been linked to a shift of the Southern Annular Mode (SAM) towards its positive phase during the second half of the 20$^{th}$ century (Thompson and Solomon, 2002; Turner et al., 2005b; Gillett et al., 2006; Marshall, 2006; Marshall et al., 2007; Thomas et al., 2008). The recently observed decrease in air temperature has in turn been attributed to an increased cyclonic activity in the northern Weddell Sea (Turner et al., 2016). The

SAM is the principal zonally-symmetric mode of atmospheric variability in extra-tropical regions of the Southern Hemisphere (Limpasuvan and Hartmann, 1999; Thompson and Wallace, 2000; Turner, 2004). During its positive phase the mid-latitude westerlies are strengthened and shifted poleward over the Southern Ocean leading to increased cyclonic activity and advection of warm and moist air towards Antarctic coastal regions (Thompson and Wallace, 2000; Thompson and Solomon, 2002; Turner, 2004; Gillett et al., 2006). Consequently, a positive SAM is associated with a warming on the AP and anomalously

low air temperatures over eastern Antarctica and the Antarctic plateau (EAIS). For a negative SAM the opposite is true, i.e. a cooling on the AP and exceptionally high air temperatures over the EAIS. Hence the slight cooling of the EAIS during recent decades is connected to a more positive SAM (Turner et al., 2005a; Stenni et al., 2017).

The rapid warming of the WAIS and the contemporaneous precipitation trends have been suggested to be driven by sea surface temperature (SST) anomalies in the central and western (sub)tropical Pacific (e.g. Schneider et al., 2012; Ding et al., 2011;

Steig et al., 2013; Bromwich et al., 2013). Furthermore, the near-surface air temperature and precipitation trends observed on the WAIS seem to be linked to the recent deepening of the Amundsen Sea Low (ASL) influencing meridional air mass and heat transport towards West Antarctica (Genthon et al., 2003; Bromwich et al., 2013; Hosking et al., 2013; Raphael et al., 2015; Burgener et al., 2013; Thomas et al., 2015). Changes in the absolute depth of the ASL are strongly related to the El Niño Southern Oscillation (ENSO) and the SAM (Raphael et al., 2015). ENSO is the largest climatic mode on Earth on decadal and

sub-decadal timescales originating in the tropical Pacific. ENSO directly influences the weather and oceanic conditions across tropical, mid- and high-latitude areas on both hemispheres (Karoly, 1989; Diaz and Markgraf, 1992; Diaz and Markgraf, 2000; Turner, 2004; L'Heureux and Thompson, 2006). The near-surface air temperature variability of both East and West Antarctica has been linked to the variability of ENSO, although not showing any consistent trend on interannual timescales (Rahaman et al., 2019).

Current trends in Antarctic climate and their drivers are still not completely understood, especially on regional scales. Consequently, there is a strong need for extended observations and monitoring in all regions of Antarctica. Therefore, data on meteorological parameters such as air temperature, precipitation (accumulation), moisture sources and transport pathways of precipitating air masses are vital to assess past and recent changes of Antarctic climate. Direct observations of these parameters are lacking for most of Antarctica. Thus, proxy data derived from firn and ice cores, e.g. stable water isotopes, provide

important information on past and recent climate variability on local to regional scales (Thomas and Bracegirdle, 2015). For the region at the intersection of the AP, the WAIS and the EAIS data are especially sparse as little or no long-term meteorological records are available for this part of the Antarctic continent (Stenni et al., 2017; Thomas et al., 2017). The region is located at the transition to the Ronne-Filchner Ice Shelf, for which a recent modelling study suggests a high susceptibility to destabilization and disintegration under a warming climate (Hellmer et al., 2012; Hellmer et al., 2017). Similar

has been already observed for ice shelves around the AP and the WAIS (e.g. Pritchard and Vaughan, 2007; Cook and Vaughan, 2010; Scambos et al., 2014; Rignot et al., 2014; Joughin and Alley, 2011).

This study aims at improving our understanding of regional climate variability at the intersection of the AP, the WAIS and the EAIS based on firn-core stable water isotope data from Union Glacier (UG), located in the Ellsworth Mountains at the northern edge of the WAIS (79°46'S, 83°24'W; 770 m above sea level [asl]; Fig. 1a). The UG region has not been intensively

investigated before. Rivera et al. (2010, 2014) mapped the surface and subglacial topography and determined ice-dynamical and basic glaciological characteristics of UG. Meteorological and stake measurements yielded a mean daily air temperature of -20.6°C (2008-2012) and a mean snow accumulation of 0.12 m w.eq.a$^{-1}$ (2008-2009). However, the available data records are very short and do not allow conclusions on long-term trends.

In this study we use high-resolution density and stable water isotope data of six firn cores drilled at various locations in the UG region for reconstructing accumulation rates and inferring connections to recent changes in meteorological parameters, such as air temperature, on local to regional scales. We further investigate how these variables are related to temporal changes of moisture source regions, sea ice extent and concentration (SIE and SIC) and atmospheric modes such as SAM and ENSO. Backward trajectory analysis is applied to determine potential source regions and transport pathways of precipitating air masses reaching the UG region. We aim to characterize the UG region with isotope geochemical methods in order to place it in the regionally diverse pattern of Antarctic climate variability. The main focus of this study lies on the following question: Do the UG region and surrounding areas experience the same strong and rapid air temperature and accumulation increases as observed for the neighbouring AP and WAIS and, if yes, to what extent? Or does the UG region follow the rather constant air temperature and accumulation conditions as observed for most of the EAIS?

## 2. Data and Methodology

### 2.1 Fieldwork, sample processing and analysis

Two field campaigns were conducted in the UG region in austral summers 2014 and 2015. Union Glacier is one of the major outlet glaciers within the Ellsworth Mountains and flows into the Ronne-Filchner Ice Shelf in the Weddell Sea sector of Antarctica (Fig. 1). It is composed of several glacier tributaries covering a total area of 2561 km$^2$. UG has a total length of 86 km, a maximum ice thickness of 1540 m and a maximum depth of the snow-ice boundary layer of 120 m. The subglacial topography of the glacier valley is smooth with U-shaped flanks. The bedrock is located below sea level (-858 m; Rivera et al., 2014).

We examine six firn cores (GUPA-1, DOTT-1, SCH-1, SCH-2, BAL-1 and PASO-1) retrieved at different locations (Fig. 1b), ranging between 760 m asl (GUPA-1 and DOTT-1) and 1900 m asl (PASO-1) in altitude, using a portable solar-powered and electrically-operated ice-core drill (Backpack Drill; icedrill.ch AG). GUPA-1 was drilled near the ice-landing strip on UG. DOTT-1 originates from an ice rise towards the Ronne-Filchner Ice Shelf. SCH-1 was retrieved from the ice divide between the glaciers Schneider and Schanz. SCH-2 and BAL-1 were both taken in U-shaped glacial valleys (Glacier Schneider and Glacier Balish). PASO-1 originates from a plateau west of the Gifford Peaks (Fig. 1b). Details on the drill locations and basic core characteristics are given in Table 1.

For firn cores BAL-1 and PASO-1 high-resolution (< 1 mm) density profiles were obtained using X-ray microfocus computer tomography (ICE-CT; Freitag et al., 2013) at the ice-core processing facilities of the Alfred Wegener Institute, Helmholtz Centre for Polar and Marine Research (AWI) in Bremerhaven, Germany. The cores were sampled at 0.025 m resolution and analysed for stable water isotopes using a cavity ring-down spectrometer (L2130-*i*; Picarro Inc.) coupled to an auto-sampler (PAL HTC-xt; CTC Analytics AG) at the Stable Isotope Laboratory of the AWI in Potsdam, Germany. Stable water isotope raw data were corrected for linear drift and memory effects following the procedures suggested by van Geldern and Barth (2012) using six repeated injections per sample, from which the first three were discarded. The drift- and memory-corrected stable water isotope compositions were calibrated with a linear regression analysis using four different in-house standards that have been calibrated to the international VSMOW2 (Vienna Standard Mean Ocean Water)/SLAP2 (Standard Light Antarctic Precipitation) scale. Stable water isotope ratios are reported in per mil (‰) versus VSMOW2. Precision of the measurements is ±0.08 ‰ for $\delta^{18}O$ and ±0.5 ‰ for $\delta D$.

For firn cores GUPA-1, DOTT-1, SCH-1 and SCH-2 density profiles were constructed by section-wise determining the core volume and weight. Accordingly, average resolution of density profiles is 0.25 m for GUPA-1, 0.4 m for DOTT-1, 0.27 m for SCH-1 and 0.78 m for SCH-2. Cores GUPA-1, DOTT-1 and SCH-1 were sampled at 0.05 m resolution for stable water isotope analysis carried out at the Stable Isotope Laboratory of the Universidad Nacional Andrés Bello (UNAB) in Viña del MarChile. For the measurements an off-axis integrated cavity output spectrometer (TLWIA 45EP; Los Gatos Research) was used with a precision of 0.1 ‰ for $\delta^{18}O$ and 0.8 ‰ for $\delta D$ (Fernandoy et al., 2018). Each sample was measured twice in different days

using ten repeated injections, from which the first four were discarded on each measurement. Stable water isotope raw data were corrected for linear drift and memory effects and then normalized to the VSMOW2/SLAP2 scale using the software LIMS (Laboratory Information Management System; Coplen and Wassenaar, 2015). For data normalization three different in-house standards and one USGS standard (USGS49) calibrated to the VSMOW2/SLAP2 scale were used.

Core SCH-2 was analysed at the British Antarctic Survey (BAS) in Cambridge, UK. Stable water isotopes were determined at 0.05 m resolution using a Picarro L2130-$i$ analyser with measurement precision of 0.08 ‰ for $\delta^{18}O$ and 0.5 ‰ for $\delta D$. Major ions (Cl$^-$, NO$_3^-$, SO$_4^{2-}$, MSA$^-$, Na$^+$, K$^+$, Mg$^{2+}$, Ca$^{2+}$) were measured at 0.05 m resolution using a Dionex reagent-free ion chromatography system (ICS-2000). Furthermore, longitudinal subsections of the core (32 mm x 32 mm in size) were melted continuously on a chemically inert, ultra-clean melt head to measure electrical conductivity, dust and $H_2O_2$ at high resolution

(~1 mm; Continuous Flow Analysis [CFA]; McConnell et al., 2002). In addition, core PASO-1 was analysed for liquid conductivity, $H_2O_2$, NO$_3^-$, NH$_4^+$, Black Carbon and various chemical elements at the Trace Chemistry Laboratory of the Desert Research Institute (DRI) in Reno, Nevada, USA, according to methods described by Röthlisberger et al. (2000), McConnell et al. (2002) and McConnell et al. (2007). Chemical elements were measured using two Thermo Finnigan Element2 ICP-MS instruments. Subsequently, values of ssNa (sea-salt Na) and nssS (non-sea-salt S), which are used for the core chronology of

PASO-1, were calculated from Na- and S-concentrations according to Röthlisberger et al. (2002) and Sigl et al. (2013), respectively, applying relative abundances from Bowen (1979; Eq. S1).

The corrected and calibrated $\delta^{18}O$ and $\delta D$ values were used to calculate the d excess (d excess = $\delta D$-8*$\delta^{18}O$; Dansgaard, 1964) and to derive a co-isotopic relationship ($\delta D = m* \delta^{18}O + n$) for each core. Since there are no stable water isotope data of recent precipitation available for the UG region, a Local Meteoric Water Line (LMWL) was inferred from the co-isotopic relationship

of all cores. Based on findings by Fernandoy et al. (2010) we believe that this composite co-isotopic relationship - at a first approximation - can be referred to as LMWL at UG.

## 2.2 Dating of firn cores and time series construction

Recent studies have used the seasonality of stable water isotopes and chemical parameters to reliably date firn cores from

Antarctica (e.g. Vega et al., 2016; Caiazzo et al., 2016). Stable water isotope profiles ($\delta^{18}O$ and $\delta D$) of the six firn cores are displayed with respect to depth in Figure S2. Data gaps in GUPA-1, DOTT-1 and SCH-1 are due to leaking sample bags (GUPA-1: 2 samples; DOTT-1: 3 samples; SCH-1: 2 samples). Firn cores for which glacio-chemical data are available, i.e. SCH-2 and PASO-1, were dated using annual layer counting (ALC) of stable water isotopes and chemical parameters focusing on $H_2O_2$ for SCH-2 (Fig. 2) and nssS for PASO-1 (Fig. 3). $H_2O_2$ and nssS exhibit both clear seasonal alternations between

highest and lowest values, since the former parameter is directly linked to the annual insolation cycle (Lee et al., 2000; Stewart et al., 2004) and the latter to marine biogenic activity (phytoplankton; Kaufmann et al., 2010). In addition, for the age model construction of SCH-2 methanesulfonic acid (MSA), SO$_4^{2-}$ and the Cl/Na-ratio were used for corroborating the years identified by ALC of $H_2O_2$ and stable water isotopes (Vega et al., 2018). For the dating of PASO-1 the signal of the Mt. Pinatubo eruption (1991) could be found via the coincidence of an above-average peak in the nssS/ssNa-record with a lacking winter minimum

in the nssS-record at 9.5-9.7 m depth (Fig. 3). Similar changes in wintertime nssS were observed in the well-dated ice cores PIG2010, DIV2010 and THW2010 from West Antarctica (Pasteris et al., 2014). Hence, the signal attributed to the Mt. Pinatubo eruption was used as additional tie point for the age model construction of PASO-1. For cores GUPA-1, DOTT-1, SCH-1 and BAL-1 we applied ALC of stable water isotopes and then matched the preliminary age-depth relationships to the SCH-2 age scale.

Snow accumulation rates at the firn core sites were determined and converted to meters of water equivalent per year (m w.eq.a$^{-1}$) based on measured densities (Fig. S3).

Composite stable water isotope and accumulation records were constructed for the entire UG region by combining time series of annually averaged stable water isotopes and accumulation rates of the individual firn cores. In order to assign each firn core

the same weight and to not overrepresent a certain firn core drill site annually averaged data were standardized before stacking (Stenni et al., 2017; Eq. S4). Linear trends were calculated and tested for their statistical significance using the non-parametric Mann-Kendall Tau and Sen slope (s) estimator trend test (Mann, 1945; Kendall, 1975; Sen, 1968) with correction for autocorrelation according to Yue and Wang (2004). Signal-to-noise ratios for stable water isotope and accumulation rate time series of the UG firn cores were estimated according to Fisher et al. (1985; Eq. S5).

**2.3 Meteorological database and backward trajectory analysis**

Meteorological data from an automatic weather station (AWS) located at the UG ice runway (79°46'S, 83°16'W, 705 m asl; Fig. 1b) cover the 4-year period from 1st February 2010 to 8th February 2014. The AWS (station: Wx7) records near-surface air temperature (2 m), wind speed, wind direction, relative humidity and air pressure every ten minutes. Additionally, hourly-resolved data of the same meteorological parameters are available from a second AWS (station: Arigony) operated on Union Glacier (79°46'S, 82°54'W, 693 m asl; Fig. 1b) covering the period from 14th December 2013 to 29th March 2018. Since differences between the Wx7 and the Arigony records are small throughout the overlapping period (14th December 2013 to 8th February 2014; e.g. mean difference in air temperature/pressure: 0.74°C/5.76 hPa) the two datasets were combined in order to expand the meteorological record.

The meteorological data from the two AWS (air temperature and air pressure) along with measured surface densities (Fig. S3) and derived accumulation rates (Fig. 6 and Table 1) were used to model depth-dependent diffusion lengths for each firn core across the entire length following the approach described by Münch and Laepple (2018) and Laepple et al. (2018), respectively. Local near-surface air temperatures at the firn-core drill sites were estimated from the AWS measurements by rescaling using the altitude of the respective site and a lapse rate of 1°C/100 m. Local surface air pressures were estimated from the barometric height formula.

Furthermore, we use fields of near-surface air temperature (2 m), precipitation-evaporation and geopotential heights (850 mbar) from the European Centre for Medium-Range Weather Forecasts (ECMWF) Interim Reanalysis (ERA-Interim; 1979-2019, spatial resolution: 79 km, temporal resolution: 6 hours; Dee et al., 2011; available at: https://www.ecmwf.int/en/forecasts/datasets/reanalysis-datasets/era-interim) for comparison with UG composite records of stable water isotopes and accumulation rates. ERA-Interim reanalysis data were aggregated to daily, monthly or annual values. Annually-averaged stable water isotope compositions and accumulation rates were also related to time series of climate modes such as SAM and ENSO as well as to SIE and SIC in order to identify potential dominant drivers of climate variability in the UG region.

For comparison with the UG stable water isotope and accumulation records mean monthly SIE for different Antarctic sectors (Weddell Sea: 60°W-20°E; Indian Ocean: 20°E-90°E; Western Pacific: 90°E-160°E; Ross Sea: 160°E-130°W; Bellingshausen-Amundsen Sea: 130°W-60°W) was obtained from the National Aeronautics and Space Administration (NASA; Cavalieri et al., 1999, 2012; available at: https://neptune.gsfc.nasa.gov/csb/index.php). Note that these data are only available for the period 1979-2012. Satellite-derived SIC data were acquired from the National Snow and Ice Data Center (NSIDC). The data set NSIDC-0079 - Bootstrap SIC from Nimbus-7 SMMR and DMSP SSM/I-SSMIS, Version 3 - with a spatial resolution of 25 km x 25 km was used (available at: https://nsidc.org/data/nsidc-0079; Comiso, 2017). Furthermore, we use the Marshall SAM Index (Marshall, 2003) as indicator for the prevailing SAM phase (available at: https://legacy.bas.ac.uk/met/gjma/sam.html). The Multivariate ENSO Index (MEI; Wolter and Timlin, 1993, 1998) serves as indicator for the occurrence and strength of El Niño and La Niña events (available at: https://www.esrl.noaa.gov/psd/enso/mei/table.html).

Spatial and cross-correlation analyses between UG composite time series of stable water isotopes and accumulation rates and time series of ERA-Interim reanalysis data, SIE, SIC, the Marshall SAM Index and the MEI, respectively, were carried out

calculating Pearson correlation coefficients (r) and p-values (p) at the 95% confidence level (i.e. α = 0.05). For calculating spatial correlations all time series were detrended.

We performed backward trajectory analysis using the Hybrid Single Particle Lagrangian Integrated Trajectory (HYSPLIT) model (Draxler and Hess, 1998; Stein et al., 2015; available at: https://ready.arl.noaa.gov/index.php) in order to determine potential moisture source regions and transport pathways of precipitating air masses for the UG region. The drill site of firn core SCH-1 was taken as initial point for the calculation of 5-day backward trajectories. This location was selected as it is less influenced by post-depositional processes as well as wind-induced redistribution and removal of snow. Thus, it is among all available firn-core drill sites the most representative for snow fall events in the UG region. ERA-Interim time series of daily precipitation extracted from the nearest grid point at 1061 m asl (Table 1) for the period 2010-2015 (resolution: 0.75°x0.75°; available at: http://apps.ecmwf.int/datasets/data/interim-full-daily/levtype=sfc/) served as input to identify precipitation events on a daily scale. We used a minimum threshold corresponding to 1% of the mean annual snow accumulation at UG (see below), a value that in terms of percentage is equivalent to the one applied by Thomas & Bracegirdle (2009). Based on input data from the Global Data Assimilation System (GDAS-1; available at: https://www.ready.noaa.gov/archives.php) 121 backward trajectories were calculated corresponding to daily precipitation events occurring in the period 2010-2015. They were then subdivided into their respective seasons (DJF, MAM, JJA and SON). Individual backward trajectories were combined to four different clusters that group precipitation events with statistically similar transport pathways according to their spatial variance (Stein et al., 2015).

## 3. Results

### 3.1 Meteorological data

The mean daily air temperature for the composite record of near-surface air temperature (2010-2018; Fig. 4) is -21.3°C ± 8.5°C with an absolute minimum of -46.8°C recorded on 17 July 2017 and an absolute maximum of +3.3°C measured on 21 February 2013. Daily air temperatures are the highest during December and January with mean values of -9.2°C ± 2.9°C and -9.1°C ± 3.0°C, respectively, and lowest from April to September with mean values below -25°C. The coldest month of the whole composite record period is July with a mean air temperature of -28.8°C ± 5.6°C. Daily wind speeds have a mean value of 6.9 ms$^{-1}$ ± 5.1 ms$^{-1}$ with a predominant direction from SW (220°). The maximum wind speed recorded is 29.7 ms$^{-1}$. Wind speeds are higher than 1.2 ms$^{-1}$ in > 75% of all data. The observations are in line with those made by Rivera et al. (2014) for the period 2008-2012 and imply the possibility of substantial redistribution of snow due to wind drift.

### 3.2 Firn core age model

The results of firn core dating are given in Table 1. The estimated dating uncertainty is ±1 year for cores dated with stable water isotopes and glacio-chemistry (SCH-2 and PASO-1) and ± 2 years for cores dated with stable water isotopes only and subsequent matching to the SCH-2 age scale (GUPA-1, DOTT-1, SCH-1 and BAL-1). Core DOTT-1 (16 years, 1999-2014) exhibits the shortest and PASO-1 (43 years, 1973-2015) the longest record. Note that for the age-model construction of GUPA-1 and BAL-1 two and three years, respectively (1990 and 2001for GUPA-1; 1981, 1983 and 1994 for BAL-1), were only identified from linear interpolation and are not clearly visible in the stable water isotope records due to smoothing in the respective core sections. Furthermore, for SCH-1 the first year (1986) was identified by linear extrapolation at the lowest end of the core using the depth-age-relationship between the previous two clearly identifiable peaks (years 1987 and 1988). For all cores the last year (either 2014 or 2015) was excluded from further analyses as it is incomplete.

### 3.3 Firn core stable water isotopes and accumulation rates

The mean stable oxygen isotope composition of the six firn cores ranges from -36.6 ‰ (PASO-1) to -29.9 ‰ (DOTT-1) with standard deviations varying between 2.3 ‰ (PASO-1) and 3.9 ‰ (DOTT-1; Table 1). Absolute minimum δ$^{18}$O (δD) values

are found in GUPA-1, absolute maximum $\delta^{18}O$ ($\delta D$) values in DOTT-1. Note that the results for GUPA-1 have to be handled with caution as this core was drilled next to the UG field camp and ice-landing strip. Therefore, snow relocation effects due to wind drift and/or human activities (e.g. runway maintenance) might have biased its stable water isotope composition. This is also indicated by less pronounced seasonal alternations in the GUPA-1 stable water isotope records (Fig. S2). Despite different

drill locations and altitudes, the range in mean d excess values of the six firn cores is small (from 4.9 ‰ [SCH-2] to 7.0 ‰ [PASO-1]). The slope of the co-isotopic relationship (Table 1 and Fig. S6a-f) is close to that of the Global Meteoric Water Line (GMWL; Craig, 1961) for all cores ranging between 7.94 (GUPA-1) and 8.24 (PASO-1). Hence, the original (oceanic) stable water isotope signal is preserved during moisture transport and snow deposition at the study site (Clark and Fritz, 1997). For all cores $\delta^{18}O$ and $\delta D$ values are highly correlated ($R^2 \geq 0.98$) with the largest variation in d excess observed for SCH-2

(values range from -5.6 ‰ to 17.2 ‰). The LMWL of the UG region was determined as $\delta D = 8.02*\delta^{18}O + 6.57$ ($R^2 = 0.99$, p = 0; Fig. 5).

Time-series analysis of $\delta^{18}O$ annual means (Fig. S7) reveals statistically significant trends only for cores SCH-1 (s = +0.039 ‰ a$^{-1}$, p < 0.05) and BAL-1 (s = +0.054 ‰ a$^{-1}$, p < 0.0001). Statistically significant positive trends of d excess annual means (Fig. S8) have been found for GUPA-1 (s = +0.085 ‰ a$^{-1}$, p < 0.01), SCH-2 (s = +0.085 ‰ a$^{-1}$, p < 0.0001) and PASO-1 (s =

+0.016 ‰ a$^{-1}$, p < 0.01), whereas for DOTT-1 (s = -0.110 ‰ a$^{-1}$, p < 0.05) and SCH-1 (s = -0.052 ‰ a$^{-1}$, p < 0.0001) the d excess trend is negative. The BAL-1 d excess record exhibits no trend.

Mean annual accumulation rates vary between ~0.18 m w.eq.a$^{-1}$ (GUPA-1 and PASO-1) and ~0.29 m w.eq.a$^{-1}$ (SCH-2) with the lowest standard deviations found at the DOTT-1 and PASO-1 sites (~0.04-0.05 m w.eq.a$^{-1}$) and the highest ones exhibited by cores SCH-2 and BAL-1 (~0.08 m w.eq.a$^{-1}$; Table 1). Annual minimum accumulation ranges between 0.1 and 0.2 m w.eq.a$^{-1}$

. Annual maximum accumulation reaches - except at the PASO-1 site - values of higher than 0.3 m w.eq.a$^{-1}$ with the absolute maximum found at the SCH-2 site in 1985 (0.47 m w.eq.a$^{-1}$). In the same year, snow accumulation reaches a local minimum of 0.098 m w.eq.a$^{-1}$ at the PASO-1 site (Fig. 6). At the GUPA-1 and BAL-1 sites snow accumulation decreased at the same rate of -0.003 m w.eq.a$^{-1}$ (p < 0.01). At the SCH-1 and SCH-2 sites the decrease in snow accumulation amounts to -0.002 m w.eq.a$^{-1}$ (p < 0.05) and -0.004 m w.eq.a$^{-1}$ (p < 0.0001), respectively. In contrast, snow accumulation exhibits a slight, albeit

statistically significant increase at the PASO-1 site (s = +0.001 m w.eq.a$^{-1}$, p < 0.0001).

## 4. Discussion

### 4.1 Potential noises influencing UG firn core records

In order to properly assess the environmental signals in stable water isotope and accumulation data of the UG firn cores, several

aspects that could potentially induce noise need be taken into account: the intermittency of precipitation, the redistribution of snow by wind drift as well as the diffusion in firn.

The analysis of diffusion lengths for the maximum depth of the UG firn cores revealed values between 0.07 m (PASO-1) and 0.11 m (GUPA-1), which are much lower than the mean annual layer thicknesses ranging between 0.35 m (PASO-1) and 0.60 m (DOTT-1; Table S11). Therefore, we assume diffusion to be of minor importance for inducing noise to the stable water

isotope records of the UG firn cores.

In contrast, wind drift and redistribution of snow certainly play an important role in the UG region, which is supported by data on wind speed (on average 7 ms$^{-1}$) and wind direction (predominantly from SW) from the two AWS. The location of the individual firn cores at different altitudes as well as in different topographic positions (near the ice-landing strip, valley vs. plateau) causes site-specific differences in the susceptibility towards wind drift and thus to the reallocation of snow. However,

as there are AWS data from only one location, i.e. the GUPA-1 drill site, for the period February 2010 - March 2018 available, we have no reliable meteorological information for the other firn core drill sites in order to assess the local effects of wind drift in detail. Rivera et al. (2014) report that in the UG region katabatic winds can cause strong snow drift due to acceleration by the slope of the glacial valleys feeding UG. Studies of drifting snow in Antarctica using satellite remote sensing (Palm et al., 2011)

and regional climate modelling (Lenaerts et al., 2012; Lenaerts and van den Broeke, 2012) provide evidence for a drifting snow frequency (defined as the fraction of days with drifting snow) of about 15-30% in the UG region. Also, studies from close-by areas such as Patriot Hills (Casassa et al., 1998) and the Ronne-Filchner Ice Shelf (Graf et al., 1988) as well as from DML (Schlosser and Oerter, 2002; Kaczmarska et al., 2004) revealed that wind drift and random sastrugi formation are the main

reasons for large spatial and temporal variations in accumulation rates. However, linkages between the different firn core drill sites at UG are speculative, e.g. the potential uptake of snow at the PASO-1 drill site (local minimum accumulation) and its redistribution towards the lower-altitudinal core sites (SCH-2; absolute maximum accumulation) by the predominantly south-westerly winds in 1985. The calculation of signal-to-noise ratios for stable water isotopes and accumulation rates can help to further assess the extent to which the UG firn cores are influenced by noise-inducing processes such as wind drift and diffusion.

In the following, firn core GUPA-1 is excluded from statistical evaluation due to the likely biasing and smoothing of its stable water isotope and accumulation records as a consequence of its position near the UG ice-landing strip. Based on the two-by-two signal-to-noise ratios between the individual cores (Tables S9 and S10), we obtained for the UG firn cores a $\delta^{18}$O signal-to noise ratio of 0.60 for the entire record period (1973-2014), and 0.78 when referring to the overlapping period (1999-2013). The signal-to-noise ratios are similar when considering only the three neighbouring cores BAL-1, SCH-1 and SCH-2 situated

within 10 km distance in similar valley positions. They yield 0.60 for 1973-2014 and 0.86 for 1999-2013. The signal-to-noise ratios of $\delta^{18}$O in the UG region of between 0.6 and 0.86 are quite high, i.e. they are similar or slightly higher than signal-to-noise ratios of $\delta^{18}$O on the WAIS for interannual timescales (~0.5-0.7), and much higher than those in DML (< 0.2) for multiannual to decadal timescales (Münch and Laepple, 2018).

Calculation of the signal-to-noise ratio for accumulation rate time series of the UG firn cores revealed a value of 0.27 for the

entire record period and 0.33 for the overlapping period, respectively. Compared to the signal-to-noise ratios of $\delta^{18}$O these values are much lower, probably reflecting the strong influence of the site-specific characteristics on accumulation rates (e.g. the different exposure to wind drift and the subsequent relocation of snow). The signal-to-noise ratio increases significantly when referring to the close-by cores BAL-1, SCH-1 and SCH-2 or to SCH-1 and SCH-2 only (Tables S9 and S10). It yields 0.79 and 1.67, respectively, when referring to the entire record period. This might be due to the proximity of the drill sites

(within 10 km distance) and the similarity of the site-specific characteristics of the three cores, i.e. the location in northwest-southeast oriented, U-shaped glacial valleys, leading to similar accumulation patterns. When referring to the overlapping period only, the signal-to-noise ratio stays at a low value of 0.24 for the three cores, but remains high for SCH-1 and SCH-2 (1.2; Tables S9 and S10). In summary, the signal-to-noise ratios of accumulation rates at UG are consistent with those determined by Schlosser et al. (2014) and Altnau et al. (2015) for firn cores from Fimbul Ice Shelf, DML, and much higher

than those found in low-accumulation areas of the Amundsenisen mountain range, DML (Graf et al., 2002; Altnau et al., 2015). Generally, the higher the accumulation rates at a specific site are, the higher are the signal-to-noise ratios for stable water isotopes and accumulation rates (Hoshina et al., 2014; Münch et al., 2016). Hence, the UG region with higher accumulation rates as compared to low-accumulation sites on the EAIS (e.g. Oerter et al., 2000) exhibits relatively high signal-to-noise ratios for both stable water isotopes and accumulation rates, despite the likely influence by post-depositional processes, in particular

the redistribution of snow due to wind drift. Hence, we conclude that the UG firn core records allow to deduce temporal changes in stable water isotopes and accumulation rates on a regional scale.

## 4.2 Spatial and temporal variability of firn core stable water isotope composition and relation to near-surface air temperatures, sea ice and climate modes

### 4.2.1 Spatial and temporal variability of stable water isotopes

From the inter-comparison of mean, minimum and maximum values of $\delta^{18}$O annual means for the overlapping period (1999-2013, excluding GUPA-1; Table 1 and Fig. S7), generally a depletion of the stable water isotope composition with increasing height ("altitudinal effect") and distance from the sea ("continentality effect") has been detected as expected for a Rayleigh

distillation process (Clark and Fritz, 1997). Core PASO-1 situated at the highest altitude and greatest distance from the sea shows the lowest mean $\delta^{18}O$ values (-36.3 ‰). In contrast, the low-altitude core DOTT-1, which is situated closest to the sea, displays a distinctly higher mean stable water isotope composition (-29.7 ‰). However, core SCH-2 exhibits less depleted mean $\delta^{18}O$ values (-34.1 ‰) than SCH-1 (-34.7 ‰), although located at about 250 m higher altitude. This inverted pattern,

which is also visible in the mean, minimum and maximum $\delta^{18}O$ ($\delta D$) values derived from all samples of the respective core (Table 1), might originate from snow reallocation processes within the glacial valley due to katabatic winds. Nonetheless, the existence of an "altitudinal effect" is confirmed by the $\delta^{18}O$-altitude-relationship calculated from $\delta^{18}O$ annual means for the overlapping period (excluding GUPA-1) yielding: altitude = -152.7*$\delta^{18}O$ - 3803.4 with $R^2$ = 0.85 and p < 0.05 (Fig. S12).

The individual firn cores are generally well correlated with each other for $\delta^{18}O$ and $\delta D$, for both the maximum overlapping

period between two individual cores and the overlapping period (1999-2013; Tables S9 and S10). All cores display a common $\delta^{18}O$ maximum in summer 2002 (Fig. 7). Correlation coefficients are highest for nearby cores such as SCH-1 and SCH-2 reaching up to r = 0.66 (1999-2013: r = 0.58) for $\delta^{18}O$ (p < 0.05) and r = 0.71 (1999-2013: r = 0.66) for $\delta D$ (p < 0.01). Only core BAL-1 shows relatively low correlations with all other cores except for SCH-1 (r = 0.44 for $\delta^{18}O$ and r = 0.43 for $\delta D$, p < 0.05; 1999-2013: r = 0.49 for $\delta^{18}O$ and r = 0.50 for $\delta D$, p < 0.1; Tables S9 and S10). The results of the cross-correlation

analysis (Tables S9 and S10), the negligible influence of diffusion and the relatively high signal-to-noise ratios allowed to construct a composite $\delta^{18}O$-record (UG $\delta^{18}O$ stack) based on standardized annually averaged data spanning the period that comprises at least three core records per year (1980-2014, excluding GUPA-1; Fig. 8). From the UG $\delta^{18}O$ stack a more regional picture of the isotopic characteristics of precipitation at UG can be drawn. As a result we found only a negligible positive trend in $\delta^{18}O$, which is statistically not significant. Thus, we infer that at UG regional changes in $\delta^{18}O$ values must have been

negligible during the last 35 years. This is in line with findings from ice cores retrieved from the nearby Ronne-Filchner Ice Shelf and the Weddell Sea sector, respectively, that show no statistically significant trends in their stable water isotope time series (e.g. Foundation Ice Stream [Graf et al., 1999], Berkner Island [Mulvaney et al., 2002; Stenni et al., 2017]). However, it is worth mentioning, that stacking firn core $\delta^{18}O$ time series without previous standardization (excluding GUPA-1) yields a statistically significant positive trend of +0.054 ‰ $a^{-1}$ (p < 0.0001; Fig. 8). This trend most likely results from the dominance

of SCH-1 and BAL-1 as these are the only cores showing statistically significant (positive) trends in their $\delta^{18}O$ records (Fig. S7).

In order to detect possible changes in the origin of precipitating air masses reaching the UG region, d excess time series of individual firn cores have been inter-compared (Fig. S8). In line with $\delta^{18}O$, the d excess does not show clear trends or similar contemporaneous changes among the firn cores suggesting little change in the main moisture sources and the origin of air

masses precipitating over the UG region at least during the last four decades. The composite d excess record (UG d excess stack; excluding GUPA-1), which has been calculated despite the individual firn cores showing no cross-correlations in the d excess (Tables S9 and S10), corroborates this finding. It shows no statistically significant trend for the period 1980-2014 (Fig. 8). Dissimilarities between d excess trends of individual firn cores might be partly explained by the presence of orographic barriers that separate the different drill sites from each other. This might lead to dissimilarities in moisture sources, even though

the drill sites are located within only 50 km horizontal distance.

### 4.2.2 Relation of stable water isotopes to near-surface air temperature records

Linear regression between stacked seasonal means of non-standardized firn core $\delta^{18}O$ and seasonal means of AWS-derived near-surface air temperature for the overlapping period February 2010 - November 2015 (Fig. S13) revealed a statistically

significant positive $\delta^{18}O$-T relationship ($\delta^{18}O$ = 0.175*T-31.6, $R^2$ = 0.21, p < 0.05). In order to test the $\delta^{18}O$-T relationship for the entire period covered by the UG $\delta^{18}O$ stack (1980-2014), monthly means of ERA-Interim near-surface air temperature were correlated with those derived from the AWS records for the period February 2010 - November 2015. We used ERA-Interim near-surface air temperatures extracted for the grid point which is nearest to the GUPA-1 drill site (Table 1) as this is

the firn core site closest to the two AWS. Monthly mean air temperatures from both datasets are highly and statistically significantly correlated ($R^2 = 0.99$, $p < 0.0001$). This allows to calculate a $\delta^{18}$O-T relationship for the period 1980-2014 based on seasonal means of ERA-Interim near-surface air temperatures and stacked seasonal means of non-standardized UG $\delta^{18}$O. The latter yields $\delta^{18}$O = 0.128*T-31.7 ($R^2 = 0.22$, $p < 0.0001$) that is very similar to the $\delta^{18}$O-T relationship calculated based on AWS data for the period 2010-2015 (Fig. S13). Thus, independent of the period considered we conclude that increasing air temperatures are generally linked with increasing $\delta^{18}$O values. However, as this relationship is only weak, a proper inference of near-surface air temperatures from $\delta^{18}$O values of precipitation is not yet possible for the UG region. This is due to (1) the shortness of the available near-surface air temperature record and (2) the arbitrary calculation of $\delta^{18}$O seasonal means assuming that precipitation at the study site is evenly distributed throughout the year.

In order to further test the $\delta^{18}$O-T relationship, the UG $\delta^{18}$O stack was spatially correlated with ERA-Interim near-surface air temperatures for the period 1980-2014 (Fig. 9a). No correlation was found with ERA-Interim based near-surface air temperatures at the UG site, but with those further to the east (Coats Land and DML; up to r > 0.4, p < 0.05; Fig. 1a and 9a). This might be due to the ERA-Interim model not properly capturing the local small-scale orography of the Ellsworth Mountains. Hence, the ERA-Interim model does not truly reflect the local climate at the UG site, but rather the regional climate along the Weddell Sea coast. The large differences between the actual altitudes of the firn core drill sites and the elevations of the respective nearest ERA-Interim grid points (Table 1), ranging from about 100 m (DOTT-1) up to 700 m (PASO-1), might support this hypothesis. A second possible explanation for the observed correlations might be, that both Coats Land and DML constitute important moisture source areas from which precipitating air masses approach the UG region. This aspect is further discussed below.

The observation of a period of rather constant stable water isotope time series at UG contrasts with the significant warming on the WAIS (Bromwich et al., 2013), but is in line with the absence of a clear regional air temperature change on the AP (since the late 1990s; Turner et al., 2016) and the EAIS (Turner et al., 2005a; Schneider et al., 2006; Nicolas and Bromwich, 2014; Smith and Polvani, 2017). For instance, a slightly negative, but statistically not significant trend in near-surface air temperatures has been observed in the instrumental record of Halley research station (1957-2000: -0.11±0.47°C; Turner et al., 2005a). No trend has been found at Neumayer research station (1981-2010; Schlosser et al., 2014). However, Medley et al. (2018) have recently shown, that near-surface air temperatures in western DML have significantly increased between 1998 and 2016 (+1.15±0.71°C per decade). In summary, we assume that mean $\delta^{18}$O values in the UG region capture regional air temperature variations only to some extent during a period of rather constant climate conditions.

### 4.2.3 Relation of stable water isotopes to large-scale climate modes and sea ice variability

Mean annual d excess values (UG d excess stack) exhibit a weak positive correlation with the SAM Index (r = 0.38, p < 0.05; Table 2). Stronger contraction of the polar vortex during positive SAM phases facilitates the advection of warm and moist air from mid- and lower latitudes, i.e. from regions with higher SST and lower relative humidity towards Antarctica (Thompson and Wallace, 2000; Thompson and Solomon, 2002; Gillett et al., 2006). Hence atmospheric water vapour with higher d excess values is expected to reach Antarctica (Uemura et al., 2008; Stenni et al., 2010). This relationship is confirmed by spatial correlations of mean annual d excess values with ERA-Interim geopotential heights (850 mbar) calculated for the period 1980-2014 (up to r > 0.4, p < 0.05; Fig. 9b). Increased geopotential heights above the mid-latitudes, as occurring during positive SAM phases (Thompson and Wallace, 2000; Thompson and Solomon, 2002), imply increased d excess values of precipitation in the UG region. However, mean annual $\delta^{18}$O ($\delta$D) values (UG $\delta^{18}$O [$\delta$D] stack) show no correlation with the SAM Index (Table 2). Furthermore, none of the isotopic values (i.e. $\delta^{18}$O, $\delta$D, d excess) exhibits a significant correlation with the MEI Index (Table 2). Hence, oceanic circulation changes associated with the alternation between El Niño and La Niña events seem to have no detectable influence on the stable water isotope composition of precipitation in the UG region.

Kohyama and Hartmann (2016) showed that the SAM Index is statistically significantly positively correlated with sea ice extent (SIE) in the Indian Ocean sector of Antarctica. When comparing SIE in the different Antarctic sectors (Weddell Sea, Bellingshausen-Amundsen Sea, Ross Sea, West Pacific, Indian Ocean) with the UG $\delta^{18}$O and d excess stacks, the only (weak) positive correlation is found between UG d excess and SIE in the Indian Ocean sector (r = 0.32, p < 0.1; Table 2). However, this correlation does not necessarily indicate predominant moisture transport from the Indian Ocean sector towards UG. Cluster analysis and seasonal frequency distribution of backward trajectories performed with the HYSPLIT model (Fig. 10 and 11) suggests, that the Weddell Sea sector, including the Ronne-Filchner Ice Shelf, is the dominant source region for precipitating air masses reaching the UG site (46%) throughout the year, closely followed by Coats Land and DML (35%). The latter finding is consistent with the observation of a positive correlation between the UG $\delta^{18}$O stack and ERA-Interim near-surface air temperatures in these regions (Fig. 9a; see above). Nevertheless, during winter (JJA) and spring (SON) a small percentage of precipitating air masses reaching UG might also originate from the Indian Ocean sector (Fig. 11) supporting the findings from cross-correlation analysis (Table 2).

Spatial correlations with SIC yield a more specific regional picture of the interplay between sea ice distribution and UG moisture sources. We found that only SIC in the northern Weddell Sea exhibits a strong negative correlation with mean annual $\delta^{18}$O and d excess values in the UG region (up to r < -0.6, p < 0.05; 1980-2014; Fig. 12a and b) corroborating the results of the backward trajectory analysis. Thus, higher or lower $\delta^{18}$O and d excess values at the UG site correspond to a reduction or increase in SIC in the northern Weddell Sea, respectively. Consequently, reduced SIC in the northern Weddell Sea implies enhanced availability of proximal surface level moisture (Tsukernik and Lynch, 2013) which would support higher $\delta^{18}$O ($\delta$D) values in the UG region during low SIC phases. For a detailed interpretation of the negative correlation between SIC and UG d excess further data on the moisture sources' relative humidity are needed which are not yet available.

## 4.3 Spatial and temporal variability of accumulation rates and relation to sea ice and climate modes
### 4.3.1 Spatial and temporal variability of accumulation rates

For the overlapping period (1999-2013; excluding GUPA-1; Table 1), the highest accumulation rates are observed at the DOTT-1 site, which is located at the lowest elevation and closest to the sea. However, accumulation rates are very similar at the drill sites of SCH-1, SCH-2, BAL-1 and PASO-1, despite the clear differences in altitude and distance from the sea (Fig. 1b and Table 1). Accumulation rates have decreased at all sites throughout the respective record period, except at the PASO-1 site (Fig. 6). The observed negative trends in accumulation rates are at the same order of magnitude as those reported by Burgener et al. (2013) for central West Antarctica for the period 1975-2010 (ranging between -0.0022 and -0.0072 m w.eq.a$^{-1}$) and by Schlosser et al. (2014) for Fimbul Ice Shelf, DML, for the period 1995-2009 (ranging between -0.006 and -0.021 m w.eq.a$^{-1}$), respectively. However, it seems that snow accumulation in the UG region is not directly related to altitude and distance from the sea. Furthermore, spatially varying accumulation trends likely reflect the strong influence of site-specific characteristics on accumulation rates, in particular the different exposure to wind drift. DOTT-1 and PASO-1 - the former located on an ice rise and the latter located on a high-altitude plateau - might be more exposed to wind drift than the sites of SCH-1, SCH-2 and BAL-1. The latter three are all located within U-shaped glacial valleys stretching from northwest to southeast (Fig. 1b), and, thus, are potentially better protected from the predominant south-westerly winds. Hence, accumulation trends might be better preserved in these records.

Based on the annual accumulation rates of all firn cores (excluding GUPA-1) for the period 1980-2014, an average accumulation rate in the UG region of 0.245 ± 0.07 m w.eq.a$^{-1}$ (n = 148) has been calculated. This value is roughly consistent with accumulation rates determined from stake measurements on UG (79°42'32" - 80°12'40"S, 80°56'13" - 82°55'52"W, 460-814 m asl, 2008-2009: up to 0.2 m w.eq.a$^{-1}$; Rivera et al., 2014), regional atmospheric model outputs (1980-2004: 0.086 ±2 - 0.328 ± 8 m w.eq.a$^{-1}$, horizontal model resolution: 55 km; van den Broeke et al., 2006) and the closest ITASE (International Trans-Antarctic Scientific Expedition) ice core 01-5 (77°03'32.4"S, 89°08'13,2"W, 1246 m asl, 1958-2000: 0.342 m w.eq.a$^{-1}$;

Genthon et al., 2005). It is also in line with accumulation rates reconstructed from firn cores retrieved along a 920 km long, northeast-southwest traverse on the Ronne-Filchner Ice Shelf (from 51°32'W, 77°59.5'S to 59°38.1'W, 84°49.1'S; 65-1191 m asl), that range between 0.09 and 0.182 m w.eq.a$^{-1}$ (1946-1994; Graf et al., 1999).

Analogous to the UG $\delta^{18}$O stack, a composite accumulation time series has been constructed from standardized annually averaged data and analysed for the period that comprises at least three core records (1980-2014; excluding GUPA-1; Fig. 8). The UG composite accumulation record shows a small statistically significant negative trend (Fig. 8; s = -0.020, p < 0.01). This finding is in line with Burgener et al. (2013), who observed a statistically significant negative trend in accumulation rates reconstructed from five firn cores from the central WAIS for a similar period as covered by the UG firn cores (1975-2010: on average 3.8 mm w.eq.a$^{-1}$). Furthermore, firn and ice-core based studies of the surface mass balance on Fimbul Ice Shelf, DML, revealed a negative trend for the second half of the 20$^{th}$ century (Kaczmarska et al., 2004; Schlosser et al., 2014; Altnau et al., 2015). Medley and Thomas (2019) also report a generally decreasing trend in snow accumulation for the EAIS since 1979. The negative accumulation trend in the UG region is in contrast to the absence of a long-term trend in accumulation rates in Adélie Land (Goursaud et al., 2017) as well as positive precipitation and accumulation trends observed on the AP (Turner et al., 2005b; Thomas et al., 2008; Frieler et al., 2015; Thomas et al., 2017), in coastal Ellsworth Land (WAIS; Thomas and Bracegirdle, 2015; Thomas et al., 2015), costal DML (Philippe et al., 2016) and in western DML (EAIS; Medley et al., 2018), respectively, during the 20$^{th}$ and early 21$^{st}$ centuries.

### 4.3.2 Relation of accumulation rates to sea ice variability and large-scale climate modes

The linkage between snow accumulation and SIE (SIC) in Antarctica is complex. Generally, the smaller the SIE (SIC) is, i.e. the closer the open water areas are, the more surface level moisture is available and may be transported towards the Antarctic continent causing an increase in snow accumulation (Tsukernik and Lynch, 2013; Thomas et al., 2015). Furthermore, higher air temperatures may lead to more open water areas, and warm air can transport more moisture than cold air (Clark and Fritz, 1997). However, these simplistic considerations might be much more complex in specific settings such as the UG region and might be superimposed by manifold factors and processes such as wind directions, wind drift, moisture content of air masses etc.

Backward trajectory analysis revealed that the Weddell Sea sector as well as Coats Land and DML are likely strong moisture source regions for UG in all seasons of the year (Fig. 10 and 11). This is confirmed by a strong positive correlation of the UG composite accumulation record with ERA-Interim precipitation-evaporation time series for these regions (up to r > 0.6, p < 0.05; 1980-2014), whereas there is no correlation with the UG site (Fig. 9c). This might be due to post-depositional processes (wind drift, snow removal and/or redeposition) influencing snow accumulation at the firn core sites, beside the above mentioned incapacities of the ERA-Interim model. Surprisingly, there appears to be only a weak negative relationship between SIC in the Weddell Sea sector and snow accumulation at UG (up to r < -0.3, p < 0.05; Fig. 12c). Cross-correlation analysis revealed no correlation with SIE in the Weddell Sea sector, but a weak negative correlation with the Indian Ocean sector (r = -0.30, p < 0.1; Table 2). The latter points towards increased precipitation and thus snow accumulation at UG during periods of low SIE in the Indian Ocean sector. However, the positive relationship between snow accumulation at UG and ERA-Interim precipitation-evaporation in the Indian Ocean sector is negligible compared to the Weddell Sea sector (Fig. 9c).

Furthermore, there is a weak positive correlation of snow accumulation at UG with SIC in the Bellingshausen Sea sector (up to r > 0.4, p < 0.05; Fig. 12c). Reduced sea ice in the Bellingshausen Sea sector and the increased availability of surface level moisture has been used to explain the increases in snow accumulation along the AP and in coastal Ellsworth Land during the 20$^{th}$ century (Thomas et al., 2015). However, the UG site is distant from the sea-ice edge and, thus, changes in sea ice conditions appear to be less important for snow accumulation in this region. There is no correlation between snow accumulation at UG with either SAM or ENSO (Table 2). This suggests that snow accumulation in the UG region is likely not directly driven by large-scale modes of climate variability and that UG seems to be located in a transition zone between West and East Antarctic climate.

## 5. Conclusions

We examined six firn cores from the Union Glacier region in the Ellsworth Mountains (79º46' S, 83º24' W) situated at the northern edge of the West Antarctic Ice Sheet. For all analysed firn cores time series of $\delta^{18}O$ ($\delta D$), d excess and accumulation rates were established covering periods between 16 and 43 years. A Local Meteoric Water Line was derived ($\delta D = 8.02 * \delta^{18}O + 6.57$) from the co-isotopic relationship of all firn cores confirming the preservation of the original (oceanic) stable water isotope signal during moisture transport towards and snow deposition at UG. Diffusion was found to have only a negligible influence on the stable water isotope records of the UG firn cores. Spatial and temporal variability of accumulation rates is likely to be rather influenced by post-depositional processes, i.e. wind drift, removal and redistribution of snow.

The analysis of signal-to-noise ratios for the UG firn cores revealed relatively high values for $\delta^{18}O$ of between 0.60 for the entire record period (1973-2014) and 0.78 for the overlapping period (1999-2013). For accumulation rates the signal-to-noise ratios for the two periods are lower amounting to 0.27 and 0.33, respectively, and hence, reflecting the strong influence of site-specific characteristics on snow accumulation (e.g. the different exposure to wind drift). The results of the signal-to-noise ratio analysis allowed to stack the single firn core stable water isotope and accumulation time series, respectively, for the period 1980-2014 in order to draw conclusions on a regional scale. For the UG composite $\delta^{18}O$ record (UG $\delta^{18}O$ stack) no statistically significant trend was found suggesting that regional changes in near-surface air temperature have been small at least since 1980. The absence of a $\delta^{18}O$-trend in the UG region is consistent with recent findings in the AP region and parts of the EAIS. Furthermore, mean annual $\delta^{18}O$ and d excess values in the UG region (UG $\delta^{18}O$ and d excess stacks) are likely related to sea ice conditions in the northern Weddell Sea. Also, there is a weak positive correlation between mean annual d excess values and SIE in the Indian Ocean sector. However, backward trajectory analysis suggests that the Weddell Sea sector, Coats Land and DML are the dominant source regions for precipitating air masses reaching the UG site. Mean annual $\delta^{18}O$ ($\delta D$) values are neither correlated with SAM nor with ENSO. Mean annual d excess values exhibit a weak positive correlation with the SAM implying that a positive SAM facilitates higher d excess values of precipitation in the UG region. However, no statistically significant trend has been found for the UG d excess stack suggesting overall little change in the main moisture sources and the origin of air masses precipitating over the UG region since 1980.

On average mean annual snow accumulation in the UG region amounts to 0.245 m w.eq.a$^{-1}$ in 1980-2014 and has slightly decreased during this period (Sen slope s = -0.020). This finding is in line with observations from the central and western WAIS and coastal parts of the EAIS, but contrasts positive precipitation and accumulation trends on the AP, the eastern WAIS and in inner parts of the EAIS. There is only a weak negative correlation between snow accumulation at UG and SIC in the close-by Weddell Sea sector. Furthermore, weak correlations with SIE in the Indian Ocean sector (negative relationship) and SIC in the Bellingshausen Sea sector (positive relationship) were found. However, sea ice conditions in the surrounding Antarctic sectors generally appear to play only a minor role for snow accumulation at UG. There is no direct relationship of mean annual snow accumulation at UG with large-scale modes of atmospheric variability (SAM or ENSO). We conclude that neither the rapid warming nor the large increases in snow accumulation, as observed on the AP and in coastal regions and the interior of West Antarctica during the last decades, have extended inland to the Ellsworth Mountains. Hence, the UG region, although being located at the northern edge of the WAIS and relatively close to the AP, exhibits rather East than West Antarctic climate characteristics.

*Author Contribution.*

F.F. designed the study and carried out the fieldwork supported by D.R.O. K.H. and H.M. performed the stable water analyses of firn cores BAL-1 and PASO-1. K.H. and J.R.M. performed the glacio-chemical analyses of PASO-1. Stable water isotopes of firn cores GUPA-1, DOTT-1 and SCH-1 were measured by F.F. and M.A. Stable water isotope and glacio-chemical analyses of SCH-2 were carried out by L.R.T. High-resolution density profiles of firn cores BAL-1 and PASO-1 were measured by

K.H. and J.F. J.A.-N. provided AWS data from Union Glacier. F.F. and D.T. modelled backward trajectories. L.R.T. helped with data standardization and spatial correlation analysis. K.H. was responsible for data analysis, interpretation, and writing of the manuscript supported by H.M., C.S., F.F., L.R.T. and T.O. All authors contributed to data interpretation and the preparation of the final manuscript.

*Competing Interests.*

The authors declare that they have no conflict of interest.

*Acknowledgements.*

10 The presented work was partially funded by the FONDECYT project 11121551. Kirstin Hoffmann was funded by an Elsa-Neumann PhD scholarship awarded by the state of Berlin, Germany. We thank the Chilean government, i.e. the Instituto Antártico Chileno (INACH) and the Fuerza Área de Chile (FACH) for their support in the organization of field campaigns and for providing logistical facilities. We highly acknowledge the help of the Centros de Estudios Científicos (CECs) and especially of Dr. Andrés Rivera, who contributed with geophysical data and field-site recommendations. We thank Antarctic

15 Logistics and Expeditions and in particular Marc de Keyser and Ronald Ross for providing us with meteorological data from their AWS (Wx7) on Union Glacier. We also thank Thomas Münch for calculating diffusion lengths as well as Andrew Dolman and Thomas Laepple for help with statistical analyses. Furthermore, we highly acknowledge the support of the involved laboratory personnel at AWI, BAS, DRI and UNAB. We also thank Sentia Goursaud, Massimo Frezzotti and one anonymous referee for their constructive comments.

*Data Availability.*

Stable water isotope compositions and accumulation rates of the six UG firn cores are available at: https://doi.pangaea.de/10.1594/PANGAEA.908205.

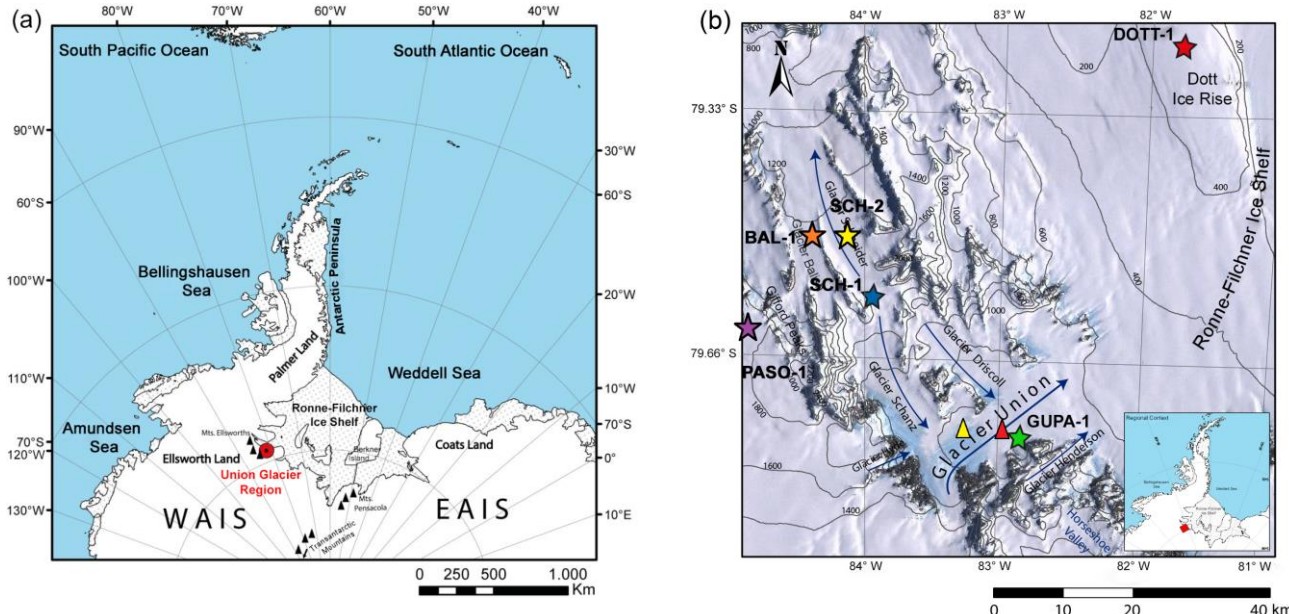

Figure 1a-b: Location of the UG region within Antarctica (a) and location of the drill sites of the six firn cores (GUPA-1, DOTT-1, SCH-1, SCH-2, BAL-1, PASO-1) within the UG region (b). The triangles in (b) denote the location of two automatic weather stations on UG (yellow: station Wx7; red: station Arigony; further explanations in the text). The background image in (b) was extracted from the Landsat Image Mosaic of Antarctica (LIMA) and the contour lines were obtained from the Radarsat Antarctic Mapping Project Digital Elevation Model, Version 2 (Liu et al., 2015).

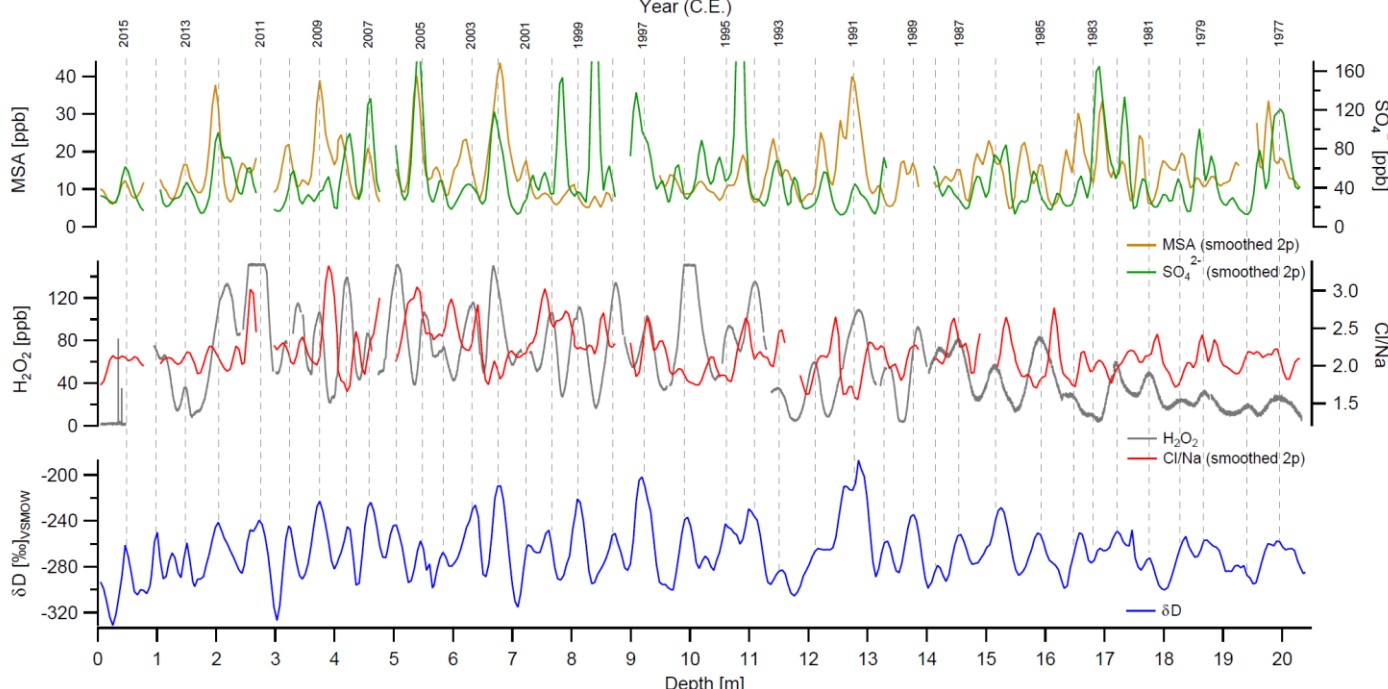

Figure 2: Age scale for firn core SCH-2 constructed by counting and inter-matching of maximum peaks (dashed lines) in CFA-derived profiles of stable water isotope composition ($\delta D$) and different chemical parameters ($H_2O_2$, Cl/Na, MSA and $SO_4^{2-}$). Smoothed 2P is a 2-point running average. Note, that the maxima of the chemical parameters do not necessarily coincide with the respective maxima in $\delta D$ due to the different seasonality of the proxies.

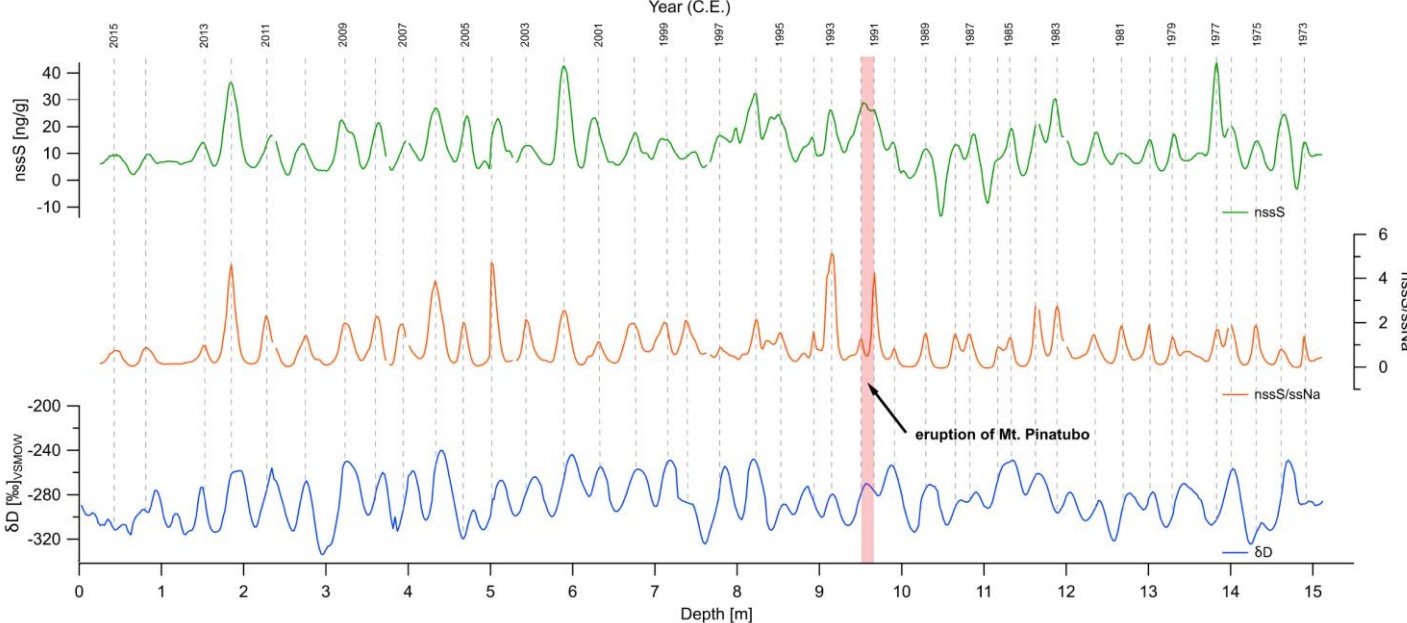

Figure 3: Age scale for firn core PASO-1 constructed by counting and inter-matching of maximum peaks (dashed lines) in profiles of stable water isotope composition ($\delta D$) as obtained from discrete sample measurements and in CFA-derived profiles of nssS and nssS/ssNa, respectively. The year of the eruption of Mt. Pinatubo (1991) that is used as tie point for annual layer counting is highlighted. Note, that the maxima of nssS and nssS/ssNa do not necessarily coincide with the respective maxima in $\delta D$ due to the different seasonality of the proxies.

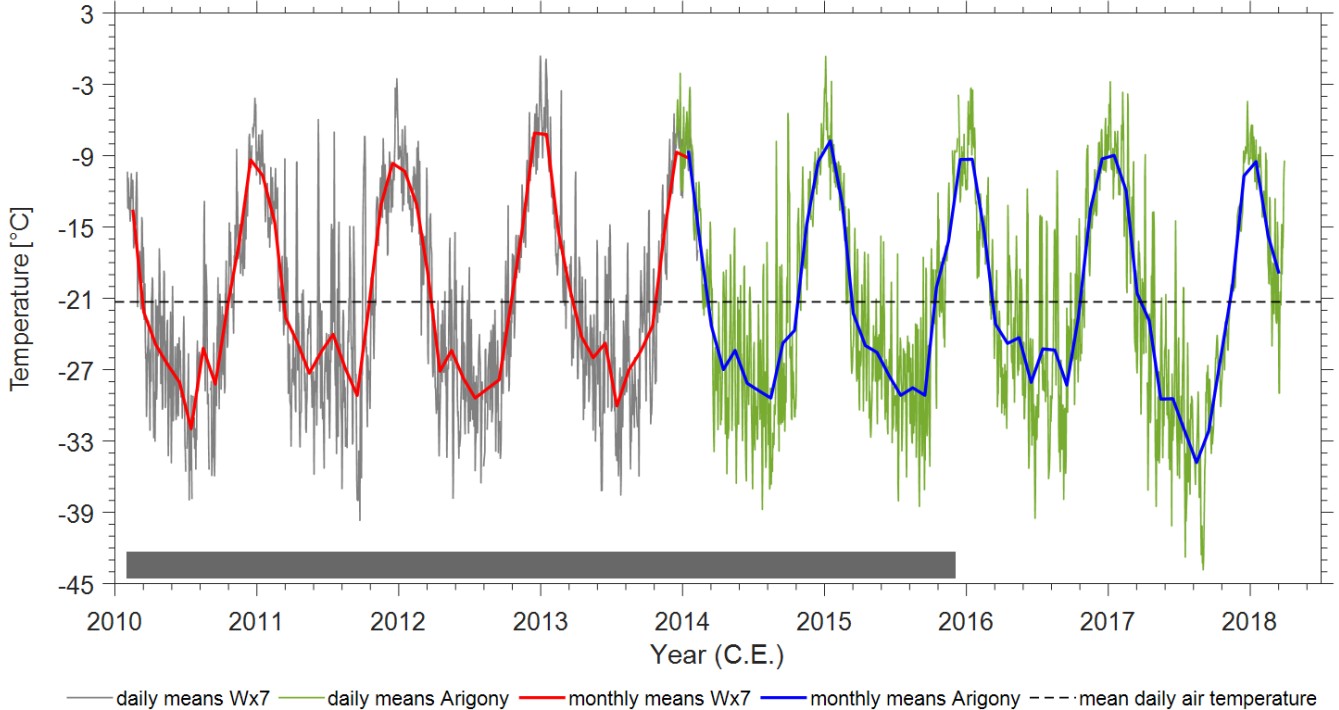

Figure 4: Composite record of mean daily and mean monthly air temperatures recorded at two nearby Union Glacier AWS sites (stations: Wx7, Arigony) from February 2010 to March 2018. The mean daily air temperature for the entire composite record period (-21.3°C; dashed black line) and the period overlapping with Union Glacier core records (February 2010 to November 2015; grey bar) are also indicated.

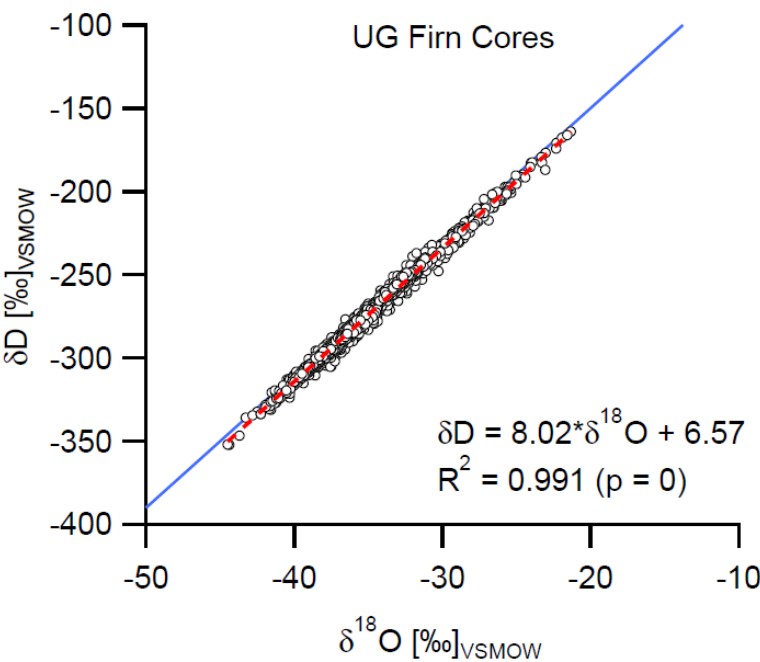

Figure 5: Composite co-isotopic relationship ($\delta^{18}O$ vs. $\delta D$) based on all individual samples (n = 2348; white dots) of the six firn cores from Union Glacier with the equation, the coefficient of determination ($R^2$) and the p-value (p) of the linear regression (red dashed line). The Global Meteoric Water Line (GMWL) is indicated in blue. The composite co-isotopic relationship is referred to as the Local Meteoric Water Line (LMWL) of the Union Glacier region.

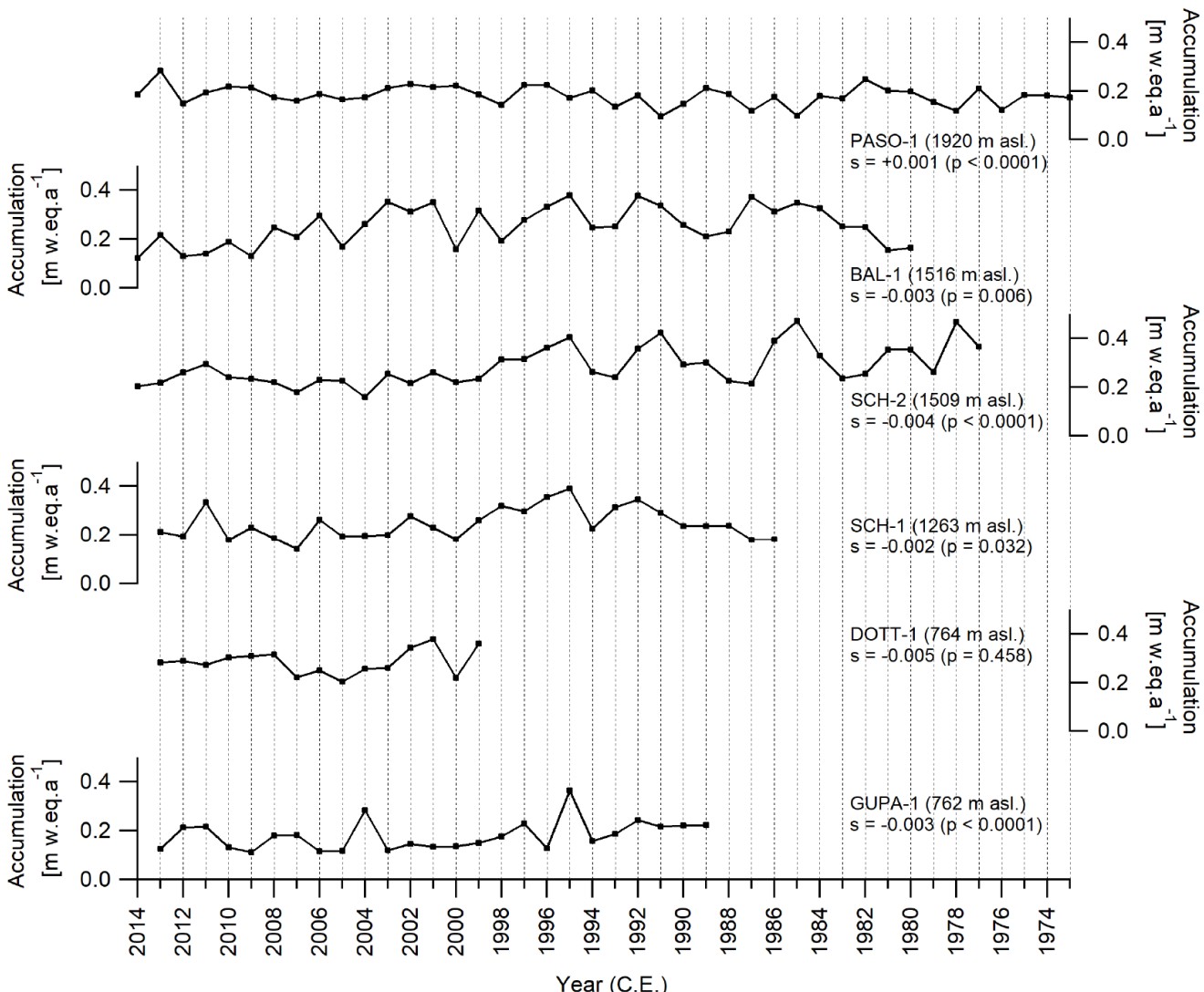

Figure 6: Annual accumulation rates at the drill sites of the six firn cores from Union Glacier for the period covered by the respective firn core. Sen slopes (s in m w.eq.a$^{-1}$) and p-values (p) are given for all firn cores indicating that snow accumulation has decreased at most sites since the beginning of the record period.

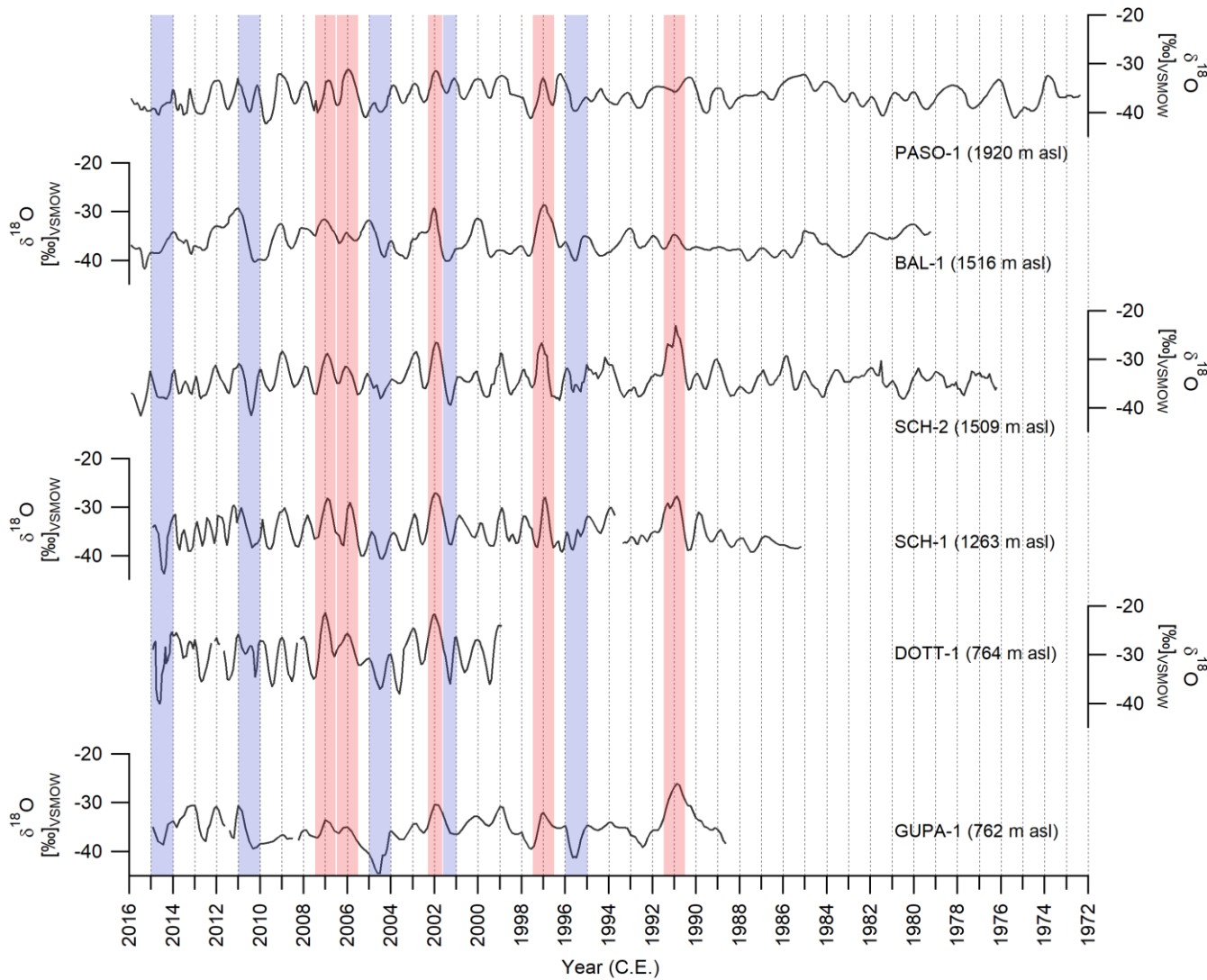

Figure 7: Profiles of the stable water isotope composition (δ<sup>18</sup>O) of the six firn cores from Union Glacier with respect to time. Years with prominent maxima and minima in δ<sup>18</sup>O-time series - although not visible in all cores - are highlighted by red and blue shading, respectively. The peak in summer 2002 is the only maximum found in all cores.

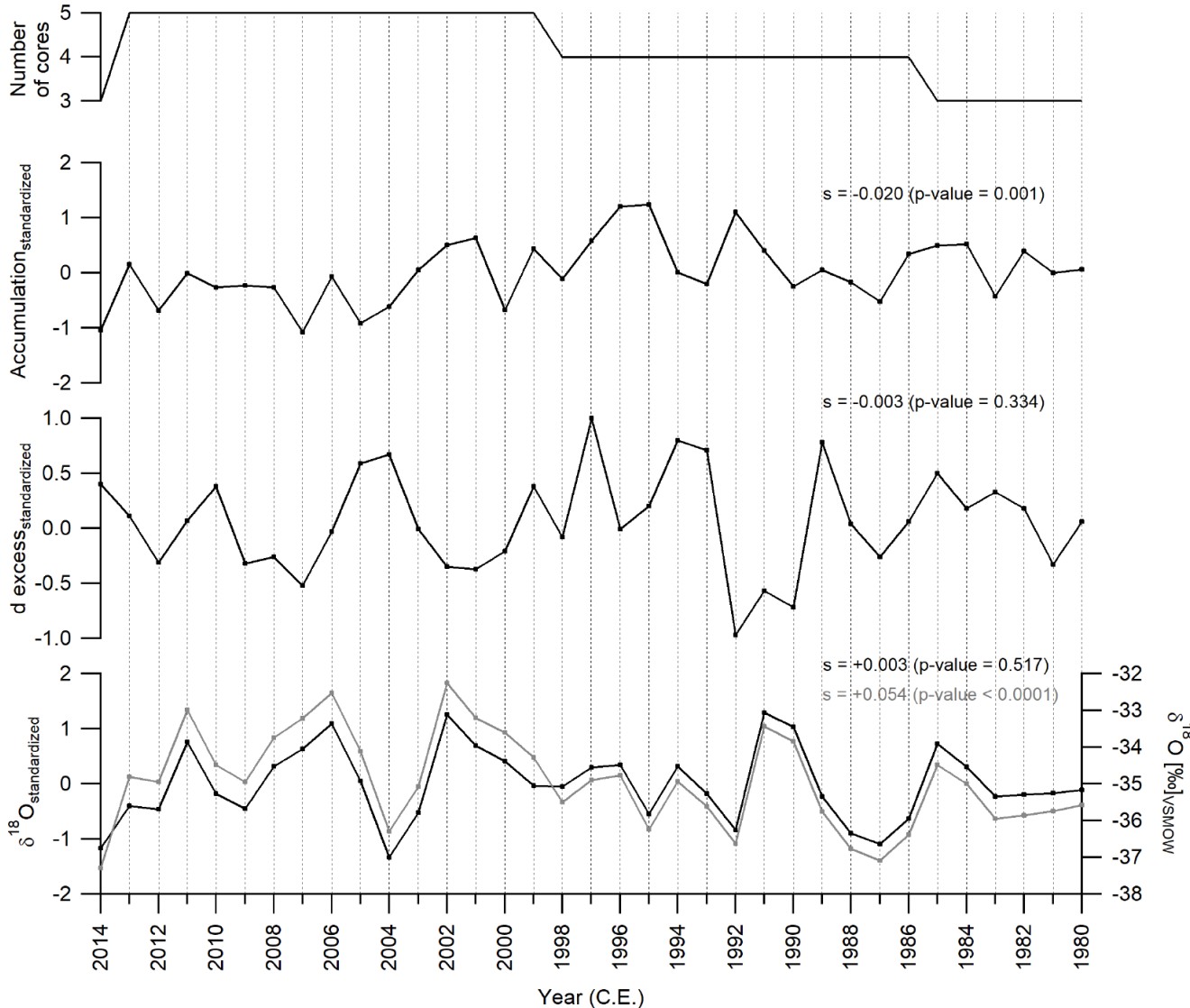

Figure 8: Composite records of mean annual δ<sup>18</sup>O, d excess and snow accumulation in the UG region for the period 1980-2014 derived from standardized data (black; excluding firn core GUPA-1). Sen slopes (s) and p-values (p) are given for each record. For δ<sup>18</sup>O, the composite record derived from non-standardized annually averaged data (grey) is also displayed as it exhibits a worth mentioning statistically significant positive trend (s in ‰ a<sup>-1</sup>).

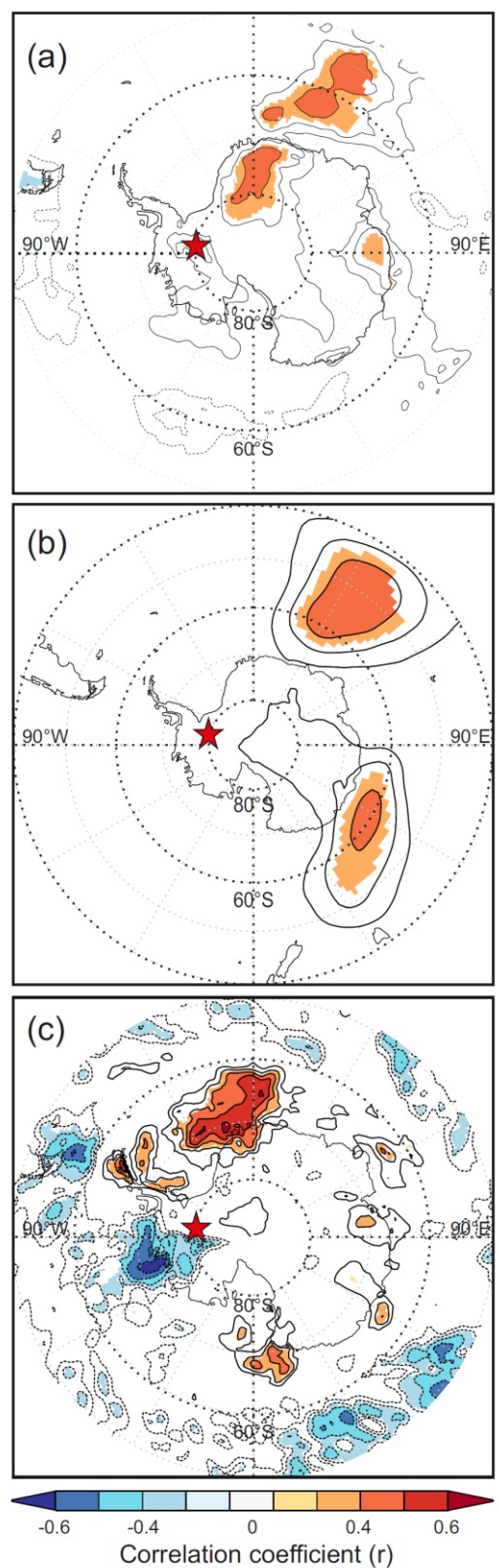

Figure 9a-c: Spatial correlations of annually averaged ERA-Interim (a) near-surface air temperatures (2 m), (b) geopotential heights (850 mbar) and (c) precipitation-evaporation with standardized mean annual (a) $\delta^{18}O$, (b) d excess and (c) snow accumulation in the UG region (red star) for the period 1980-2014 (excluding firn core GUPA-1). All time series were detrended before calculating spatial correlations. Only statistically significant correlations ($p < 0.05$) are displayed. For (a) and (c) December-November annual averages (calendar year) were used, whereas for (b) August-July annual averages (winter-winter) were considered as spatial correlations appear more significant than for December-November annual averages.

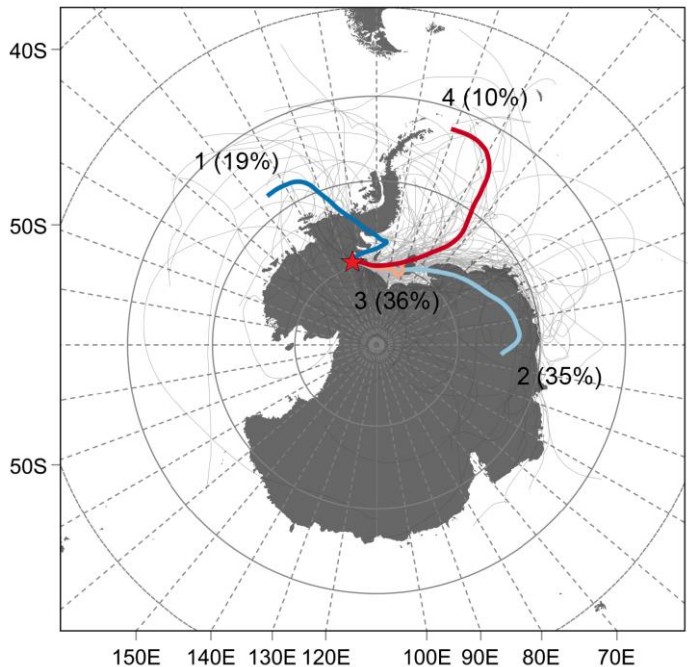

Figure 10: Cluster analysis of 5-day backward trajectories transporting air masses towards the UG region (red star) as calculated for all days with precipitation events (in total 121, represented by thin grey lines) for the period 2010-2015 using the model HYSPLIT. Bold lines represent main transport pathways calculated as percentage (%) of all trajectories associated with the respective cluster.

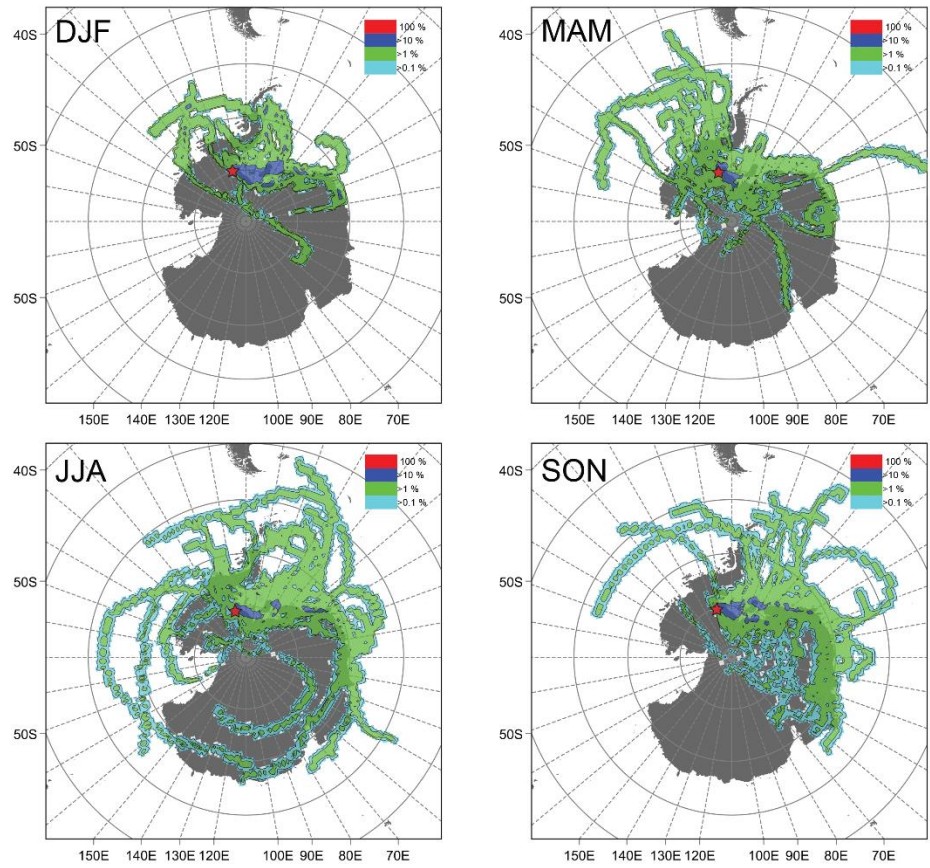

Figure 11: Seasonal frequency distribution of single 5-day backward trajectories during precipitation events extracted from ERA-Interim time series of daily precipitation for the UG region (red star) for the period 2010-2015. In total 121 precipitation events with ≥1% of the mean annual snow accumulation (~2.5 mm d$^{-1}$) were considered for this analysis performed with the HYSPLIT model.

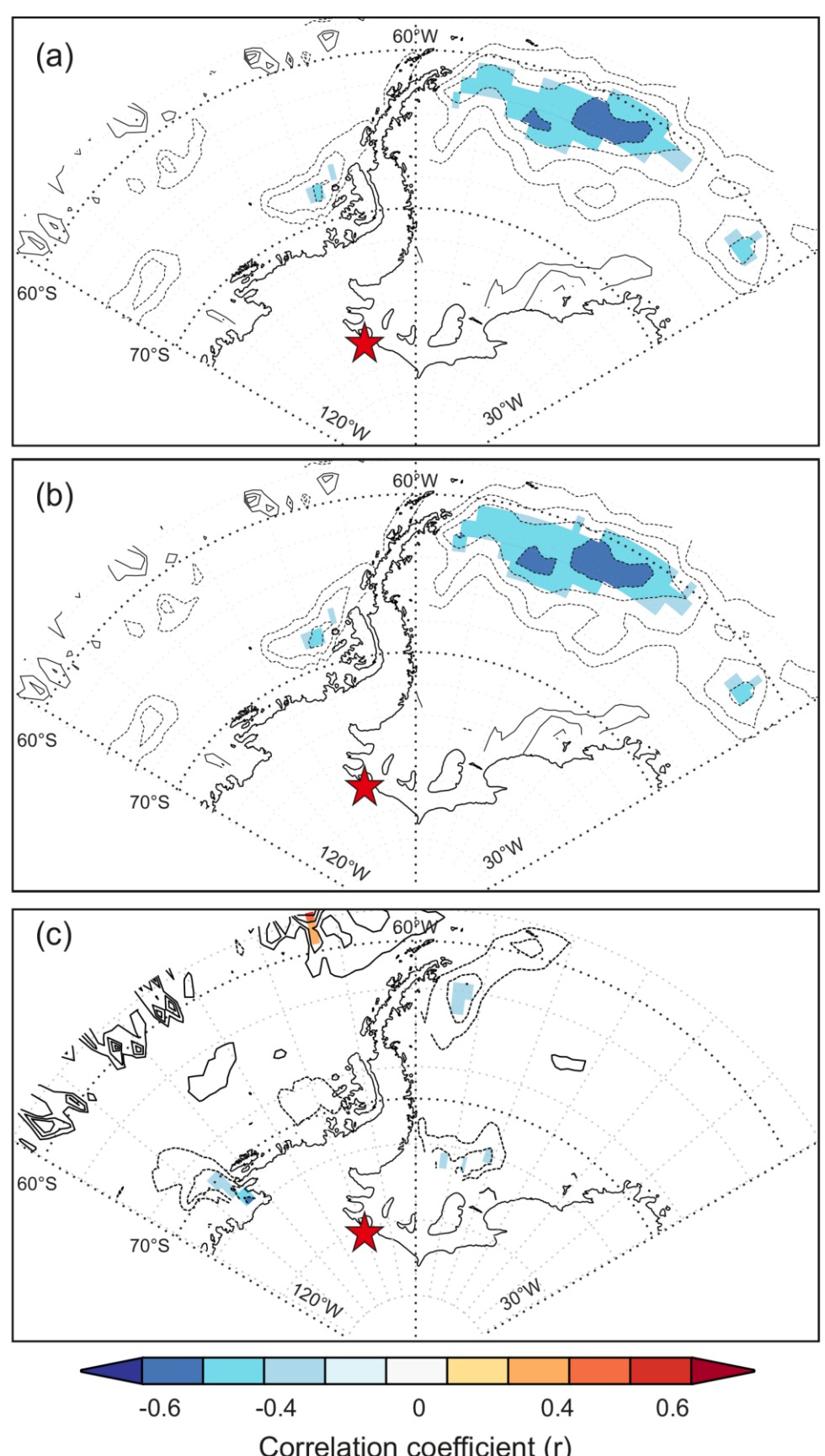

Figure 12a-c: Spatial correlations of mean annual sea ice concentrations in the Weddell and Bellingshausen Sea sectors with standardized mean annual (a) $\delta^{18}O$, (b) d excess and (c) snow accumulation in the UG region (red star) for the period 1980-2014 (excluding firn core GUPA-1). For the calculations all time series were detrended and December-November annual averages (calendar year) were considered. Only statistically significant correlations ($p < 0.05$) are shown.

Table 1: Details on drill locations and basic statistics of the stable water isotope composition and accumulation rates of the six firn cores retrieved from Union Glacier. Basic statistics are also given for the composite stable water isotope and accumulation records spanning the period 1980-2014. In addition, minimum, mean and maximum values of stable oxygen isotope annual means and accumulation rates are given for the period covered by all cores (1999-2013).

| Firn Core | GUPA-1 | DOTT-1 | SCH-1 | SCH-2 | BAL-1 | PASO-1 |
|---|---|---|---|---|---|---|
| Coordinates | 79°46'07.00"S 82°54'33.44"W | 79°18'38.84"S 81°39'09.33"W | 79°31'14.02"S 84°08'56.48"W | 79°33'17.76"S 84°03'11.46"W | 79°31'27.69"S 84°26'32.09"W | 79°38'00.68"S 85°00'22.51"W |
| Altitude [m asl] | 762 | 764 | 1263 | 1509 | 1516 | 1920 |
| Depth [m] | 9.58 | 9.57 | 14.13 | 20.25 | 17.28 | 15.04 |
| Drilling date | Nov 2014 | Nov 2014 | Nov 2014 | Nov 2015 | Nov 2015 | Nov 2015 |
| Age [period/years] | 1989-2014 (26) | 1999-2014 (16) | 1986-2014 (29) | 1977-2015 (39) | 1980-2015 (36) | 1973-2015 (43) |
| Coordinates and elevation [m asl] of nearest ERA-Interim grid point | 79°30'S 83°15'W 911.3 | 79°30'S 81°45'W 641.5 | 79°30'S 84°00'W 1061.2 | 79°30'S 84°00'W 1061.2 | 79°30'S 84°45'W 1208.4 | 79°30'S 84°45'W 1208.4 |
| $\delta^{18}O$ [‰] | | | | | | |
| Min | -44.5 | -40.0 | -43.7 | -41.6 | -41.7 | -42.3 |
| **Mean** | **-35.6** | **-29.9** | **-35.0** | **-34.1** | **-36.2** | **-36.6** |
| Max | -26.1 | -21.3 | -27.1 | -23.1 | -28.5 | -31.1 |
| Sdev | 3.1 | 3.9 | 3.1 | 2.8 | 2.5 | 2.3 |
| $\delta D$ [‰] | | | | | | |
| Min | -352.1 | -314.6 | -346.6 | -331.2 | -330.7 | -333.7 |
| **Mean** | **-278.6** | **-233.1** | **-273.3** | **-268.0** | **-284.1** | **-285.9** |
| Max | -202.5 | -163.9 | -211.2 | -187.1 | -222.3 | -240.2 |
| Sdev | 25.0 | 31.6 | 24.6 | 23.2 | 20.3 | 19.2 |
| d excess [‰] | | | | | | |
| Min | -0.2 | -2.3 | -2.3 | -5.6 | -0.4 | 0.3 |
| **Mean** | **5.8** | **5.8** | **6.5** | **4.9** | **5.5** | **7.0** |
| Max | 11.8 | 10.5 | 15.5 | 17.2 | 9.7 | 11.5 |
| Sdev | 2.6 | 2.2 | 2.6 | 3.7 | 1.6 | 1.7 |
| slope of co-isotopic relationship | 7.94 | 8.09 | 7.95 | 8.19 | 8.05 | 8.24 |
| n (samples) | 190 | 189 | 280 | 418 | 675 | 596 |
| $\delta^{18}O$ [‰] of annual means for 1999-2013 | | | | | | |
| Min | -41.1 | -33.4 | -37.7 | -36.0 | -38.0 | -38.4 |
| **Mean** | **-35.7** | **-29.7** | **-34.7** | **-34.1** | **-35.4** | **-36.3** |
| Max | -32.5 | -27.0 | -32.3 | -31.7 | -31.5 | -33.8 |
| Accumulation [m w.eq.a$^{-1}$] | 1989-2013 | 1999-2013 | 1986-2013 | 1977-2014 | 1980-2014 | 1973-2014 |
| Min | 0.111 | 0.203 | 0.142 | 0.159 | 0.121 | 0.096 |
| **Mean** | **0.180** | **0.284** | **0.247** | **0.285** | **0.253** | **0.181** |
| Max | 0.364 | 0.378 | 0.390 | 0.472 | 0.378 | 0.283 |
| Sdev | 0.061 | 0.052 | 0.064 | 0.078 | 0.079 | 0.039 |
| Accumulation [m w.eq.a$^{-1}$] for 1999-2013 | | | | | | |
| Min | 0.111 | 0.203 | 0.142 | 0.159 | 0.130 | 0.148 |
| **Mean** | **0.158** | **0.284** | **0.219** | **0.229** | **0.231** | **0.198** |
| Max | 0.284 | 0.378 | 0.333 | 0.295 | 0.353 | 0.283 |

Table 2: Results of cross-correlation analysis for stable water isotope and accumulation composite records from Union Glacier and time series of SAM Index, Multivariate ENSO Index (MEI) and sea ice extent (SIE) in five different Antarctic sectors (annual means). Cross-correlations were calculated considering the record period that is covered by a minimum of three firn cores (1980-2014; excluding firn core GUPA-1). Prominent correlations with a low p-value (p < 0.1) are marked bold and if statistically significant (p < 0.01, α = 0.05) bold and red.

| 1980-2014 | $\delta^{18}O$ | $\delta D$ | d excess | Accumulation | SAM Index | MEI | SIE Weddell | SIE Bellingshausen-Amundsen | SIE Indian Ocean | SIE West Pacific | SIE Ross |
|---|---|---|---|---|---|---|---|---|---|---|---|
| **$\delta^{18}O$** | 1 | **0.995** | -0.214 | 0.192 | -0.089 | -0.200 | -0.247 | -0.116 | -0.036 | 0.061 | 0.116 |
| p | 0 | 0.000 | 0.217 | 0.269 | 0.613 | 0.249 | 0.165 | 0.521 | 0.841 | 0.737 | 0.520 |
| **$\delta D$** | | 1 | -0.128 | 0.188 | -0.054 | -0.214 | -0.264 | -0.106 | -0.003 | 0.085 | 0.119 |
| p | | 0 | 0.464 | 0.279 | 0.758 | 0.218 | 0.138 | 0.559 | 0.985 | 0.639 | 0.511 |
| **d excess** | | | 1 | -0.051 | **0.376** | 0.024 | -0.205 | 0.103 | **0.315** | 0.169 | -0.068 |
| p | | | 0 | 0.772 | 0.026 | 0.890 | 0.253 | 0.570 | 0.074 | 0.348 | 0.707 |
| **Accumulation** | | | | 1 | -0.110 | 0.055 | -0.056 | 0.127 | **-0.295** | -0.005 | -0.140 |
| p | | | | 0 | 0.531 | 0.753 | 0.759 | 0.482 | 0.095 | 0.980 | 0.438 |
| **SAM Index** | | | | | 1 | -0.101 | -0.223 | -0.213 | **0.629** | **0.403** | **0.390** |
| p | | | | | 0 | 0.563 | 0.211 | 0.233 | 0.000 | 0.020 | 0.025 |
| **MEI** | | | | | | 1 | 0.165 | 0.171 | **-0.411** | 0.025 | **-0.512** |
| p | | | | | | 0 | 0.358 | 0.343 | 0.018 | 0.891 | 0.002 |

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
