# Peer review of "Stable water isotopes and accumulation rates in the Union Glacier region, Ellsworth Mountains, West Antarctica over the last 35 years"

_The Cryosphere, 2018_

## Referee Comment (RC1) · S. Goursaud (Referee) · 8 Nov 2018

Summary This study describes new stable water isotope and surface mass balance records from six ices cores in the Ellsworth region, at the crossing point between West Antarctic Ice Sheet, East Antarctic Ice Sheet, and the Peninsula. This region is poorly understood in terms of climate variability. Thus, the new datasets provide substantial inputs for extending our current knowledge of recent climate variability of Antarctica, and the manuscript fully fits within the framework of TC. However, I have a few major concerns. First, I suggest that the paper should be re-articulated at some points, to show more explicitly the results which are robust, the uncertainties and clarify the

underlying hypotheses. The methods used in this study, which are mainly based on statistics, should be justified, and the associated uncertainties or confidence levels should be reported. I thus recommend this study to be accepted after some revisions.

General comments This coastal region is particular, as it is located in the mountains. A spatio-temporal variability in surface mass balance could be then expected, related to wind drift, but this aspect is not mentioned.

Introduction This introduction is quite difficult to follow, and not enough explicit about the scientific questions which are addressed in the manuscript. I provide hereafter suggestions to make it more understandable (specific comments). There is limited information and citations of existing literature related to climate variability in the region of the Ellsworth Mountains, as well as for the state of the art for surface mass balance and stable water isotopes. I expect the surface mass balance and winds in the region are to be highly variable both in space and time. My recommendation is to further describe the state of the knowledge for these aspects (including knowledge gaps), referring to literature specific or relevant for this region, and to explicitly frame the question of potential deposition and post-deposition effects imprinted in firn core records, as evidenced in several other regions.

Methods - For the ice core chronology, please make sure anyone can reproduce it thanks to a more detailed description. I am not sure that this is possible from the information given in the paper. For instance, Figure 2 suggests that some peaks of MSA and nssSO4 were not counted, contrary to dD peaks. Why? What makes water stable isotopes more reliable in this site? Other studies of coastal locations such as Goursaud et al. (2018b) have shown that water stable isotopes are less reliable than chemical signals for dating firn cores in Adelie Land. It would be also relevant to have an objective assessment of the age scale uncertainty due to ambiguity in peak detection. - Why did you use a Mann-Kendall test? From what I know, it was used to detect inflections (Turner et al., Nature, 2016). - When reporting the outcomes of linear correlation analyses, can you please systematically provide the correlation coefficient?

[Figure]

Also, no need to give the exact p-value. It is sufficient to precise if it is <0.05 or <0.001. - I do not fully understand the relevance of a composite signal based on individual, non-correlated signals. Can you please justify the robustness, or point the limit of such an approach? How confident are you that this reflects a common climate signal, rather than noise (and thus influenced by the number of underlying records)? Could you maybe consider principal component rather than mean to extract a signal which would explain the maximum variance, thus potentially more representative of a regional signal? - I do not think that you should make two composites based on standardised and non-standardized data. You should first try to explain why the individual signals are not correlated to decide then to consider standardised or no standardised data. If it turns out that the spatial variability results from deposition and post-deposition effects, it will be more consistent to use standardised data. Otherwise, non-standardised data should be used. Note that in Stenni et al. (2017), we used standardised data. If your initial idea differs, please specify as it is not straightforward.

Results - Results of not significant linear relationships (slope and p-value) should not be given. - A discussion of potential noise should be added. For deposition, why do not you compare your reconstructed BMS with stack data and the climate model you cite in the discussion (Part 4.2.1, p11 l1)? For the effects of isotopic diffusion, you could apply a simple diffusion model (Johnsen et al., 2000;Jones et al., 2017), or at a least evaluate if there is a loss of seasonal dO18 amplitude along the core and report the corresponding results.

Discussion - I suggest to begin by discussing potential noises (see above), before discussing the potential common climatic signal. - I recommend to dedicate a paragraph to the assessment of the dO18-temperature relationship in this region and the comparison with other Antarctic regions. Also, I do not understand why you suggest to reconstruct regional temperature based on your dO18 composite record whereas r $^2$ = 0.21. Why explains the 80% variance left? I note a contradiction between p9 l15 and your conclusion, p11 l32. - The discussion of negative slopes from the reconstructed

SMB appears surprising given the fact that slopes are in fact very close to 0.

Specific comments Introduction P2 l12: Can you please add a transition word so we understand that you move to Peninsula, eg "In AP...". P2 l15: The sentence is difficult to read, please reword it to: "Factors affecting mechanisms forcing" P2 l23: The references Âń Thompson and Wallace 2000; Thompson and Solomon, 2002; Gillet et al., 2006" are repeated in line 25. Please remove the second call to these references. P2 l28: I checked Turner et al., Intern. J of Climatology, 2013, surprised by your assertion that the positive SAM values from the 50s are partly attributed to "local confined sea-ice loss". However, I found only a suggestion that the decrease in sea-ice extension (SIE) in AP could be linked to SAM (and not the opposite). Besides, Turner et al, Nature, 2016 shows that the changes in circulation lead to a decrease in the sea ice concentration in AP (see Fig 3). Only for the two last decades, for which a shift in circulation occurred (shifting warming in AP to cooling), a positive retroaction has been noted between the increase in SIE and changes in circulation. I suggest a careful introduction to the links between regional sea ice changes and circulation changes. P2 l34: ENSO is a mode that has a specific pseudo-periodic behaviour, I thus suppose you meant "mode" instead of "cycle". P2 l34: Please add a comma after "scales". P2 l15 – P3 l3: You refer to near-surface temperature positive trends of the last decades in WAIS (1st paragraph) and AP (2nd paragraph) since the 1950s, and discuss the state-of-the-art of understanding potential causes. If you want to present your drilling site as a crossing area between WAIS, EAIS and AP, you should also write a third paragraph browsing a short state-of-the-art of recent climate variability of EAIS, citing for instance Stenni et al. (2017) (last 2k temperature reconstruction of Antarctica), and emphazing the challenge to detect any trend. Note that even a weak cooling trend is not seen in some coastal areas such as the Adelie Land (Goursaud et al., 2017 and references herein). P3 l4: Your transition in unclear. You describe changes in trends in AP temperature associated with a cooling for the first part of the 21st century, the WAIS warming amplitude being part of the natural multi-decadal variability of last 2000 years (308 in the Thomas'study, and 2000 in the Steig's one), but also the weak cooling in the EAIS.

I think that you rather give the limits of the comprehension of the very last decades climate variability at the end of the last three paragraphs, ie tackle with the WAIS warming in your paragraph from P2 l29. But please when giving this information, stress that it is the rapidity of this warming which is unprecedented. Then, introducing the need for extended observations and monitoring in Antarctica, just give a short sentence to resume the limits of our comprehension of the recent climatic variability, whatever the considered Antarctic region. P3 l11: There is a lack of data not only for the interior of Antarctica, but also for coastal areas, for instance in the Indian Ocean coastal region and in Adelie Land. You can refer to Jones et al. (2016) for the temperature, and the updated water stable isotope Antarctic database(Goursaud et al., 2018b). P3 l3: From "For the region", please add a new paragraph, where you focus on this crossing point between the three main regions of Antarctica. In this paragraph, please be more explicit on the questions that you address in this study. I also recommend to report the fundamental literature related to the climate of Ellsworth region. P4 l1: change "WAIS in the south" to "southern WAIS".

2 Data and Methodology 2.1 P4 l9: I cannot find neither in the paragraph nor in the Table 1 the ray in which the drilling sites are located. Could you please add it here? I would be also very interested in knowing here which sites are in crests or peaks, as surface mass balance should differ between these sites (Agosta et al., 2018), as well as the isotopic signature, at least at the second order. p4 l15: Please add a comma after "(2012)". P4 l19, P4 l25: Please add a space between numbers and units throughout the manuscript. P24 l4: How can you quantify that the precision is better than a specific threshold. Please give a precise uncertainty. P5 l9: I understand that you compute a local meteoric water line based on ice core data, and especially only 6 points, which can be affected by deposition and post-deposition effects.

2.2 P5 l13: which ratio did you use to compute nssSO4? Did you use summer or winter ratio (Jourdain and Legrand, 2002)? P5 l15: Please provide references for the seasonality of the aerosol signals preferentially for your region. For instance, in Adelie

Land, there is no seasonal cycle in ssNa, so please check. You can also cite recent studies which apply such an annual layer counting to date firn ice cores and discuss the uncertainties (Vega et al., 2016;Caiazzo et al., 2016). P5 l18: How did you estimate your uncertainty? I would have liked to see on the figures 1 and 2 what constitutes this age scale uncertainty (e.g. peaks that you did not count). P5 l17: I do not fully understand the rationale behind the method used to match GUPA, DOTT, SCH-1 and BAL-1 dating to SCH-2: are the isotope records highly correlated, justifying such a method? (what is the implicit hypothesis and can you test it)? P5 l23: Why did you use non-standardized data for composites? Please take into consideration the general comments. P5 l24: Why did you use the Mann-Kendall tests? It is usually used to detect inflections. Is it the case here? Please justify.

2.3 P6 l4: please add a line break after "isotopes and accumulation." to split observations from climatic modes.

Results 3.1 P6 l24: what initial point (coordinates and height) did you indicate for back-trajectory simulations? Which drilling site does it correspond to? P6 l29 and P7 l1: change "was" to "is". P7 l1: you choose in the manuscript the convention m we a-1 for accumulation, so please change "m/s" to "m s-1". Also please change "was" to "is".

3.2 P7 l8: "the longest record" P7 l11: what kind of extrapolation did you apply? P7 l11: please replace "furthermore" by another word as repeated from previous sentence. Could you give the uncertainties associated with your dating for each firn core at the end of this paragraph? It is actually results and not methods.

3.3 p7 l16: As you give the dO18 mean range, I do not think it is necessary to also give dD. I would also rather go for mean and standard deviation, which give more information about the variability. P7 l21: These values are not so low. If you refer to Figure 6 in Goursaud et al. (2018a), that shows the spatial variability of d, you will notice that d can reach minimum values of 0 to 4 per mille in Ross sea, and Amery sector, but also close in the Ellsworth sector ! P7 l25: Please change into the brackets to "range values

from . . . to...". I am not convinced by the method used to estimate the LMWL obtained based on ice core data. P7 l27 to 30: there is a low spatial variability of your mean reconstructed SMB, and thus you could just give the mean and standard deviation of the SMB averages, instead of describing the core with the highest and lowest SMB. Details are then given in Table 1. Then, your next message is substantial as you show that particular high (low) values are not concomitant between the firn cores.

Discussion 4.1.1 p8 l5 and p8 l8: I do not see the point to report minimum and maximum values. What is your message from such information? You could first discuss the differences in the mean dO18 values from one ice core to another noted in the results, and consistent as your write with continental effect. You could then discuss the lack of similarities in inter-annual variabilities (remaining results?), and confirming here by testing the correlation between the different dO18 over the overlapping time period 1998-2013. Here, you could write that 2002 is a maximum value in all firn core data. There is some ambiguity on the structure of the manuscript, where some key results are reported in the discussion, and not in the section on results. P8 l14 to l16: relationships which are not significant, are useless. Please remove it. The two positive trends are results and not discussed here, so it should go to the "results" part. P8 l17: I do not find that dO18 firn cores data are well correlated. Some are not correlated at all, eg PASO-1 with DOTT-1 etc. . ., and the highest r is 0.658, ie $r^2$ of 0.433, so I really would not go for a regional signal by a simple average of the time series. In the assessment of potential trends, the discussion should be explicit that only two of your firn cores present such trends (P8 l21). P8 l22: Why did you prefer Sen slope rather than linear simulations? P8 l23: You cannot conclude an increase in near-surface temperature from a positive d18O trend (which is I think not robust, see what I wrote before), whereas you did not test the multi-year dO18-T relationship in your region. P8 l23 to l32: You discuss here the inter-annual variability in temperature. It should be in another paragraph dedicated to it, and using the results of dO18-temperature relationship you find in your data, and that should be cited in the results. P8 l33 – P9 l6: Is it Sen slopes of from linear regressions? If linear, please give the correlation coefficients for each

simulated linear relationship. I am very surprised for such different trends. Either the relationships are very weak or trends can be neglected, or other effects than change in moisture origin might act here to explain the differences, as your drilling sites are relatively close to each other. Once more, I do not find it robust at all to make a mean for time series showing opposite trends, and where only two firn cores are correlated (DOTT-1 and SCH1).

4.1/2 This paragraph should come before discussing a potential temperature reconstruction.

P9 l12: You could compare near-surface temperature from the wx7 and Arigony AWS with ERA-interim. If the correlation is strong enough, you could test the linear relationship between dO18 and the near-surface temperature over longer period than for observations, thus using ERA-interim temperature. You could also test the relationship with each of your firn core. Have you considered that local processes could affect the signal, and could be more important at some locations? P9 l15-16: You cannot use dO18 to reconstruct the temperature: the linear relationship is much to weak (l10), and this sentence is not consistent with l11: "However, a proper inference of near-surface temperature from dO18 values of precipitation in the UG region is not yet possible".

4.1.3 This part is very interesting and show the potential of the isotopic signal to provide information about the Weddell sea ice extent. It would be valuable to describe these results with due care, as they are based on composite analyses from individual series that are weakly correlated, challenging the confidence in a strong common climate signal. What is the likelihood of obtaining a link with sea ice extent using pure noise with a given frequency range? p10 l5: why do you suggest this correlation to be an artefact? It is very weak (r = 0.315, ie $r^2$ = 0.0992 !).

4.2.1 p10 l22: you do not need to give the precise value of p-values. Just write p<0.001 or p<0.05. Change throughout the manuscript. P10 l24: Remove the insignificant relationship for firn core DOTT-1. P10 l22-26: Deposition effects related to wind should

be mentioned at the beginning of your study, as it can significantly modify your signal. P11 l2: Precise how far are the stake measurements from the firn cores, and the grid resolution of the model. What are the covered periods? P11 l3: Can you also precise the location of Patriot Hills and the shortest distance between the closest ITASE ice core and your firn cores. P11 l6: Could you compare the inter-annual from your firn cores with the stake measurements and model outputs? We have shown in our paper under review, the added value of such time series 1) to make the dating more robust, and 2) extract a regional signal.

4.2.2 P11 l22: you suggest that ERA-interim fails to capture the effects of orography. You can show it by giving the surface height of the grid covering the drilling sites, while these ones differs from each other. You suggest also at the very end that your data could be affected by post-deposition effects. But couldn't you test it, for instance by looking at the evolution of the seasonal amplitude along the cores (see again Goursaud et al., 2018b), or applying a proper diffusion model as done in Jones et al., J of Geoph. Research., 2017.

Conclusions P12 l10: Remove the slope for non-standardized data.

Can you make the data available, either in supplementary material, on any public access depositary?

Figures and tables Figure 1b: Names of the drilling sites are hardly readable (except GUPA-1 and DOTT-1). Figure 2: What does "smooth 2p" correspond to? It is not specified in the legend or caption. Specify that depth is in water equivalent in unit (as well as for the following figures). Figure 3: Why did you use nssSO4/ssNa? What does it correspond to? This is not explained in the paper. How can you justify that you did not count peaks $\sim$ 1.2 and $\sim$5.2 m w.e.? Figure 4: Monthly mean from Arigony AWS is hardly readable. Figure 5: I suggest to move these figures to supplementary material. Figure 6: for all figures displaying annual-scale data, can you draw it with cityscape vectors? Once more, slope is almost equal 0. Thus, the discussion of negative trends is not a robust finding. Figure 7: Please remove the composite based on non-standardized data or standardised data, and use cityscape vectors. Table 2: Periods for reconstruction are given it Table 1. Please move this table to supplementary material.

References Agosta, C., Amory, C., Kittel, C., Orsi, A., Favier, V., Gallée, H., van den Broeke, M. R., Lenaerts, J. T. M., van Wessem, J. M., and Fettweis, X.: Estimation of the Antarctic surface mass balance using MAR (1979-2015) and identification of dominant processes, The Cryosphere Discuss., 2018, 1-22, 10.5194/tc-2018-76, 2018. Caiazzo, L., Becagli, S., Frosini, D., Giardi, F., Severi, M., Traversi, R., and Udisti, R.: Spatial and temporal variability of snow chemical composition and accumulation rate at Talos Dome site (East Antarctica), Science of the Total Environment, 550, 418-430, 2016. Goursaud, S., Masson-Delmotte, V., Favier, V., Preunkert, S., Fily, M., Gallée, H., Jourdain, B., Legrand, M., Magand, O., and Minster, B.: A 60-year ice-core record of regional climate from Adélie Land, coastal Antarctica, The Cryosphere, 11, 343-362, 2017. Goursaud, S., Masson-Delmotte, V., Favier, V., Orsi, A., and Werner, M.: Water stable isotope spatio-temporal variability in Antarctica in 1960–2013: observations and simulations from the ECHAM5-wiso atmospheric general circulation model, Clim. Past, 14, 923-946, 10.5194/cp-14-923-2018, 2018a. Goursaud, S., Masson-Delmotte, V., Favier, V., Preunkert, S., Legrand, M., Minster, B., and Werner, M.: Challenges associated with the climatic interpretation of water stable isotope records from a highly resolved firn core from Adélie Land, coastal Antarctica, The Cryosphere Discuss., 2018, 1-55, 10.5194/tc-2018-121, 2018b. Johnsen, S. J., Clausen, H. B., Cuffey, K. M., Hoffmann, G., Schwander, J., and Creyts, T.: Diffusion of stable isotopes in polar firn and ice: the isotope effect in firn diffusion, Physics of ice core records, 2000, 121-140, Jones, J. M., Gille, S. T., Goosse, H., Abram, N. J., Canziani, P. O., Charman, D. J., Clem, K. R., Crosta, X., De Lavergne, C., and Eisenman, I.: Assessing recent trends in high-latitude Southern Hemisphere surface climate, Nature Climate Change, 6, 917-926, 2016. Jones, T., Cuffey, K., White, J., Steig, E., Buizert, C., Markle, B., McConnell, J., and Sigl, M.: Water isotope diffusion in the WAIS Divide ice core during the Holocene and last glacial, Journal of Geophysical Research: Earth Surface,

122, 290-309, 2017. Jourdain, B., and Legrand, M.: Year around records of bulk and size segregated aerosol composition and HCl and HNO3 levels in the Dumont d'Urville (coastal Antarctica) atmosphere: Implications for sea salt aerosol fractionation in the winter and summer, Journal of Geophysical Research: Atmospheres, 107, 1-13, 2002. Stenni, B., Curran, M. A., Abram, N. J., Orsi, A., Goursaud, S., Masson-Delmotte, V., Neukom, R., Goosse, H., Divine, D., and Van Ommen, T.: Antarctic climate variability on regional and continental scales over the last 2000 years, Climate of the Past, 13, 1609-1634, 2017. Vega, C. P., Schlosser, E., Divine, D. V., Kohler, J., Martma, T., Eichler, A., Schwikowski, M., and Isaksson, E.: Surface mass balance and water stable isotopes derived from firn cores on three ice rises, Fimbul Ice Shelf, Antarctica, The Cryosphere, 10, 2763-2777, 2016.

---

## Referee Comment (RC2) · M. Frezzotti (Referee) · 19 Nov 2018

This manuscript provides a new stable records over the last 35 years of water isotope and snow accumulation from 6 firn cores of the Ellsworth Mountains area and ice rise on Filchner-Ronne Ice Shelf . The result of isotope and accumulation records are compared with re-analysis and large-scale modes of climate variability such as the Southern Annular Mode (SAM) and the El Niño–Southern Oscillation (ENSO) and sea ice extent.

The water isotope and snow accumulation records are very valuable because are representative of an area with very limited records. While I believe that this manuscript will

make an important contribution for the characterisation of climatic history of this area, major comments should be addressed before its publication.

The title referred to Union Glacier is misleading, Ellsworth Mountains probably is more appropriate.

The firn cores where collected in a complex area of about 400 km2 at the boundary between WAIS plateau (PASO-1), Mountain glacier (BAL-1, SCH-1/2), outlet glacier/blue ice area (GUPA-1) and Ice rise on Filchner-Ronne Ice Shelf (DOTT-1). The Authors must be describing the site cores from morphological and climatological of point view and taking well in account their location/characteristic during the interpretation of the data, not only elevation and distance from the open sea determine the snow fall intensity and relative isotope compositions.

The storms that provide snow precipitation could be "similar" for all 6 cores, but the orographic effect on precipitation and the post depositional effect could be very different, as the records shown. Significant wind drift occurs at AWS with mean wind speed of 6.9 m-1, this agrees with the extensive presence of blue ice along Union Glacier, in particulate for GUPA-1. At this site probably the anthropogenic effect is limited respect to wind scouring. The transportation by suspension (drift snow) starts at velocities greater than 5 m s-1 (within 2 m), and blowing snow (snow transportation higher than 2 m) starts at velocities of 7 m s-1.

The authors compare the data without a clear analysis of the ratio between signal vs noise and their representativeness at local/regional scale (see ex RUPPER et al., 2015, Eisen et al., 2008) .

The Authors must be provide firstly evidence of a correlation with the ERA re-analysis with meteo station and/or core records before looking at large scale modes such as SAM or ENSO. The assumption of relationship between snow accumulation/isotope temperature with sea ice extent must be demonstrate in general. The d-excess is correlated to the moisture source region (sea ice) and distillation effect along the trajectory, instead the oxygen and deuterium rate are strictly correlated to snow precipitation temperature, their seasonality and frequency.

The Authors compares the result mainly with the coastal part of WAIS (Amundsen and Bellingshausen Sea) and AP, with very small attention to the closer Filchner-Ronne Ice Shelf (Berkner island ex.), WAIS inner site (Kaspari, et al., Burgener et al., ) and DML (Coats Land) with analogous moisture source area Weddell Sea.

The area is not a "coastal area", open sea is around 1000 km far from Weddell Sea

Detail:

Introduction too long, without a clear finalization of the paper.

2.3 dating,

NO clear evidence of Pinatubo nssS signal in SCH-2 and PASO-1, value similar to other annual peaks (SCH-2) or much lower (PASO-1) .

How as been composed and which is the grade of confidence of the time series at annual scale if the error associated to ALC vary from 1 yr (2 cores) to 2 years (4 cores) and without taking in account the ratio of signal/noise due to sastrugi?

See Noise vs signal , 1985 snow accumulation at SCH-2 vs PASO-1 pag 7

Which is the cross correlation between the different cores for isotope and accumulation?

2.4 Which is the difference between the two AWS station? show both data in figure 4

Which threshold of snow precipitation is used from ERA Interim for HYSLPIT? Why the analysis is performed only 4 years from 2010 to 2014?

3.3 SCH-2/1 and BAL-1 are within 10 km and show similar accumulation, the comments about higher and lower accumulation should be addressed for these sites also at annual scale or better a pluriannual (eg. 3 years), to see the ratio signal/noise.

SCH-2 is isotopic "less depleted" than SCH1 with 250 m of difference in elevation and BAL-1 at the same elevation, PASO1 presents a similar isotope mean ith Bal1 with 400 m of difference in elevation. GUPA-1 is "more depleted" than SCH-1/SCH2 with a difference in elevation of 500-800 m. Before any consideration in discussion about the isotope and accumulation some comments must be addressed on these difference and their significant, also in comparison with the other core sites in different geographical position and much far.

4.1.2 Line17-21, Is ERA-Interim or the staked records that are not able to capture the Climate of Ellsworth Mountain? ERA-interim should be firstly compared with AWS data and than with firn records. Isotope and snow accumulation represent the snow fall events plus the noise due to post-deposition process, the absence of correlation with ERA-Interim must be better analysed also in comparison with AWS.

4.2.1

PASO-1 presents an accumulation from 37 to 27% less than SCH1/2 and BAL1.

No sense the average value of 0.25 weq a-1 and their comparison with other measurements at hundred km far.

Line 7-9 These data must be compared with the closer site of inner WAIS, Filchner-Ronne Ice Shelf and DML (Coats Land) instead of AP and Coastal Ellsworth Land with difference moisture source.

---

## Referee Comment (RC3) · Anonymous Referee #3 · 2 Dec 2018

This manuscript presents a new dataset of stable water isotopes and accumulation rates from firn cores in the Ellsworth mountains at the northern edge of the West Antarctic Ice Sheet for the period 1980 – 2014. As measurements at the intersection between the Antarctic peninsula, the West Antarctic Ice Sheet and the East Antarctic Ice Sheet are particularly sparse, this new dataset is an important contribution toward a better understanding of climate variability in this region. I see the value of the manuscript therefore primarily in the publication of this dataset, while the accompanying meteorological analysis is limited. This is acceptable, as the dataset itself deserves a publication, but I recommend minor revisions before final publication.

[Figure]

General comments

The drilling and measuring process is nicely explained, but I miss details on the trajectory analysis. Where did you start the trajectories from (lat/lon/lev), and how many per precipitation event (I hope more than one)? Also three days might not be enough, and the origin of the trajectories does not always reflect where the moisture comes from, because some trajectories could be very dry and not contribute to precipitation at all. I suggest using the moisture source diagnostic from Sodemann et al. (2008), which is also based on backward trajectories, but specifically identifies moisture uptake regions.

Referring to the lack of correlation between the firn core data set and local ERA-Interim data, you speculate that either the model is unable to capture the orography of the Ellsworth mountains or this is evidence of post-depositional processes at the firn core sites. You could get a better idea which of the two is the case by correlating local ERA-Interim data with the two weather stations near the firn core sites.

Looking at Table 3 and the discussion, there are barely any significant correlations between the isotope signal / accumulation rates and the other variables, meaning that there must be other factors influencing their variability. It would be nice to have some discussion on what these factors could be.

Specific comments

Page 5, line 23: Please explain the terms standardized and non-standardized, and why you use both (why not only standardized).

Page 5, line 28: Introduce abbreviation AWS

Page 7, line 8: "t" missing in "the".

Page 7, line 23: Could you explain what you mean by that? Meteoric as opposed to what?

Page 9, line 29: Fig. 9b instead of 9c.

[Figure]

Page 9, line 32: "a significant correlation" instead of "a correlation".

Page 11, line 20: Fig. 9c instead of 9b.

Page 11, line 21: "Similar results have been found" instead of "Similar has been found".

Page 12, line 14: Delete "are".

Fig. 5: For completeness, please explain what the red dashed line and the dots show.

Fig. 6 and 7: Since the timeseries all have the same units, it might be possible to show them all in one plot (with one y axis) using different colors for the different sites. In this way it would be easier to compare them.

Fig. 10: Please add numbers to the colorbar. Why the irregular spacing?

Table 1: Why only $\delta$18O for the period covered by all cores?

References

Sodemann, H., C. Schwierz, and H. Wernli (2008), Interannual variability of Greenland winter precipitation sources: Lagrangian moisture diagnostic and North Atlantic Oscillation influence. J. Geophys. Res., 113, D03107, doi:10.1029/2007JD008503

---

## Author Comment (AC1) · 25 Sep 2019

**Reply to the the Interactive reviewers comments on "Stable water isotopes and accumulation rates in the Union Glacier region, West Antarctica over the last 35 years" by Kirstin Hoffmann et al.**

**Ref #1: S. Goursaud (Referee)**

sentia.goursaud@lsce.ipsl.fr

Summary This study describes new stable water isotope and surface mass balance records from six ices cores in the Ellsworth region, at the crossing point between West Antarctic Ice Sheet, East Antarctic Ice Sheet, and the Peninsula. This region is poorly understood in terms of climate variability. Thus, the new datasets provide substantial inputs for extending our current knowledge of recent climate variability of Antarctica, and the manuscript fully fits within the framework of TC. However, I have a few major concerns. First, I suggest that the paper should be re-articulated at some points, to show more explicitly the results which are robust, the uncertainties and clarify the underlying hypotheses.

The methods used in this study, which are mainly based on statistics, should be justified, and the associated uncertainties or confidence levels should be reported. I thus recommend this study to be accepted after some revisions.

General comments: This coastal region is particular, as it is located in the mountains. A spatio-temporal variability in surface mass balance could be then expected, related to wind drift, but this aspect is not mentioned.

*Answer: We agree that in high mountain areas post-depositional processes such as the relocation of snow due to wind drift are likely. In the old manuscript this subject is first touched when describing meteorological data (Chapter 3.1, p 7), the site-specific characteristics of firn core GUPA-1 (Chapter 3.3, p 7) and further when we compare the accumulation rates of single firn cores (Chapter 4.2.1, p 10). Chapter 4.2.1 (p 10) starts with an extensive comparison of the different firn cores and these are discussed in the framework of their site-specific characteristics, including their potential influence by wind drift (1st paragraph). We do not see the necessity to change the text. Rev#2 also requested a more detailed description of the differences between coring localities that has been added to Chapter 2.1 (p 4) in the new manuscript.*

Introduction: This introduction is quite difficult to follow, and not enough explicit about the scientific questions which are addressed in the manuscript. I provide hereafter suggestions to make it more understandable (specific comments). There is limited information and citations of existing literature related to climate variability in the region of the Ellsworth Mountains, as well as for the state of the art for surface mass bal- ance and stable water isotopes. I expect the surface mass balance and winds in the region are to be highly variable both in space and time. My recommendation is to further describe the state of the knowledge for these aspects (including knowledge gaps), referring to literature specific or relevant for this region, and to explicitly frame the question of potential deposition and post-deposition effects imprinted in firn core records, as evidenced in several other regions.

*Answer: We are grateful for this comment. We completely revised the introduction chapter and, as suggested by Rev #1, introduced with three neighboring regions EAIS, WAIS and AP) as our study site is located in the intersection area between them. We also sharpened the argumentation towards the key scientific questions to be addressed in our manuscript. Unfortunately, there is very little published data specifically for the region around Union Glacier. The only existing data (Rivera et al., 2010 and Rivera et al., 2014) is cited in the manuscript in the introduction, in the meteorology chapter (Chapter 3.1) as well as used for the discussion (Chapter 4.2). Here also the information is given about available local and regional accumulation data (Chapter 4.3.1).*

Methods - For the ice core chronology, please make sure anyone can reproduce it thanks to a more detailed description. I am not sure that this is possible from the information given in the paper. For instance, Figure 2 suggests that some peaks of MSA and nssSO4 were not counted, contrary to dD peaks. Why? What makes water stable isotopes more reliable in this site? Other studies of coastal locations such as Goursaud et al. (2018b) have shown that water stable isotopes are less reliable than chemical signals for dating firn cores in Adelie Land.

*Answer: We changed the description of the ice-core chronology and now provide more details in Chapter 2.2 of the new manuscript (p 5). We hope that our clarification fit the requirements of Rev #1. Generally, we used the SCH-2firn core chronology, which bases on annual layer counting (ALC) of stable water isotope and chemistry data, as anchor to which we compared and matched the other cores, from which only stable water isotope data are available. However, firn core PASO-1 was dated independently based on ALC of stable water isotopes and chemical parameters as well as the use of the volcanic peak of Mt. Pinatubo as tie point.*
*Regarding Figure 2: It is obvious that neither all MSA, nor $SO_4^{2-}$, nor all stable water isotope peaks are considered for the chronology of SCH-2. In general, we consider $H_2O_2$ as most reliable parameter for dating as it is directly connected to the insolation cycles (thus to solar radiation). Most maxima and minima in $H_2O_2$correspond to maxima/minima in the dD record (a temperature proxy), supporting the $H_2O_2$-based dating. In contrast, MSA and $SO_4^{2-}$ are connected to marine-biogenic processes, which reflect the presence/absence of sea ice, which in most, but not all cases show a seasonal cycle. Hence, the dating of SCH-2 is primarily based upon $H_2O_2$ with MSA and $SO_4^{2-}$ only used for corroboration and this revealed that, contrary to Goursaud (2018b), dD also seems to reflect seasonal cycles quite well.*

It would be also relevant to have an objective assessment of the age scale uncertainty due to ambiguity in peak detection.

*Answer: In order to get a first depth-age relationship, i.e. an idea of the maximum dating uncertainty, we did an independent annual layer counting based on all firn core minima and maxima in stable water isotopes (see Table 1 below this answer). Based on this approach only, the firn cores yield a maximum dating uncertainty range of ±1 year (DOTT-1) and ±6 years (BAL-1).*
*However, this was only an experiment, which has been carried out before using seasonally varying chemical parameters for constructing the final age scale. As already described above the chemistry-based age scale of SCH-2, which we consider as most reliable, has been used as reference and the other firn cores were matched to this age scale. For PASO-1 an idependent dating has been carried out, which includes a nssS peak attributed to Mt. Pinatubo. We found that for most firn cores the final age scale, i.e. the total number of years, does not correspond to the isotope-based ALC range. Furthermore, there is no systematic offset between the final age scale and the isotope-based ALC range (see Table 1). We estimated the error associated to stable water isotope and glaciochemistry ALC, firn core inter-matching as well as the volcanic peak tie point and included the following sentence in Chapter 3.2:*
*"The estimated error associated to ALC is ±1 year for cores dated with glacio–chemistry (SCH-2 and PASO-1) and ±2 years for cores dated with stable water isotopes only (GUPA-1, DOTT-1, SCH-1 and BAL-1)."*

*Table 1: Results of annual layer counting (ALC) as well as final dating results for the six firn cores from Union Glacier. For each core the minimum and maximum number of years was determined by counting only clearly pronounced peaks and by counting all visually identifiable peaks in the respective stable water isotope profiles ($\delta^{18}O/\delta D$ vs. depth). Determination of the final number of years and the corresponding period covered by each core bases on the seasonality of stable water isotopes and chemical parameters and subsequent inter–matching of stable water isotope profiles taking SCH–2 as reference.*

| Firn Core | | Winter Minima ($\delta^{18}O/\delta D$) | Summer Maxima ($\delta^{18}O/\delta D$) | ALC uncertainty (years) | Period | Total number of years (summer maxima) | Offset with respect to ALC dating (Min /Max summer maxima) |
|---|---|---|---|---|---|---|---|
| GUPA–1 | Min | 18 | 18 | ±3.5 | 1989–2014 | 26 | +8/+1 |
| | Max | 25 | 25 | | | | |
| DOTT–1 | Min | 18 | 19 | ±1 | 1999–2014 | 16 | -3/-5 |
| | Max | 20 | 21 | | | | |
| SCH–1 | Min | 35 | 35 | ±2 | 1986–2014 | 29 | -6/-10 |
| | Max | 39 | 39 | | | | |
| SCH–2 | Min | 40 | 40 | ±5 | 1977–2015 | 39 | -1/-11 |
| | Max | 50 | 50 | | | | |
| BAL–1 | Min | 28 | 28 | ±6.5/±6 | 1980–2015 | 36 | +8/-4 |
| | Max | 41 | 40 | | | | |
| PASO–1 | Min | 37 | 36 | ±4 | 1973–2015 | 43 | +7/-1 |
| | Max | 45 | 44 | | | | |

Why did you use a Mann-Kendall test? From what I know, it was used to detect inflections (Turner et al., Nature, 2016). -

**Answer:** *Answer see below (p 9).*

When reporting the outcomes of linear correlation analyses, can you please systematically provide the correlation coefficient?

**Answer:** *The correlation coefficients are consistently given in Table 3, and have been inserted in the text wherever necessary.*

Also, no need to give the exact p-value. It is sufficient to precise if it is <0.05 or <0.001.

**Answer:** *This has been done consistently throughout the text where appropriate*

I do not fully understand the relevance of a composite signal based on individual, non-correlated signals. Can you please justify the robustness, or point the limit of such an approach? How confident are you that this reflects a common climate signal, rather than noise (and thus influenced by the number of underlying records)? Could you maybe consider principal component rather than mean to extract a signal which would explain the maximum variance, thus potentially more representative of a regional signal? - I do not think that you should make two composites based on standardised and non-standardized data. You should first try to explain why the individual signals are not correlated to decide then to consider standardised

or no standardised data. If it turns out that the spatial variability results from deposition and post-deposition effects, it will be more consistent to use standardised data. Otherwise, non-standardised data should be used. Note that in Stenni et al. (2017), we used standardised data. If your initial idea differs, please specify as it is not straightforward.

*Answer: This is a fundamental and important point for every glaciological investigation and it became more common in recent years to test for depositional and post-depositional effects in order to assess the signal-to-noise ratio at a given site.*

*As also requested by Rev #2 and #3, we assessed the signal-to-noise ratios of the stable water isotope and accumulation data (compare answer to Rev #2 and #3). We estimate the signal-to-noise ratio of stable water isotopes in the UG region (of between 0.6 and 0.86) as quite high, i.e. higher than signal-to-noise ratios at WAIS, and even better than for Dronning Maud Land cores on interannual timescales (Münch and Laepple, 2018). A new chapter discussing signal-to-noise ratios at the study site has been added to the manuscript (Chapter 4.1, p 8).*

*In line with comments from Rev#2 and #3, we have decided to consistently use standardized data in order to give each firn core the same statistical value. This information has been added to the text. However, as $\delta^{18}O$ shows a statistically significant positive trend in the non-standardized data, which is not visible in the standardized data anymore, we kept this information to demonstrate the effect of standardization on $\delta^{18}O$ trends. As this is not relevant for d excess and accumulation trends, we omitted the information about non-standardized data from the manuscript in this respect. Further arguments are given below in the more specific comments.*

*Münch, T. and Laepple, T.: What climate signal is contained in decadal- to centennial-scale isotope variations from Antarctic ice cores?, Clim. Past, 14, 2053-2070, doi: 10.5194/cp-14-2053-2018, 2018.*

Results - Results of not significant linear relationships (slope and p-value) should not be given.

*Answer: The manuscript has been changed accordingly.*

A discussion of potential noise should be added.

*Answer: We discuss potential noises in a newly added chapter: 4.1 Potential noises influencing UG firn core records (see answer above).*

For deposition, why do not you compare your reconstructed BMS with stack data and the climate model youcite in the discussion (Part 4.2.1, p11 l1)?

*Answer: We do not understand this comment as the sentence Rev #1 refers to already **is** a comparison of the average accumulation rate derived from our firn cores for the the UG region with accumulation rates from model simulations and stake measurements carried out for the same and/or close-by site(s). The individual time series (stake measurements) from Rivera et al. (2014) are unfortunately not freely available. Van den Broeke (2006) provide modeled accumulation rates for a period from 1980 to 2004 only, and no data for individual years is freely available that could be compared to our dataset.*

For the effects of isotopic diffusion, you could apply a simple diffusion model (Johnsen et al., 2000;Jones et al., 2017), or at a least evaluate if there is a loss of seasonal dO18 amplitude along the core and report the corresponding results.

*Answer: We are aware that diffusion can significantly change the seasonal $\delta^{18}O$ signal in firn and ice cores, but due to the comparably high accumulation in the UG region ranging between ~0.18 m w.eq.a$^{-1}$ (GUPA-1 and PASO-1) and ~0.29 m w.eq.a$^{-1}$ (SCH-2) diffusion likely plays only a minor role at this site. However, as suggested by Rev #1 we have calculated the diffusion lengths for the six firn cores as described by Münch and*

*Laepple (2018) and Laepple et al. (2018). The meteorological data from the two AWS (air temperature and air pressure), measured surface densities and derived accumulation rates are used as input data. Accordingly, the diffusion length for the maximum depth of the firn cores (excluding GUPA-1) was determined to range between 7.2 cm (PASO-1) and 8.7 cm (DOTT-1) and is , thus much lower than the mean annual layer thickness of between 34.6 cm (PASO-1) and 59.7 cm (DOTT-1). We therefore assume diffusion to be of minor importance for inducing noise to the stable water isotope records of the UG cores. This information has been added to the manuscript (Chapters 2.4 and 4.1).*

*Laepple, T., Münch, T., Casado, M., Hoerhold, M., Landais, A. and Kipfstuhl, S.: On the similarity and apparent cycles of isotopic variations in East Antarctic snow pits, The Cryosphere, 12, 169–187, doi:10.5194/tc-12-169-2018, 2018.*

Discussion - I suggest to begin by discussing potential noises (see above), before dis- cussing the potential common climatic signal. - I recommend to dedicate a paragraph to the assessment of the dO18-temperature relationship in this region and the comparison with other Antarctic regions. Also, I do not understand why you suggest to reconstruct regional temperature based on your dO18 composite record whereas $r^2$ =0.21. Why explains the 80% variance left?

*Answer: We gratefully acknowledge the commentary of Rev #1. We introduced a new chapter at the beginning of the discussion part (Chapter 4.1: Potential noises influencing UG firn core records). In line with the other reviewers comments we incorporated the signal-to-noise calculations in this new discussion chapter.*
*We then continue with the temporal and spatial variability of the stable water isotope data and its relationship to meteorological data. In fact, we did (and do) not want to suggest that all isotopic changes are related to climate, but for assessing this, we first need to correlate near-surface air temperature (T) and δ-values. The correlation coefficient between T and $\delta^{18}O$ is rather low, correct, but statistically significant. Nonetheless, we reduced the emphasis on the interpretation of temperature variability in the whole manuscript.*

I note a contradiction between p9 l15 and your conclusion, p11 l32

*Answer: As we do not emphasize the temperature interpretation from the stable water isotope data anymore, this contradiction does not further exist.*

The discussion of negative slopes from the reconstructed SMB appears surprising given the fact that slopes are in fact very close to 0.

*Answer: See answer to specific comments below (p 17).*

Specific comments

Introduction

P2 l12: Can you please add a transition word so we understand that you move to Peninsula, eg "In AP. . .".

*Answer: Sentence changed to: "For the AP, time series of near-surface air temperature from weather stations (Vaughan et al., 2003; Turner et al., 2005a) as well as from stable water isotope records from ice cores (Thomas et al., 2009; Abram et al. 2011; Stenni et al., 2017) provide evidence of a significant warming over the last 100 years reaching more than 3°C since the 1950s."*

P2 l15: The sentence is difficult to read, please reword it to: "Factors affecting mechanisms forcing"

*Answer: Sentence reworded to "Factors affecting mechanisms that force ..."*

P2 l23: The references "Thompson and Wallace 2000; Thompson and Solomon, 2002; Gillet et al., 2006" are repeated in line 25. Please remove the second call to these references.

*Answer: Citations removed.*

P2 l28: I checked Turner et al., Intern. J of Climatology, 2013, surprised by your assertion that the positive SAM values from the 50s are partly attributed to "local confined sea-ice loss". However, I found only a suggestion that the decrease in sea-ice extension (SIE) in AP could be linked to SAM (and not the opposite). Besides, Turner et al, Nature, 2016 shows that the changes in circulation lead to a decrease in the sea ice concentration in AP (see Fig 3). Only for the two last decades, for which a shift in circulation occurred (shifting warming in AP to cooling), a positive retroaction has been noted between the increase in SIE and changes in circulation. I suggest a careful introduction to the links between regional sea ice changes and circulation changes.

*Answer: As our interpretation with SAM in the discussion part is very short, we decided not to mention the connection between SAM and sea ice in the introduction, and omitted the part of the sentence about the "local confined sea-ice loss".*

P2 l34: ENSO is a mode that has a specific pseudo-periodic behaviour, I thus suppose you meant "mode" instead of "cycle".

*Answer: "cycle" changed to "mode".*

P2 l34: Please add a comma after "scales".

*Answer: Comma added.*

P2 l15 – P3 l3: You refer to near-surface temperature positive trends of the last decades in WAIS (1st paragraph) and AP (2nd paragraph) since the 1950s, and discuss the state- of-the-art of understanding potential causes. If you want to present your drilling site as a crossing area between WAIS, EAIS and AP, you should also write a third paragraph browsing a short state-of-the-art of recent climate variability of EAIS, citing for instance Stenni et al. (2017) (last 2k temperature reconstruction of Antarctica), and emphazing the challenge to detect any trend. Note that even a weak cooling trend is not seen in some coastal areas such as the Adelie Land (Goursaud et al., 2017 and references herein).

*Answer: We acknowledge this commentary and added a third paragraph about EAIS in the Introduction. We also wrote a summary paragraph about the challenges to detect trends in climate variables for Antarctica.*

P3 l4: Your transition in unclear. You describe changes in trends in AP temperature associated with a cooling for the first part of the 21st century, the WAIS warming amplitude being part of the natural multi-decadal variability of last 2000 years (308 in the Thomas' study, and 2000 in the Steig's one), but also the weak cooling in the EAIS.

I think that you rather give the limits of the comprehension of the very last decades climate variability at the end of the last three paragraphs, ie tackle with the WAIS warming in your paragraph from P2 l29. But please when giving this information, stress that it is the rapidity of this warming which is unprecedented. Then, introducing the need for extended observations and monitoring in Antarctica, just give a short sentence to resume the limits of our comprehension of the recent climatic variability, whatever the considered Antarctic region.

*Answer: As mentioned above we completely restructured the introduction considering all suggestions made by Rev #1. We stressed that the rapidity of the warming for both, the AP and the WAIS, is unprecedented and not the warming itself. We wrote a summary paragraph about the limits of the comprehension of the last decade's climate variability in all three regions (see above). We have the impression that the manuscript gained in clearness due to the proposed restructuring of the introduction. It also became slightly longer, which contrasts with Rev #2, who proposed to condense the introduction.*

P3 l11: There is a lack of data not only for the interior of Antarctica, but also for coastal areas, for instance in the Indian Ocean coastal region and in Adelie Land. You can refer to Jones et al. (2016) for the temperature, and the updated water stable isotope Antarctic database (Goursaud et al., 2018b).

*Answer: We decided to change the text to "for most of Antarctica" instead of referring the the Antarctic interior only.*

P3 l3: From "For the region", please add a new paragraph, where you focus on this crossing point between the three main regions of Antarctica. In this paragraph, please be more explicit on the questions that you address in this study. I also recommend to report the fundamental literature related to the climate of Ellsworth region.

*Answer: The state of the art of the selected region, i.e. the UG region, at the intersection between WAIS, EAIS and AP is given in the text of the introduction chapter. There is no additional proxy data or meteorological data for the wider study region available and that is why we think that our dataset is unique. The knowledge about the greater region, i.e. the Weddel Sea sector has been added to the respective paragraphs in the introduction. The introduction chapter finalizes with the key research questions that we wish to address in the manuscript.*

P4 l1: change "WAIS in the south" to "southern WAIS".

*Answer: Sentence changed to "Do the UG region and surrounding areas experience the same strong and rapid air temperature and accumulation increases as observed for the neighbouring AP and WAIS and, if yes, to what extent?"*

2 Data and Methodology

2.1 P4 l9: I cannot find neither in the paragraph nor in the Table 1 the ray in which the drilling sites are located. Could you please add it here? I would be also very interested in knowing here which sites are in crests or peaks, as surface mass balance should differ between these sites (Agosta et al., 2018), as well as the isotopic signature, at least at the second order.

*Answer: This is in line with Rev #2, hence we added the information about the site-specific characteristics to the text (Chapter 2.1, p 4).*

p4 l15: Please add a comma after "(2012)".

*Answer: Comma added.*

P4 l19, P4 l25: Please add a space between numbers and units throughout the manuscript.

*Answer: Spaces added.*

P24 l4: How can you quantify that the precision is better than a specific threshold. Please give a precise uncertainty.

*Answer: The precision was determined by calculating the mean standard deviation of all standards used in the measurements of the firn cores. These are the precisions given in the manuscript. "better than" has been deleted.*

P5 l9: I understand that you compute a local meteoric water line based on ice core data, and especially only 6 points, which can be affected by deposition and post-deposition effects.

*Answer: Yes. As stated in the manuscript, there are no precipitation samples from the study site available and thus no data on the stable water isotope composition of recent precipitation exists. Hence, we used the stable water isotope data (seasonal means) of the six firn cores for computing a LMWL (2348 data points), see Figure 5.*

2.2 P5 l13: which ratio did you use to compute nssSO4? Did you use summer or winter ratio (Jourdain and Legrand, 2002)?

*Answer: Values of nssS have been calculated from ssNa-concentrations and S/Na-ratios in seawater, respectively, using the following equation with relative abundances from Bowen (1979):*

$$nssS = S - ssNa*(S/Na)_{seawater}$$

*Values of ssNa were calculated according to Röthlisberger et al. (2002) using relative abundances from Bowen (1979). This information has been added to the new manuscript. Seasonal differences have not been taken into account as our manuscript does not deal with nssS chemistry, but uses this information for dating only.*

*Bowen, H.J.M.: Environmental chemistry of the elements. Academic Press. London. New York. 1979.*

*Röthlisberger, R., Mulvaney, R., Wolff, E.W., Hutterli, M.A., Bigler, M., Sommer, S. and Jouzel, J.: Dust and sea salt variability in central East Antarctica (Dome C) over the last 45 kyrs and its implications for southern high-latitude climate, Geophys. Res. Lett., 29, doi:10.1029/2002GL015186, 2002.*

P5 l15: Please provide references for the seasonality of the aerosol signals preferentially for your region. For instance, in Adelie Land, there is no seasonal cycle in ssNa, so please check. You can also cite recent studies which apply such an annual layer counting to date firn ice cores and discuss the uncertainties (Vega et al., 2016; Caiazzo et al., 2016).

*Answer: In line with the comment above, we use the seasonally changing parameters $H_2O_2$, nssS and stable water isotopes for dating only. Accordingly, the dating sequence is: annual layer counting using $H_2O_2$ (as directly connected to insolation, i.e. to the atmosphere) or nssS (as directly related to marine biogenic activity and eventually sporadic volcanic inputs) from chemical analysis, and stable water isotopes (as temperature proxy). The other chemical parameters mentioned in the manuscript – MSA, Cl/Na, $SO_4^{2-}$, nssS/ssNa - are used for confirmation only and not directly for the dating, i.e. the annual layer counting itself. The dating chapter in the manuscript has been re-written including additional references, in order to make the dating procedure more clear. However, as our study presents first firn core data for the UG region, including data on chemical parameters, no references for the seasonality of aerosol signals in the UG region are available.*

P5 l18: How did you estimate your uncertainty? I would have liked to see on the figures 1 and 2 what constitutes this age scale uncertainty (e.g. peaks that you did not count).

*Answer: As already explained above we estimate that the error associated to ALC is ±1 year for cores dated with glacio–chemistry (SCH-2 and PASO-1) and ± 2 years for the cores that were dated with stable water isotopes only and matched to the chemistry-based age scale of SCH-2 (GUPA-1, DOTT-1, SCH-1 and BAL-1). This information has been included to the text (Chapter 3.2). In Figures 2 and 3 maxima in the individual firn-core stable water isotope and chemistry records that have been included in the ALC fall on dashed lines (i.e. summers). Stable water isotope maxima between two dashed lines have not been included in the ALC (i.e. due to the $H_2O_2$ [SCH-2] or nssS [PASO-1] ALC assumed to be of higher significance; further details see below). This information has added to the Figure captions of Figures 2 and 3.*

P5 l17: I do not fully understand the rationale behind the method used to match GUPA, DOTT, SCH-1 and BAL-1 dating to SCH-2: are the isotope records highly correlated, justifying such a method? (what is the implicit hypothesis and can you test it)?

*Answer: We do not fully understand this commentary. We have two well-dated firn cores (independently dated through chemistry and stable water isotopes, SCH-2 and PASO-1). We use wiggle matching to match the other firn cores dated by ALC of stable water isotopes only to these better chronologies (see answer to previous comments above). For this purpose we used prominent minima and maxima in stable water isotopes (as visualized in Fig. 7 of the manuscript). Furthermore, forALC we counted a minimum and maximum number of stable water isotope peaks in all firn cores which brackens the age range for each firn core (see Table 1 above). SCH-1 and SCH-2 are situated nearby in the same valley at a altitudinal distance of ca. 250 m and are highly correlated in δD and $δ^{18}O$ (see statistical assessment between individual cores for isotopes and accumulation rates in Tables S6 and S7 in the supplements). Similar correlations are observed between SCH-2 with GUPA-1 and DOTT-1 despite their greater distance and different site-specific characteristics. GUPA-1 is being dealt with caution due to its proximity to the landing strip and has been excluded for statistical evaluations in the discussion. Hence the only critical core in this respect is the BAL-1, (firn core in the neighbouring valley at a same altitude than SCH-2) which shows a very smooth isotope record in the lower part. We have nonetheless confidence in this core down to 1991 (see Fig. 7) as prominent minima and maxima are both visible in SCH-2 and BAL-1.*

P5 l24: P5 l23: Why did you use non-standardized data for composites? Please take into consideration the general comments.

*Answer: This is in line with comments of Rev #2 and #3. We have decided to consistently use standardized data throughout the study in order to give each firn core the same statistical weight. This information has been added to the text. However, as $δ^{18}O$ shows a statistically significant positive trend in the non-standardized data, which is not visible in the standardized data anymore, we kept this information to demonstrate the effect of standardization on $δ^{18}O$ trends. As this is not relevant for d excess and accumulation trends, we omitted the information about non-standardized data from the manuscript.*

Why did you use the Mann-Kendall tests? It is usually used to detect inflections. Is it the case here? Please justify.

*Answer: For the study site no inflections are present in the time series. Turner et al. (2016) use a sequential Mann-Kendall test, whereas we use a Mann-Kendall Tau and Sen slope estimator trend test. As in our study, the latter was also used by Turner et al. (2016) to estimate the statistical significance of linear trends. The Mann–Kendall Tau and Sen slope estimator trend test is commonly employed to detect linear and non–linear trends in time series of environmental, climate or hydrological data with its power and significance being independent of the actual distribution of the input data (Hamed, 2008). This means, that the input data, in our*

*case stable water isotope and accumulation data of firn cores, does not necessarily need to be normally distributed and the test is not very sensitive to abrupt breaks caused by inhomogenities in the analysed time series. Hence, we believe that the Mann-Kendall trend test is the appropriate test to use as it is more conservative than simple linear regression.*

*Hamed, K.H.: Trend detection in hydrologic data: The Mann-Kendall trend test under the scaling hypothesis, J. Hydrol., 349, 350-363, doi:10.1016/j.jhydrol.2007.11.009, 2008.*

2.3 P6 l4: please add a line break after "isotopes and accumulation." to split observations from climatic modes.

*Answer: Line break added.*

Results

3.1 P6 l24: what initial point (coordinates and height) did you indicate for back trajectory simulations? Which drilling site does it correspond to?

*Answer: For the re-calculation of backward trajectories – now 5 day- instead of 3 day-trajectories – we used the coordinates of the drill site of firn core SCH-1, that is located at the ice devide between the glaciers Schneider and Schanz (see Fig. 1b in the manuscript) as starting point. This location was selected, as it is less influenced by post-depositional processes as well as shows little snow-drift and snow-redistribution, and is therefore among all available firn-core drill sites the most representative for snow-fall events in the UG region. The information is included in the revised version of the manuscript.*

P6 l29 and P7 l1: change "was" to "is".

*Answer: "was" changed to "is".*

P7 l1: you choose in the manuscript the convention m we a-1 for accumulation, so please change "m/s" to "m s-1". Also please change "was" to "is".

*Answer: "m/s" changed to $ms^{-1}$. "was" changed to "is".*

3.2 P7 l8: "the longest record"

*Answer: "t" added (compare RC3).*

P7 l11: What kind of extrapolation did you apply?

*Answer: A linear extrapolation was applied using the linear depth–age–relationship between the previous two clearly identifiable peaks (years 1987 and 1988). The explanation has been added in the manuscript.*

P7 l11: please replace "furthermore" by another word as repeated from previous sentence. Could you give the uncertainties associated with your dating for each firn core at the end of this paragraph? It is actually results and not methods.

*Answer: "Furthermore" deleted without replacement because it is unnecessary.*

*The revised sentence "The estimated error associated to ALC is ±1 year for cores dated with glacio–chemistry (SCH-2 and PASO-1) and ± 2 years for cores dated with stable water isotopes only (GUPA-1, DOTT-1, SCH-1 and BAL-1)." was moved from chapter 2.3 to chapter 3.2.*

3.3 p7 l16: As you give the dO18 mean range, I do not think it is necessary to also give dD. I would also rather go for mean and standard deviation, which give more information about the variability.

*Answer: Mean range for δD deleted and standard deviations for $\delta^{18}O$ added in the text.*

P7 l21: These values are not so low. If you refer to Figure 6 in Goursaud et al. (2018a), that shows the spatial variability of d, you will notice that d can reach minimum values of 0 to 4 per mille in Ross sea, and Amery sector, but also close in the Ellsworth sector !

*Answer: We argue here that the range of mean d excess values is small among the firn cores as the difference between the firn core with the lowest and the highest mean d excess value is only 2.1 ‰. We do not state that the d excess values in general are low as we also find single d excess values in the cores that are < 0‰ (see minimum d excess values for each core in Table 1).*

P7 l25: Please change into the brackets to "range values from . . . to...".

*Answer: "range" changed to "values range from ... to ..." in the brackets.*

I am not convinced by the method used to estimate the LMWL obtained based on ice core data.

*Answer: We explained in Chapter 2.1 of the old manuscript that unfortunately no fresh precipitation samples are available for the Union Glacier region. The study site is not manned year-round and therefore no continuous monitoring is possible. Hence, the only chance to derive co-isotopic relationships is with our firn-core data. We do that for individual cores (shifted to Supplement S3a-f) and for the composite (compare Figure 5) which yield similar co-isotopic slopes of between 7.94 and 8.24 (composite slope 8.02). For one other coastal site (near Neumayer station), firn cores of similar age yielded similar slopes as the local precipitation samples (Fernandoy et al., 2010). We believe that these co-isotopic relationships are, at first approximation, very similar to the LMWL at a given site. We therefore referred to the composite co-isotope relationship as LMWL (see Figure caption for Figure 5) knowing that LMWLs are originally based on monthly precipitation data. For clarification we added the information on individual firn cores' co-isotopic relationships to the supplements (S3a-f).*

*Fernandoy, F., Meyer, H., Oerter, H., Wilhelms, F., Graf, W. and Schwander, J.: Temporal and spatial variation of stable-isotope ratios and accumulation rates in the hinterland of Neumayer station, East Antarctica, J. Glaciol., 56, 673-687, doi: 10.3189/002214310793146296, 2010.*

P7 l27 to 30: there is a low spatial variability of your mean reconstructed SMB, and thus you could just give the mean and standard deviation of the SMB averages, instead of describing the core with the highest and lowest SMB. Details are then given in Table 1.

Then, your next message is substantial as you show that particular high (low) values are not concomitant between the firn cores.

*Answer: The description of cores with highest and lowest mean accumulation rates was replaced by a description of mean values and standard deviations. The sentences "Highest mean accumulation rates (≥ 0.28 m w.eq.a$^{-1}$) were found for DOTT–1and SCH–2, despite the site of SCH–2 being located further inland and at about 750 m higher altitude compared to DOTT–1 (Table 1 and Fig. 1b). Lowest mean accumulation occurs at the GUPA–1 and PASO–1 sites (~0.18 m w.eq.a$^{-1}$)." were replaced by "Mean accumulation rates vary*

*between ~0.18 m w.eq.a⁻¹ (GUPA-1 and PASO-1) and ~0.29 m w.eq.a⁻¹ (SCH-2) with the lowest standard deviations found at the DOTT-1 and PASO-1 sites (~0.05 m w.eq.a⁻¹) and the highest ones exhibited by cores SCH-2 and BAL-1 (~0.08 m w.eq.a⁻¹; Table 1).".*

Discussion 4.1.1 p8 l5 and p8 l8:  I do not see the point to report minimum and maximum values.  What is your message from such information?

***Answer:*** *We omitted the reported minimum and maximum values and shifted this information to the Figure captions of Fig 7.*

You could first discuss the differences in the mean dO18 values from one ice core to another noted in the results, and consistent as your write with continental effect.  You could then discuss the lack of similarities in inter-annual variabilities (remaining results?), and confirming here by testing the correlation between the different dO18 over the overlapping time period 1998-2013.  Here, you could write that 2002 is a maximum value in all firn core data.

***Answer:*** *We started this discussion chapter with the comparison of the stable water isotope composition of the individual firn cores (excluding GUPA-1) as suggested. We based the cross-correlations on the maximum overlapping period between two individual cores (as given in Table S6). We also computed the cross-correlations for the common overlapping period from 1999-2013, which yielded similar results (see Table S7). This information has been added to the manuscript. The information about a common maximum in 2002 has been placed where suggested.*

There is some ambiguity on the structure of the manuscript, where some key results are reported in the discussion, and not in the section on results.

***Answer:*** *Section "At SCH–1, SCH-2 and BAL–1 accumulation decreased at a rate of –0.002 m w.eq.a⁻¹ (p-value = 0.032), –0.004 m w.eq.a⁻¹ (p-value < 0.0001) and –0.003 m w.eq.a⁻¹ (p-value = 0.006), respectively. The decrease is highest at the DOTT–1 site, but not statistically significant (s = –0.005 m w.eq.a⁻¹; p–value = 0.458). In contrast, accumulation exhibits a slight, albeit statistically significant increase at the PASO-1 site (s = +0.001 m w.eq.a⁻¹; p–value = 0)." moved from chapter 4.2.1 to chapter 3.3 and reduced to "At SCH–1, SCH-2 and BAL–1 accumulation decreased at a rate of –0.002 m w.eq.a⁻¹, –0.004 m w.eq.a⁻¹ and –0.003 m w.eq.a⁻¹ (p-values < 0.05), respectively. In contrast, accumulation exhibits a slight, albeit statistically significant increase at the PASO-1 site (s = +0.001 m w.eq.a⁻¹, p–value < 0.05)."*

*Section "From time–series analysis of d excess annual means (S5) statistically significant positive trends have been found for SCH–2 (s = +0.085 ‰ a⁻¹, p–value < 0.0001) and PASO–1 (s = +0.016 ‰ a⁻¹, p–value = 0.002), whereas for DOTT–1 (s = –0.110 ‰ a⁻¹, p–value = 0.015) and SCH–1 (s = –0.052 ‰ a⁻¹, p–value < 0.0001) the d excess trend is negative. The BAL–1 d excess record exhibits no trend." moved from chapter 4.1.1 to chapter 3.3 and changed to "At SCH–1, SCH-2 and BAL–1 accumulation decreased at a rate of –0.002 m w.eq.a⁻¹, –0.004 m w.eq.a⁻¹ and –0.003 m w.eq.a⁻¹ (p-values < 0.05), respectively. In contrast, accumulation exhibits a slight, albeit statistically significant increase at the PASO-1 site (s = +0.001 m w.eq.a⁻¹, p–value < 0.05)."*

P8 l14 to l16: relationships which are not significant, are useless. Please remove it. The two positive trends are results and not discussed here, so it should go to the "results" part.

***Answer:*** *Section "Time–series analysis of δ¹⁸O annual means (S3) reveals positive trends for SCH–1 (s = +0.039 ‰ a⁻¹) and BAL–1 (s = +0.054 ‰ a⁻¹), whereas for DOTT–1 (s = –0.071 ‰ a⁻¹), SCH–2 (s = –0.011 ‰ a⁻¹) and PASO–1 (s = –0.009 ‰ a⁻¹) the δ¹⁸O trend is negative. However, only the positive trends for SCH–1 and BAL–1 are statistically significant (α = 0.05, p–value < α; p–value (SCH–1) = 0.028; p–value (BAL–1) < 0.0001)." moved from chapter 4.1.1 to chapter 3.3 and reduced to "Time–series analysis of δ¹⁸O annual*

*means (S3) reveals statistically significant trends only for cores SCH–1 (s = +0.039 ‰ a$^{-1}$, p-value < 0.05) and BAL–1 (s = +0.054 ‰ a$^{-1}$, p-value < 0.05)."*

P8 l17: I do not find that dO18 firn cores data are well correlated. Some are not correlated at all, eg PASO-1 with DOTT-1 etc. , and the highest r is 0.658, ie r$^2$ of 0.433, so I really would not go for a regional signal by a simple average of the time series. In the assessment of potential trends, the discussion should be explicit that only two of your firn cores present such trends (P8 l21).

*Answer: see above. We believe that correlations are not too bad given all the uncertainties between the individual core sites (potential differences in local precipitation seasonality, redistribution due to wind drift etc.). We know that the data is neither spatially nor temporally equally distributed. However, we would not like to base our interpretation on a single firn core (where we always would have to deal with the spatial representability of this single core). We discuss the differences between individual cores. For instance, five out of six individual cores yield similar accumulation trends (except PASO-1). Moreover, all six cores show no clear trend in the δ-values (either very low positive or negative), which allows for a general assumption, which would not be possible with one single core only. We interpret the individual cores first and then find similar patterns in the stack. We believe that this substantiates our approach to derive a stack to be more representative for the region.*

P8 l22: Why did you prefer Sen slope rather than linear simulations?

*Answer: The Sen slope estimator allows to determine the magnitude of the trend as the median of the slopes of all lines through pairs of two–dimensional sample points. It is more robust than the least–square estimator due to its lesser sensitivity to outliers (Theil, 1950; Sen, 1968). Hence, we preferred to take the Sen slope estimator because it is the more conservative approach.*
*Sen, P.K.: Estimates of the Regression Coefficient Based on Kendall's Tau. J.Am.Stat.Assoc.. 63. 1379-1389. 1968.*

*Theil, H.: A Rank-Invariant Method of Linear and Polynomial Regression Analysis. Proceedings of the Royal Netherlands Academy of Sciences. 53. 386-392. 1950*

P8 l23: You cannot conclude an increase in near-surface temperature from a positive $\delta^{18}O$ trend (which is I think not robust, see what I wrote before), whereas you did not test the multi-year dO18-T relationship in your region.

P8 l23 to l32: You discuss here the inter-annual variability in temperature. It should be in another paragraph dedicated to it, and using the results of dO18-temperature relationship you find in your data, and that should be cited in the results.

*Answer: We restructured the related paragraphs and excluded any inference towards temperature change at UG based on firn-core stable water isotope data. Instead we put the UG firn-core stable water isotope data (not changing much) in the context of large-scale climate (also not changing much in the past 30 years except for WAIS).*

P8 l33 – P9 l6: Is it Sen slopes of from linear regressions? If linear, please give the correlation coefficients for each simulated linear relationship. I am very surprised for such different trends. Either the relationships are very weak or trends can be neglected, or other effects than change in moisture origin might act here to explain the differences, as your drilling sites are relatively close to each other. Once more, I do not find it robust at all to make a mean for time series showing opposite trends, and where only two firn cores are correlated (DOTT-1 and SCH1).

*Answer: Yes, Sen slopes have been used consistently throughout the manuscript. We agree that the time series do not show consistent trends in d excess as these relationships are weak. We reduced the discussion but left the information on the d excess (both individual and stack) for completeness. Even though the drilling sites are within 50 km horizontal distance (not too close actually), they are separated by orographic barriers, which might explain the dissimilarities in d excess/moistures sources. This information has been added to the text (Chapter 4.2.1, p 10).*

4.1/2 This paragraph should come before discussing a potential temperature reconstruction.

*Answer: We restructured both chapters and start with the spatial and temporal variability of stable water isotopes (Chapter 4.2.1 in the new manuscript), which are then related to near-surface air temperature changes (both instrumental and ERA-Interim data; Chapter 4.2.2 in the new manuscript). We conclude that the isotope-temperature relationship is weak and give reasons for this. A part of the $\delta^{18}O$ variability is explained by near-surface air temperature though.*

P9 l12: You could compare near-surface temperature from the wx7 and Arigony AWS with ERA-interim. If the correlation is strong enough, you could test the linear relationship between dO18 and the near-surface temperature over longer period than for observations, thus using ERA-interim temperature. You could also test the relationship with each of your firn core. Have you considered that local processes could affect the signal, and could be more important at some locations?

*Answer: This is in line with a comment of Rev #2 and Rev #3 and has been addressed in detail in the replies to the review of Rev #2. Please see below.*

P9 l15-16: You cannot use dO18 to reconstruct the temperature: the linear relationship is much too weak (l10), and this sentence is not consistent with l11: "However, a proper inference of near-surface temperature from dO18 values of precipitation in the UG region is not yet possible".

*Answer: We are grateful for this remark, which helped improving our argumentation further. As introduced above, we restructured the chapter. We conclude that the isotope-temperature relationship is weak and give reasons for this. A part of the $\delta^{18}O$ variability is explained by near surface air temperature though. This was changed consistently throughout the text.*

4.1.3 This part is very interesting and show the potential of the isotopic signal to provide information about the Weddell Sea ice extent. It would be valuable to describe these results with due care, as they are based on composite analyses from individual series that are weakly correlated, challenging the confidence in a strong common climate signal. What is the likelihood of obtaining a link with sea ice extent using pure noise with a given frequency range?

*Answer: We appreciate that Rev #1 recognizes the value of the sea ice extent – isotope connection. As introduced above, we believe that the correlations are not too bad for six firn cores retrieved from distinctly different sampling locations each with individual site-specific characteristics. Moreover, we tested the signal-to-noise ratios for the single stable water isotope records (see discussion above), which revealed quite high values. Therefore, we believe that the generation of an isotope stack makes sense and a signal of regional significance can be extracted and then be correlated with the sea-ice extent.*

p10 l5: why do you suggest this correlation to be an artefact? It is very weak (r = 0.315, ie r2= 0.0992!).

*Answer: True. We omitted the word "artefact" in this context.*

4.2.1 p10 l22: you do not need to give the precise value of p-values. Just write p<0.001 or p<0.05. Change throughout the manuscript.

*Answer: p-values changed throughout the manuscript.*

P10 l24: Remove the insignificant relationship for firn core DOTT-1.

*Answer: Removed the sentence "The decrease is highest at the DOTT–1 site, but not statistically significant (s = –0.005 m w.eq.a⁻¹; p–value = 0.458).".*

P10 l22-26: Deposition effects related to wind should be mentioned at the beginning of your study, as it can significantly modify your signal.

*Answer: Done. We now refer to the effects related to wind already in the results, e.g. we added the following sentence to Chapter 3.1 (p 7): "... and imply the possibility of substantial redistribution of snow due to wind drift.*

P11 l2: Precise how far are the stake measurements from the firn cores, and the grid resolution of the model. What are the covered periods?

*Answer: The required information has been added to the manuscript.*

P11 l3: Can you also precise the location of Patriot Hills and the shortest distance between the closest ITASE ice core and your firn cores.

*Answer: The required information has been added to the manuscript.*

P11 l6: Could you compare the inter-annual from your firn cores with the stake measurements and model outputs? We have shown in our paper under review, the added value of such time series 1) to make the dating more robust, and 2) extract a regional signal.

*Answer: The data from the stake measurement and the background data from the model runs from van den Broeke et al. (2006) are unfortunately not publicly available.*

4.2.2 P11 l22: you suggest that ERA-interim fails to capture the effects of orography. You can show it by giving the surface height of the grid covering the drilling sites, while these ones differs from each other.

*Answer: We are very thankful for this comment. As suggested we have extracted the elevation of the nearest ERA-Interim grid point for each firn core and now provide this information in Table 1. The elevations are [m asl.]:*
*DOTT-1: 641.5*
*GUPA-1: 911.3*
*SCH-1 and SCH-2: 1061.2*
*PAS-1 and BAL-1: 1208.4*

*The differences between the actual altitudes of the firn core drill sites and the elevations of the respective ERA-Interim grid points are quite significant (about 100 to 700 m) and therefore might provide evidence for the ERA-Interim model not capturing the local orography of the Ellsworth Mountains. This information has been added to the manuscript (Chapter 4.2.2).*

You suggest also at the very end that your data could be affected by post-deposition effects. But couldn't you test it, for instance by looking at the evolution of the seasonal amplitude along the cores (see again Goursaud et al., 2018b), or applying a proper diffusion model as done in Jones et al., J of Geoph. Research., 2017.

*Answer: This comment has been answered above.*

Conclusions

P12 l10: Remove the slope for non-standardized data.

*Answer: Slope for non-standardized data removed. .*

Can you make the data available, either in supplementary material, on any public access depositary?

*Answer: We added a link to public repository Pangaea, where the data will be uploaded after acceptance of the manuscript.*

Figures and tables
Figure 1b: Names of the drilling sites are hardly readable (except GUPA-1 and DOTT-1).

*Answer: The colouring and the font size of the names of the drilling sites have been changed to make them more readable.*

Figure 2: What does "smooth 2p" correspond to? It is not specified in the legend or caption

*Answer: Information added to figure captions. It is a 2 point running average.*

Specify that depth is in water equivalent in unit (as well as for the following figures).

*Answer: The depth is given in meters below surface not in water equivalent.*

Figure 3: Why did you use nssSO4/ssNa? What does it correspond to? This is not explained in the paper.

*Answer: We used the ratio nssS/ssNa for detecting signals of volcanic eruptions in order to use them as tie points for the dating of PASO-1. nssS can be either of marine biogenic or of volcanic origin, of which the former exhibits a seasonal cycle (phytoplankton). ssNa has only one seasonally changing source (sea salt). Hence, the ratio of nssS/ssNa gives clearer evidence of exceptionally high peaks of nssS that might point towards volcanic eruptions that superimpose the seasonal signal of marine biogenic activity. Furthermore, the questions of Rev #1 are in line with a comment of Rev #2 and have been also answered there.*

How can you justify that you did not count peaks ∼ 1.2 and ∼5.2 m w.e.?

*Answer: The small peak at 1.2 m is not found in the chemical parameters and has therefore not been counted as annual peak (location between two dashed lines; see answer to previous comment above). The large peak at 5.2 m has been counted (indicated by the dashed line). However, in general the maxima of the chemical parameters and the respective maximum of $\delta^{18}O$ and $\delta D$, respectively, do not necessarily coincide due to the different seasonality of the proxies. This information has been added to the figure captions of Figures 2 and 3.*

Figure 4: Monthly mean from Arigony AWS is hardly readable.

*Answer: The legend of the figure has been enlarged and the colouring has been changed to make the Arigony AWS data more readable.*

Figure 5: I suggest to move these figures to supplementary material.

*Answer: We decided to leave the composite co-isotopic plot in the manuscript as it displays the Local Meteoric Water Line of the UG region. The co-isotopic plots of the individual firn cores were moved to the supplements as suggested (S3a-f).*

Figure 6: for all figures displaying annual-scale data, can you draw it with cityscape vectors?

*Answer: The figure is in line with the style of accumulation time series presented by Burgener et al. 2013 and to our understanding a question of personal preferences or gusto. We decided to leave the figure as it is.*

Once more, slope is almost equal 0. Thus, the discussion of negative trends is not a robust finding.

*Answer: The accumulation rates as deduced from Figure 6 are given in m w.eq. per year. This implies that a very low number for the Sen slope (e.g. -0.004 for SCH-2) corresponds to 4 mm w.eq. less accumulation per year and is thus in the same range as slopes given by Burgener et al. (2013) for central West Antarctica. Hence, we believe that the observed trends in accumulation rates at the different firn-core drill sites should still be reported in the manuscript.*

*Burgener, L., Rupper, S., Koenig, L., Forster, R., Christensen, W.F., Williams, J., Koutnik, M., Miège, C., Steig, E.J., Tingey, D., Keeler, D. and Riley, L.: An observed negative trend in West Antarctic accumulation rates from 1975 to 2010: Evidence from new observed and simulated records, J. Geophys. Res.–Atmos., 118, 4205–4216, doi:10.1002/jgrd.50362, 2013.*

Figure 7: Please remove the composite based on non-standardized data or standardised data, and use cityscape vectors.

*Answer: We removed the composite records for non-standardized data for accumulation and d excess, but kept the one for $\delta^{18}O$ as we think, that it is worth mentioning that it reveals a statistically significant positive trend. This has also been stated in the manuscript. Concenring the use of cityscape vectors see answer above (Figure 6).*

Table 2: Periods for reconstruction are given it Table 1. Please move this table to supplementary material.

*Answer: The total periods covered by each firn core are different than the periods given for accumulation rates as the latter are annual means and therefore refer to periods with only full years. However, Table 2 has been omitted and the respective information has been included into Table 1.*

References

Agosta, C., Amory, C., Kittel, C., Orsi, A., Favier, V., Gallée, H., van den Broeke, M. R., Lenaerts, J. T. M., van Wessem, J. M., and Fettweis, X.: Estimation of the Antarctic surface mass balance using MAR (1979-2015) and identification of domi- nant processes, The Cryosphere Discuss., 2018, 1-22, 10.5194/tc-2018-76, 2018.

Caiazzo, L., Becagli, S., Frosini, D., Giardi, F., Severi, M., Traversi, R., and Udisti, R.: Spatial and temporal variability of snow chemical composition and accumulation rate at Talos Dome site (East Antarctica), Science of the Total Environment, 550, 418-430,
2016.

Schotterer U., Stichler W., Ginot P. (2004): The Influence of Post-Depositional Effects on Ice Core Studies: Examples From the Alps, Andes, and Altai. In: DeWayne Cecil L., Green J.R., Thompson L.G. (eds) Earth Paleoenvironments: Records Preserved in Mid- and Low-Latitude Glaciers. Developments in Paleoenvironmental Research, vol 9, pp. 39-59, Springer, Dordrecht.

Goursaud, S., Masson-Delmotte, V., Favier, V., Preunkert, S., Fily, M., Gallée, H., Jourdain, B., Legrand, M., Magand, O., and Minster, B.: A 60-year ice-core record of regional climate from Adélie Land, coastal Antarctica, The Cryosphere, 11, 343-
362, 2017.

Goursaud, S., Masson-Delmotte, V., Favier, V., Orsi, A., and Werner, M.: Water stable isotope spatio-temporal variability in Antarctica in 1960–2013: observa- tions and simulations from the ECHAM5-wiso atmospheric general circulation model, Clim.  Past, 14, 923-946, 10.5194/cp-14-923-2018, 2018a.

Goursaud, S., Masson- Delmotte, V., Favier, V., Preunkert, S., Legrand, M., Minster, B., and Werner, M.: Chal- lenges associated with the climatic interpretation of water stable isotope records from a highly resolved firn core from Adélie Land, coastal Antarctica, The Cryosphere Dis- cuss., 2018, 1-55, 10.5194/tc-2018-121, 2018b.

Johnsen, S. J., Clausen, H. B., Cuffey, K. M., Hoffmann, G., Schwander, J., and Creyts, T.: Diffusion of stable isotopes in polar firn and ice: the isotope effect in firn diffusion, Physics of ice core records, 2000, 121-140.

Jones, J. M., Gille, S. T., Goosse, H., Abram, N. J., Canziani, P. O., Charman, D.  J., Clem, K.  R., Crosta, X., De Lavergne, C., and Eisenman, I.: Assessing recent trends in high-latitude Southern Hemisphere surface climate, Nature Climate Change,
6, 917-926, 2016.

Jones, T., Cuffey, K., White, J., Steig, E., Buizert, C., Markle, B., McConnell, J., and Sigl, M.: Water isotope diffusion in the WAIS Divide ice core during the Holocene and last glacial, Journal of Geophysical Research: Earth Surface, C10122, 290-309, 2017.

Jourdain, B., and Legrand, M.: Year round records of bulk and size segregated aerosol composition and HCl and HNO3 levels in the Dumont d'Urville (coastal Antarctica) atmosphere: Implications for sea salt aerosol fractionation in the winter and summer, Journal of Geophysical Research: Atmospheres, 107, 1-13, 2002.

Stenni, B., Curran, M. A., Abram, N. J., Orsi, A., Goursaud, S., Masson- Delmotte, V., Neukom, R., Goosse, H., Divine, D., and Van Ommen, T.: Antarctic climate variability on regional and continental scales over the last 2000 years, Climate of the Past, 13, 1609-1634, 2017.

Vega, C. P., Schlosser, E., Divine, D. V., Kohler, J., Martma, T., Eichler, A., Schwikowski, M., and Isaksson, E.: Surface mass balance and water stable isotopes derived from firn cores on three ice rises, Fimbul Ice Shelf, Antarctica, The Cryosphere, 10, 2763-2777, 2016.
This manuscript provides a new stable records over the last 35 years of water isotope and snow accumulation from 6 firn cores of the Ellsworth Mountains area and ice rise on  Filchner-Ronne Ice Shelf .  The result of isotope and accumulation records are compared with re-analysis and large-scale modes of climate variability  such as  the Southern Annular Mode (SAM) and the El Niño–Southern Oscillation (ENSO) and sea ice extent.

The water isotope and snow accumulation records are very valuable because are representative of an area with very limited records. While I believe that this manuscript will make an important contribution for the characterisation of climatic history of this area, major comments should be addressed before its publication.

The title referred to Union Glacier is misleading, Ellsworth Mountains probably is more appropriate.

*Answer: We included "Ellsworth Mountains" to the title.*

The firn cores where collected in a complex area of about 400 km$^2$ at the boundary between WAIS plateau (PASO-1), Mountain glacier (BAL-1, SCH-1/2), outlet glacier/blue ice area  (GUPA-1)  and  Ice  rise  on Filchner-Ronne Ice Shelf (DOTT-1).  The  Authors must be describing the site cores from morphological and climatological of point view and taking well in account their location/characteristic during the interpretation of  the data, not only elevation and distance from the open sea determine the snow fall intensity and relative isotope compositions.

*Answer: The description of the coring localities has been modified and several aspects (site-specific characteristics) have been included (Chapter 2.1 of the new manuscript). There is, however, no information on differences in local climatology between the sites available.*

The storms that provide snow precipitation could be "similar" for all 6 cores, but the orographic effect on precipitation and the post depositional effect could be very different, as the records shown.  Significant wind drift occurs at AWS with mean wind speed of 6.9 m-1, this agrees with the extensive presence of blue ice along Union Glacier, in particulate for GUPA-1.  At this site probably the anthropogenic effect is limited respect to wind scouring. The transportation by suspension (drift snow) starts at velocities greater than 5 m s-1 (within 2 m), and blowing snow (snow transportation higher than 2 m) starts at velocities of 7 m s-1.

*Answer: We agree that wind drift is a major factor for redistribution of snow in the whole area, with a mean daily wind speed of ca. 7 m/s (max. 30 m/s), and we are convinced that there are differences between the sites. There are blue ice areas in the UG region, but these have not been sampled (e.g. GUPA-1 is a core in firn without contribution of any blue ice). We are grateful for the hint, but do not see any reason to revise our manuscript in this respect.*

The authors compare the data without a clear analysis of the ratio between signal vs noise and their representativeness at local/regional scale (see ex RUPPER et al., 2015, Eisen et al., 2008)

*Answer: This comment is in line with remarks by Rev #1 and #3. Please find answers in our responses to their reviews on how we addressed the signal-to-noise ratio problem.*

The Authors must be provide firstly evidence of a correlation with the ERA re-analysis with meteo station and/or core records before looking at large scale modes such as SAM or ENSO.

*Answer: This comment is in line with remarks by Rev #1 and #3. We combined ERA-Interim re-analysis and meteorological data. See answer to comment on Section 4.1.2 below.*

The assumption of relationship between snow accumulation/isotope temperature with sea ice extent must be demonstrate in general.

*Answer: The general interplay between snow accumulation, air temperatures and sea ice extent is obvious. The closer the open water, the more moisture may be generated and transported towards the continent and, thus, increase accumulation rates. Higher air temperatures will be responsible for more open water and warm air should transport more moisture than cold air. However, these simplistic considerations may turn out to be much more complex in specific settings, and may be obliterated by manifold processes such as wind drift, wind directions, moisture content of air masses etc. This general information has been added to the new manuscript (Chapter 2.3).*
*How the stable water isotopes fit into this system of moisture transport, precipitation, and redistribution is the topic of our study. According to a comment by Rev #1, we changed our conclusions regarding the inference of air temperature changes from stable water isotopes and, hence this part of the question has already been answered in the reply to the review of Rev #1.*

The d-excess is correlated to the moisture source region (sea ice) and distillation effect along the trajectory, instead the oxygen and deuterium rate are strictly correlated to snow precipitation temperature, their seasonality and frequency.

*Answer: We agree, but do not see the need for changes in the manuscript in this respect.*
The Authors compares the result mainly with the coastal part of WAIS (Amundsen and Bellingshausen Sea) and AP, with very small attention to the closer Filchner-Ronne Ice Shelf (Berkner island ex.), WAIS inner site (Kaspari, et al., Burgener et al.) and DML (Coats Land) with analogous moisture source area Weddell Sea.

*Answer: Thanks for this comment. In general, there is very little data for these regions avaiable. We have incorporated this information in the discussion (in particular Chapters 4.2.1 and 4.3.1 of the new manuscript), where we compare our data to available evidence from these regions, including the WAIS (Kaspari et al., Burgener et al.), the Ronne-Filchner Ice Shelf (Graf et al.), Berkner Island (Mulvaney et al.), Coats Land and DML (Philippe et al., Medley et al.). In line with comments of Rev #1, we changed our introduction and compared the UG region with WAIS, AP and EAIS. The reference to Burgener et al. has been added to the introduction and the discussion (Chapter 4.3.1).*

The area is not a "coastal area", open sea is around 1000 km far from Weddell Sea

*Answer: This is true. The only time we used the term "coastal" is for core DOTT-1 to indicate that it is the firn core closest to the sea. We changed this to "closest to the sea" in the text.*

Detail: Introduction too long, without a clear finalization of the paper.

*Answer: This contradicts comments by Rev #1 who requested a more detailed introduction regarding the position of the study site at the intersection between EAIS, WAIS and AP (which needs more information on the main differences between these three Antarctic sectors). However, as both reviewers recommended a clearer defined scientific target, we restructured the introduction, which closes now with a section on the main scientific objectives of our study.*

2.3 dating,

NO clear evidence of Pinatubo nssS signal in SCH-2 and PASO-1, value similar to other annual peaks (SCH-2) or much lower (PASO-1) .

*Answer: In the manuscript we only state that we have found a signal of Mt. Pinatubo in PASO-1. For the dating of SCH-2 Mt. Pinatubo could not be used as a tie point because we did not find it in the $SO_4^{2-}$ record. Identification of Mt. Pinatubo in the PASO-1 record took place via the comparison of the nssS- and nssS/ssNa-records (Fig. 3). nssS is primarily of marine biogenic origin (phytoplankton) exhibiting a clear seasonality with maxima occurring in late summer/early fall and minima occurring in winter. However, during volcanic eruptions nssS is excessively emitted into the atmosphere superimposing the seasonal cycle since atmospheric nssS-concentrations potentially stay at high levels throughout the year. Hence, regarding PASO-1 the coincidence of an above-average peak in the nssS/ssNa-record with a lacking winter minimum in the nssS-record at 9.5-9.7 m depth provide evidence for the presence of a signal of the Mt. Pinatubo eruption in the core. ALC of the respective nssS- and nssS/ssNa-peaks starting at 9.7 m depth (1991) towards the top of the core (2015) supports this hypothesis. Similar changes in wintertime nssS were observed in the well-dated PIG2010, DIV2010, and THW2010 cores from West Antarctica (Pasteris et al., 2014).*

*For clarification, the following information has been added to the manuscript:*
"*For the dating of PASO–1 the signal of the Mt. Pinatubo eruption (1991) could be found via the coincidence of an above–average peak in the nssS/ssNa (sea salt Na)–record with a lacking winter minimum in the nssS–record at 9.5–9.7 m depth (Fig. 3). Similar changes in wintertime nssS were observed in the well–dated ice cores PIG2010, DIV2010 and THW2010 from West Antarctica (Pasteris et al., 2014).*
*Hence, the signal of the Mt. Pinatubo eruption was used as additional tie point for the age model construction of PASO-1.*"

*Pasteris, D.R., McConnell, J.R., Das, S.B., Criscitiello, A.S., Evans, M.J., Maselli, O.J., Sigl, M. and Layman, L.: Seasonally resolved ice core records from West Antarctica indicate a sea ice source of sea–salt aerosol and a biomass burning source of ammonium. J. Geophys. Res. Atmos., 119, 9168–9182, doi:10.1002/2013JD020720, 2014.*

How has been composed and which is the grade of confidence of the time series at annual scale if the error associated to ALC vary from 1 yr (2 cores) to 2 years (4 cores) and without taking in account the ratio of signal/noise due to sastrugi? See Noise vs signal, 1985 snow accumulation at SCH-2 vs PASO-1 pag 7

*Answer: We are certain that wind drift and redistribution of snow play an important role in the UG region. Generally, the only parameters helpful for assessing wind drift are the wind speed (i.e. > 5 m/s) and wind direction from the AWS. Unfortunately, there is AWS data only for one firn-core drill site available (GUPA-1) covering the period 2010-2018, only. Hence, there is no way to retrieve reliable information from the different firn-core drill sites, e.g. on wind taking up snow at PASO-1 and redistribution of snow by the predominantly south-westerly winds towards the lower altitudinal core sites in 1985 (SCH-2). Such processes are possible, but not supported by any data and would lead us to speculations which we tried to avoid.*
*Our error estimate of ±1 (chemistry-dated plus isotope-based ALC) to ±2 years (isotope-based ALC only) might have an impact on correlations and their statistical significance (as it was obvious from comparison with an earlier age model), but not on the overall trends. The signal-to-noise discussion has also been requested by Rev #1 and #3 and has been inserted to a newly included discussion chapter (Chapter 4.1).*

Which is the cross correlation between the different cores for isotope and accumulation?

*Answer: Cross-correlations are given in the supplements S6 (maximum overlapping period between two individual firn cores) and S7 (common overlapping period, 1999-2013).*

2.4 Which is the difference between the two AWS station? show both data in figure 4

*Answer: Actually we display the data of both AWS in Figure 4 including the overlapping period, but the differences between the two are simply not visible as the records are nearly identical. No change needed here.*

Which threshold of snow precipitation is used from ERA Interim for HYSLPIT? Why the analysis is performed only 4 years from 2010 to 2014?

*Answer: The d excess records (individual firn cores and UG d excess-stack) show no significant trends or changes at least during the last four decades. This means that most likely the moisture sources and transport pathways for the UG region have not been subject to significant shifts during that period, which would require or justify HYSPLIT backward trayectory calculations further back in time. Thus, we believe that the HYSPLIT calculations have the highest reliability for the period where we have meteorological data overlapping with firn-core data (2010-2015). Moreover, we use this data to characterize the site rather than interpreting temporal differences. We used a minimum threshold corresponding to 1% of the annual accumulation (i.e. ~2.5 mm d$^{-1}$) for the calculations, a quantity which is percentually equivalent to the one used by Thomas & Bracegirdle (2009). This information has also been added to the manuscript.*

*Thomas, E. R. and Bracegirdle, T.J.: Improving ice core interpretation using in situ and reanalysis data, J. Geophys. Res., 114, D20116, doi:10.1029/2009JD012263, 2009.*

3.3 SCH-2/1 and BAL-1 are within 10 km and show similar accumulation, the comments about higher and lower accumulation should be addressed for these sites also at annual scale or better a pluriannual (eg. 3 years), to see the ratio signal/noise.

*Answer: We have calculated the signal-to-noise ratios based on $\delta^{18}O$ time series for all six cores (0.66) and for five cores excluding GUPA-1 (0.60). When referring to the overlapping period (1999-2013) these are 0.72 for all six cores and 0.78 for five cores excluding GUPA-1. In line with the comments of the other reviewers we included this information to the text. We estimate the signal-to-noise ratio in the UG region as quite high (i.e. comparable to signal-to-noise ratios at WAIS). The signal-to-noise ratio is similar when referring to the three neighbouring cores only (BAL-1, SCH-1 and SCH-2) and yields 0.60 for the entire cores and 0.86 for the overlapping period (1999-2013), even though BAL-1 shows relatively low correlation with all other cores (excluding SCH-1; r=0.44).*
*We have also calculated the signal-to-noise ratios based on accumulation time series for all six cores (0.23) and for five cores excluding GUPA-1 (0.29). Compared to the signal-to-noise ratios for stable water isotopes they are very low, probably reflecting the strong influence of the site-specific characteristics on accumulation rates (e.g. the different exposure to wind drift). When referring to the overlapping period (1999-2013) the signal-to-noise ratios do not improve: They are 0.10 for all six cores and 0.32 for five cores excluding GUPA-1. However, the signal-to-noise ratio is significantly higher when only referring to the three neighbouring cores BAL-1, SCH-1 and SCH-2 or even just to SCH-1 and SCH-2, which were retrieved from the same valley, and yields 0.79 and 1.67, respectively when referring to the entire core records. This might be due to the proximity of their drill sites (within 10 km distance) and the similarity between the site-specific characteristics of the three cores, i.e. the location in northwest-southeast oriented, U-shaped glacial valleys, leading to similar accumulation patterns. However, when referring to the overlapping period the signal-to-noise ratio stays at a low value of 0.24 for the three cores, but is still high for SCH-1 and SCH-2 (1.2). In line with the comments of the other reviewers we included all information on signal-to-noise calculations to the text (Chapter 4.1).*

SCH-2 is isotopic "less depleted" than SCH1 with 250 m of difference in elevation and BAL-1 at the same elevation, PASO1 presents a similar isotope mean ith Bal1 with 400 m of difference in elevation. GUPA-1 is "more depleted" than SCH-1/SCH2 with a difference in elevation of 500-800 m. Before any consideration in discussion about the isotope and accumulation some comments must be addressed on these difference and their significant, also in comparison with the other core sites in different geographical position and much far.

*Answer: We agree with Rev #2 that the isotope-altitude relationship is not straightforward. When plotting for all six firn cores mean $\delta^{18}O$ versus altitude plots, we achieve a rather low coefficient of determination of $R^2 = 0.38$ (p-value = 0.191; Fig. 1 below). However, when excluding GUPA-1, which shows very low $\delta^{18}O$ for the height of its coring position, the $\delta^{18}O$-altitude-relationship becomes statistically significant and the coefficient of determination increases substantially and yields: height = -142 \* $\delta^{18}O$ – 3477, $R^2$= 0.81, p-value = 0.036. This corroborates our decision to exclude GUPA-1 from statistical analysis. When referring to the overlapping period (1999-2013), that actually makes the cores more comparable among each other, similar results are obtained (Fig. 2 below). We included the information for the overlapping period to the manuscript.*

[Figure]

*Figure 1: Relation between mean stable water oxygen compostion and altitude for the UG firn cores considering all six cores (blue) and excluding GUPA-1 (orange), respectively. The equation, the coefficient of determination ($R^2$) and the p-value are given for both linear regressions.*

[Figure]

Figure 2: Relation between mean annual stable water oxygen compostion and altitude for the UG firn cores considering all six cores (blue) and excluding GUPA-1 (orange), respectively, referring to the overlapping period (1999-2013). The equation, the coefficient of determination ($R^2$) and the p-value are given for both linear regressions.

4.1.2 Line17-21, Is ERA-Interim or the staked records that are not able to capture the Climate of Ellsworth Mountain? ERA-interim should be firstly compared with AWS data and than with firn records. Isotope and snow accumulation represent the snow fall events plus the noise due to post-deposition process, the absence of correlation with ERA-Interim must be better analysed also in comparison with AWS.

*Answer: This is in line with comments of Rev #1 and #3. We did this exercise and compared the available meteorological data from the two AWS (near-surface air temperature) with the ERA-Interim data for the period February 2010 - November 2015 (overlapping period between AWS record and firn cores). We used ERA–Interim near–surface air temperatures extracted for the GUPA–1 drill site as this is the firn core site closest to the two AWS. Monthly mean air temperatures from both datasets are highly and statistically significantly correlated ($R^2 = 0.99$, p-value = 0; Fig. 3 below) and show the same variability. This information has been added to the text (Chapter 4.2.2 in the new manuscript).*
*As the two AWS are placed at a lower altitude (at approx. 700 m asl., near GUPA-1) compared to the nearest grid point of the ERA-Interim model (911 m asl.; see Table 1 in the manuscript and answer to Rev #1 above), the ERA-Interim data must display lower overall monthly mean air temperatures (due to the lapse rate; see Fig. 3: on average about 5°C lower). With respect to the firn core records that do capture a local climate signal obliterated by post-depositional processes the same issue arises. The mean height of all six firn cores is 1289 m and thus almost 600 m higher than the location of the AWS whose near-surface air temperature we correlate with the UG δ¹⁸O-stack (see Chapter 4.2.2 in the new manuscript). Furthermore, the differences between the actual altitudes of the firn-core drill sites and the elevations of the respective nearest ERA–Interim grid points are large, ranging from about 100 m (DOTT–1) up to 700 m (PASO–1) (see Table 1 in the new manuscript). Hence, this might be responsible for the absence of a correlation between UG stable water*

*isotopes and ERA–Interim based near–surface air temperatures as the ERA–Interim model does not capture the local orography of the study region well. This information has been also added to the text (Chapter 4.2.2).*

[Figure]

*Figure 3: Comparison between AWS and ERA-Interim monthly mean air temperatures for UG (GUPA-1 drill site) for the period February 2010 – November 2015.*

4.2.1

PASO-1 presents an accumulation from 37 to 27% less than SCH1/2 and BAL1.

No sense the average value of 0.25 weq a-1 and their comparison with other measurements at hundred km far.

**Answer:** *We agree that the accumulation is different at the different firn core sites, which is obvious as increasing distance from the ocean and increasing altitude induces a decrease in the level of moisture in the air. We first present accumulation rates for all firn core sites (Chapter 3.3 in the new manuscript). Since these vary from 0.18 to 0.29 m w.eq. a⁻¹, we found these values similar enough to draw conclusions on a larger region, i.e. to calculate an average accumulation rate for the UG region. Therefore, we kept this information in the text.*

Line 7-9 These data must be compared with the closer site of inner WAIS, Filchner- Ronne Ice Shelf and DML (Coats Land) instead of AP and Coastal Ellsworth Land with difference moisture source.

**Answer:** *As already answered above, there is generally very little data for these regions avaiable. We compared our data to the nearest data on accumulation rates available to us (Chapter 4.3.1), including UG itself (Rivera et al.), Patriot Hills (Casassa et al.) and the closest ITASE ice-core site (Kaspari et al.). In the new manuscript we also included data on the Ronne-Filchner Ice Shelf (Graf et al.) and Dronning Maud Land (e.g. Schlosser et al., Medley et al.).*
.
This manuscript presents a new dataset of stable water isotopes and accumulation rates from firn cores in the Ellsworth mountains at the northern edge of the West Antarctic Ice Sheet for the period 1980 – 2014. As measurements at the intersection between the Antarctic peninsula, the West Antarctic Ice Sheet and the East Antarctic Ice Sheet are particularly sparse, this new dataset is an important contribution toward a better understanding of climate variability in this region. I see the value of the manuscript therefore primarily in the publication of this dataset, while the accompanying meteorological analysis is limited. This is acceptable, as the dataset itself deserves a publication, but I recommend minor revisions before final publication.

*Answer: We thank Rev #3 for acknowledging the value of the manuscript.*

General comments

The drilling and measuring process is nicely explained, but I miss details on the trajectory analysis. Where did you start the trajectories from (lat/lon/lev), and how many per precipitation event (I hope more than one)? Also three days might not be enough, and the origin of the trajectories does not always reflect where the moisture comes from, because some trajectories could be very dry and not contribute to precipitation at all. I suggest using the moisture source diagnostic from Sodemann et al. (2008), which is also based on backward trajectories, but specifically identifies moisture uptake regions.

*Answer: The backward trajectory analyses starts from the position of firn core SCH-1, which is located at the Schneider-Schanz glaciers divide. The coordinates are given in Table 1 of the manuscript. This information has now been added to the new version of the paper. Regarding the number of days considered, we computed 5-day trayectories for this new version and checked if these are in line with the 3-day trayectories. Both calculations show similar results, but 5-day trajectories provide a better picture of the moisture transport and therefore we will show only these in the new version of the paper. To select precipitation events we use a slightly modified procedure as proposed by Thomas and Bracegirdle (2009). Differently to them we considered 2.5 mm d$^{-1}$ instead of 10 mm d$^{-1}$, in our case representing 1% of the total annual accumulation, which will discard events lower than the actual isotope resolution of the SCH-1 core analysis. We appreciate the hint to use more sophisticated trajectory models which include moisture uptake diagnostics. As this is not the main focus of this first manuscript about UG firn cores, we would like to stick to our approach which sufficiently allows for identification of potential moisture source regions. As the variation in d excess is rather small, we do not expect major shifts in moisture source and uptake regions, which would favor a more sophisticated diagnostics.*

Referring to the lack of correlation between the firn core data set and local ERA-Interim data, you speculate that either the model is unable to capture the orography of the Ellsworth mountains or this is evidence of post-depositional processes at the firn core sites. You could get a better idea which of the two is the case by correlating local ERA-Interim data with the two weather stations near the firn core sites.

Thomas, E. R. and Bracegirdle, T.J.: Improving ice core interpretation using in situ and reanalysis data, J. Geophys. Res., 114, D20116, doi:10.1029/2009JD012263, 2009.

*Answer: This is in line with comments of Rev #1 and #2 and has been answered in the replies to the review of Rev #2. Please see above.*

Looking at Table 3 and the discussion, there are barely any significant correlations between the isotope signal / accumulation rates and the other variables, meaning that there must be other factors influencing their variability. It would be nice to have some discussion on what these factors could be.

*Answer: We are grateful for this comment as this helps sharpening our interpretation and drawing substantial conclusions. As requested by Rev #1 and #2, we assessed the signal-to-noise ratios of our stable water isotope and accumulation data (compare answer to Rev #1 and #2). We estimate the signal-to-noise ratios for stable water isotopes in the UG region (of between 0.6 and 0.86) as quite high, i.e. higher than signal-to-noise ratios at WAIS, and much better than for Dronning Maud Land cores on interannual timescales (Münch and Laepple, 2018). The signal-to-noise ratios calculated for the accumulation records (ranging between 0.10 and 1.7) most likely reflect the strong influence of the site-specific characteristics on accumulation rates (e.g. the different exposure to wind drift), as they are very low when considering all cores but significantly increase when referring to cores from close-by and similar drill sites (BAL-1, SCH-1 and SCH-2). This has already been explained above.*
*The UG firn cores are characterized by local differences (plateau vs. valley position, altitude, slope, …). Hence, we tried to generate a regionally representative picture by combining the firn cores to a stacked record (minimum 3 cores/year). Nonetheless, the correlations with well known large-scale climate drivers such as SAM, ENSO, SIE are low. Therefore, we state in the revised version of the manuscript that either our stack is not regionally representative or not visibly linked with these drivers.*

*Münch, T. and Laepple, T.: What climate signal is contained in decadal- to centennial-scale isotope variations from Antarctic ice cores?, Clim. Past, 14, 2053-2070, doi: 10.5194/cp-14-2053-2018, 2018.*

Specific comments

Page 5, line 23: Please explain the terms standardized and non-standardized, and why you use both (why not only standardized).

*Answer: In line with comments of Rev#1 and #2, we have decided to consistently use standardized data throughout the text in order to give each firn core the same statistical weight. This information has been added to the text. However, as $\delta^{18}O$ shows a statistically significant positive trend in the non-standardized data, which is not visible in the standardized data anymore, we kept this information to demonstrate the effect of standardization on $\delta^{18}O$ trends. As this is not relevant for d excess and accumulation trends, we omitted the information about non-standardized data from the manuscript.*
*Standardization has been done using the following equation (not included in the new manuscript):*

$$z = \frac{x - \mu}{\sigma}$$

*where x is the raw value, z the standardized value, $\mu$ the mean value and $\sigma$ the standard deviation of the respective firn core record.*

Page 5, line 28: Introduce abbreviation AWS

*Answer: Abbreviation introduced.*

Page 7, line 8: "t" missing in "the".

*Answer: Missing "t" added.*

Page 7, line 23: Could you explain what you mean by that? Meteoric as opposed to what?

*Answer: We omitted the term meteoric in the sentence. We wanted to say that the original (oceanic) moisture source signal is preserved.*

Page 9, line 29: Fig. 9b instead of 9c.

*Answer: 9c changed to 9b.*

Page 9, line 32: "a significant correlation" instead of "a correlation".

*Answer: "a correlation" changed to "a significant correlation".*

Page 11, line 20: Fig. 9c instead of 9b.

*Answer: Fig. 9b changed to Fig. 9c.*

Page 11, line 21: "Similar results have been found" instead of "Similar has been found".

*Answer: "Similar has been found" changed to "Similar results have been found".*

Page 12, line 14: Delete "are".

*Answer: "are" deleted.*

Fig. 5: For completeness, please explain what the red dashed line and the dots show.

*Answer: The red dashed line gives the correlation and the dots the individual measured samples. This information has been added to the figure captions.*

Fig. 6 and 7: Since the timeseries all have the same units, it might be possible to show them all in one plot (with one y axis) using different colors for the different sites. In this way it would be easier to compare them.

*Answer: Especially Fig. 6 is quite busy already. This is the reason why we do not display all individual cores in one plot as the differences would not be visible anymore. As we decided to do so for Fig. 6, we would – for consistency reasons - also leave Fig. 7 unchanged.*

Fig. 10: Please add numbers to the colorbar. Why the irregular spacing?

*Answer: We have replaced the old Figure 10 by two new ones (Figure 10 and 11). In these two figures, the problem with the colorbar and the spacing has been solved differently.*

Table 1: Why only $\delta^{18}O$ for the period covered by all cores?

*Answer: We have used only $\delta^{18}O$ for the period covered by all cores because we used only this proxy for determining the altitudinal gradient from low-altitude, coastal to high-altitude firn cores.*

References

Sodemann, H., C. Schwierz, and H. Wernli (2008), Interannual variability of Greenland winter precipitation sources: Lagrangian moisture diagnostic and North Atlantic Oscillation influence. J. Geophys. Res., 113, D03107, doi:10.1029/2007JD008503

---

## Referee Report (RR1)

**Referee : Sentia Goursaud**

**Summary**
This manuscript is a revised version. It presents a new and first established dataset of firn core records drilled in the Ellsworth region. Water stable isotopes, and chemical analyses allowed an annual layer counting method for the dating. Accumulation rates were infered from the resulting dating, and altogether with the water stable isotope records, meteorological data, and ERA-interim reanalyses and climate modes, were used to extent our current knowledge of the recent climate in this region.
This new version brings substantial improvements compared to the original one, and should definitively be accepted after the authors have considered very minors revisions.

**General comments**
The introduction incredibly gained in clarity. I can now distinguish the different steps of your argument: (i) the challenges of understanding the recent Antarctic climate change, giving the different climate trends, even within a region (EAIS/WAIS/AP), (ii) the need to get more data, and by extension more proxy data, (iii) the added value of your records in an area located in between the 3 main Antarctic regions, and where no data have been measured yet.
However, it is undoubtedly too long. To shorten it, It might not be necessary to give all the pieces of information about the main factors of variability in the different regions of Antarctica.
But, if it is not shorten, I do not think it is a big deal as all messages are very clear, and it is always better to give too much but clear information.
I do like the fact that you explicitly adress the question you tackle in the manuscript. It is usually convenient to introduce the plan by the end of the introduction.

The description of the dating (chap 2.2) is also improved. References of statistical tools used are rigorous.

I thank the authors for the explanations of the nssSO4/nssNa ratio.

The new Section 4.1 about the signal-to-noise ratios is really valuable, and make your analyse much more robust. The accumulation rate ones rose my intention. Such low values may result from deposition processes actually. And you argue in that direction citing M.Frezotti's review about the threshold of suspension and blowing snow. This could be the reason you have nearly no dO18/T relationship. But your interpretation is given with caution.

**Specific comments**
p7 l26 « the highest »
P7 l28 (Chap3.1 « Meteorological data ») Could you give standard deviations associated with means for temperature and wind
p7 l37 Reword « Note that for age-model construction of GUPA-1 two years (1990, 2001) and of BAL-1 three years (1981, 1983 and 1994) … » so it is more understandable. I suggest : « Note that for the age-model construction of GUPA-1 and BAL-1, two and three years respectively (1990, 2001 for GUPA-1, and 1981, 1983 and 1994 for BAL-1) … »
p7 l40 replace « lower » by « lowest »
p8 l21 replace « of > » by « higher than »
p8 l22 add a coma after « year »
p8 l23 add a coma after « BAL-1 »
p8 l32 remove « of »
p8 l42 add a dot after « detail »

p9 l12 Replace « we have calculated the signal-to noise ratio of δ 18 O for the UG firn cores to 0.60 for the entire record period (1973-2014), and to 0.78 » by « we obtained  δ 18 O  signal-to noise ratio of  0.60 for the entire record period (1973-2014), and 0.78 … »

p9 l16 remove « to »

p9 l35 add a dot after « drift »

p10 l1 « the lowest »

p10 l20 If the relationship is not significant, then there is no need to specify the slope and the p-value. Also you could simplify the reading of the p-value, writing « p<0.05 » when it is significant whereas to give the value, which actually bring no additional information.

P10 l24 remove « of »

p10 l27 add a coma after « region »

p12 l14 add « with » after « corroborating »

p12 l22 after the bracket, add « , the »

Sentences generally might be very long, and coma missing.

---

## Author Response (AR2)

**Referee: Sentia Goursaud**

**Summary**
This manuscript is a revised version. It presents a new and first established dataset of firn core records drilled in the Ellsworth region. Water stable isotopes, and chemical analyses allowed an annual layer counting method for the dating. Accumulation rates were infered from the resulting dating, and altogether with the water stable isotope records, meteorological data, and ERA-interim reanalyses and climate modes, were used to extent our current knowledge of the recent climate in this region.
This new version brings substantial improvements compared to the original one, and should definitively be accepted after the authors have considered very minors revisions.

*Answer: We thank the reviewer for this assessment of the revised version of the manuscript.*

**General comments**
The introduction incredibly gained in clarity. I can now distinguish the different steps of your argument: (i) the challenges of understanding the recent Antarctic climate change, giving the different climate trends, even within a region (EAIS/WAIS/AP), (ii) the need to get more data, and by extension more proxy data, (iii) the added value of your records in an area located in between the 3 main Antarctic regions, and where no data have been measured yet.
However, it is undoubtedly too long. To shorten it, It might not be necessary to give all the pieces of information about the main factors of variability in the different regions of Antarctica.
But, if it is not shorten, I do not think it is a big deal as all messages are very clear, and it is always better to give too much but clear information.
I do like the fact that you explicitly adress the question you tackle in the manuscript. It is usually convenient to introduce the plan by the end of the introduction.

*Answer: We thank the reviewer for appreciating the efforts regarding the new version of the introduction. We follow the argument that the introduction is quite long. However, this is based on recommendations of the first review of Sentia Goursaud to place the research area into the intersection region of the WAIS, the EAIS and the AP, which substantially helped gaining clarity in the introductory part. We now shortened the paragraph dealing with the forcing factors, specifically the paragraph about the SAM, and hope to have fulfilled the recommendations of this second review.*

The description of the dating (chap 2.2) is also improved. References of statistical tools used are rigorous.

I thank the authors for the explanations of the nssSO4/nssNa ratio.

*Answer: No changes needed.*

The new Section 4.1 about the signal-to-noise ratios is really valuable, and make your analyse much more robust. The accumulation rate ones rose my intention. Such low values may result from deposition processes actually. And you argue in that direction citing M.Frezotti's review about the threshold of suspension and blowing snow. This could be the reason you have nearly no dO18/T relationship. But your interpretation is given with caution.

*Answer: We appreciate that our interpretations have been given with the necessary caution. No changes in the text are needed.*

**Specific comments**

p7 l26 « the highest »

*Answer: « The » has been added.*

P7 l28 (Chap3.1 « Meteorological data ») Could you give standard deviations associated with means for temperature and wind

*Answer: Standad deviations have been added for all means of temperature and wind speed.*

p7 l37 Reword « Note that for age-model construction of GUPA-1 two years (1990, 2001) and of BAL-1 three years (1981, 1983 and 1994) … » so it is more understandable. I suggest : « Note that for the age-model construction of GUPA-1 and BAL-1, two and three years respectively (1990, 2001 for GUPA-1, and 1981, 1983 and 1994 for BAL-1) … »

*Answer: The sentence has been reworded as suggested.*

p7 l40 replace « lower » by « lowest »

*Answer: « lower » replaced by « lowest »*

p8 l21 replace « of > » by « higher than »

*Answer : « of ≥ » replaced by « higher than ».*

p8 l22 add a coma after « year »

*Answer: Comma has been added.*

p8 l23 add a coma after « BAL-1 »

*Answer: The sentence has been reworded and we think that a comma is not needed here.*

p8 l32 remove « of »

*Answer: « of » has been removed.*

p8 l42 add a dot after « detail »

*Answer: Dot has been added.*

p9 l12 Replace « we have calculated the signal-to noise ratio of δ 18 O for the UG firn cores to 0.60 for the entire record period (1973-2014), and to 0.78 » by « we obtained δ 18 O  signal-to noise ratio of 0.60 for the entire record period (1973-2014), and 0.78 … »

*Answer: The sentence has been reworded to « we obtained for the UG firn cores a  $\delta^{18}O$ signal-to noise ratio of 0.60 for the entire record period (1973-2014), and 0.78 when referring to the overlapping period (1999-2013) ».*

p9 l16 remove « to »

*Answer: « to » has been removed.*

p9 l35 add a dot after « drift »

*Answer : Dot has been added.*

p10 l1 « the lowest »

*Answer: « the » has been added.*

p10 l20 If the relationship is not significant, then there is no need to specify the slope and the p- value. Also you could simplify the reading of the p-value, writing « p<0.05 » when it is significant whereas to give the value, which actually bring no additional information.

*Answer: The slope and p-value of the non-significant relationship have been removed. The p-values have been simplified throughout the manuscript. In order to still make different levels of significance clear we use the expressions $p < 0.1$, $p < 0.05$, $p < 0.01$ and $p < 0.0001$, respectively.*

P10 l24 remove « of »

*Answer: « the » has been removed.*

p10 l27 add a coma after « region »

*Answer: Comma has been added.*

p12 l14 add « with » after « corroborating »

*Answer: We disagree. We could either say « corresponding with » or « corroborating ». Nothing has been changed in the text.*

p12 l22 after the bracket, add « , the »

*Answer: « , the » has been added.*

Sentences generally might be very long, and coma missing.

*Answer: This has been changed for all examples mentioned above. Furthermore, wherever possible, long sentences have been divided into two or more sentences. The manuscript has also been checked for missing commas and where appropriate commas have been added.*

[revised manuscript text omitted]